# Three chromosome-scale *Papaver* genomes reveal punctuated patchwork evolution of the morphinan and noscapine biosynthesis pathway

Xiaofei Yang [1,2,3,9], Shenghan Gao[2,4,9], Li Guo [2,4,5,9], Bo Wang[4], Yanyan Jia[4], Jian Zhou[6], Yizhuo Che[2,4], Peng Jia [2,4], Jiadong Lin[2,4,7], Tun Xu[2,4], Jianyong Sun[8] & Kai Ye [2,3,4,5,7 ✉]

For millions of years, plants evolve plenty of structurally diverse secondary metabolites (SM) to support their sessile lifestyles through continuous biochemical pathway innovation. While new genes commonly drive the evolution of plant SM pathway, how a full biosynthetic pathway evolves remains poorly understood. The evolution of pathway involves recruiting new genes along the reaction cascade forwardly, backwardly, or in a patchwork manner. With three chromosome-scale *Papaver* genome assemblies, we here reveal whole-genome duplications (WGDs) apparently accelerate chromosomal rearrangements with a nonrandom distribution towards SM optimization. A burst of structural variants involving fusions, translocations and duplications within 7.7 million years have assembled nine genes into the benzylisoquinoline alkaloids gene cluster, following a punctuated patchwork model. Biosynthetic gene copies and their total expression matter to morphinan production. Our results demonstrate how new genes have been recruited from a WGD-induced repertoire of unregulated enzymes with promiscuous reactivities to innovate efficient metabolic pathways with spatiotemporal constraint.

[1] School of Computer Science and Technology, Faculty of Electronic and Information Engineering, Xi'an Jiaotong University, Xi'an, Shaanxi, China. [2] MOE Key Lab for Intelligent Networks & Networks Security, Faculty of Electronic and Information Engineering, Xi'an Jiaotong University, Xi'an, Shaanxi, China. [3] Genome Institute, The First Affiliated Hospital of Xi'an Jiaotong University, Xi'an, Shaanxi, China. [4] School of Automation Science and Engineering, Faculty of Electronic and Information Engineering, Xi'an Jiaotong University, Xi'an, Shaanxi, China. [5] School of Life Science and Technology, Xi'an Jiaotong University, Xi'an, Shaanxi, China. [6] College of Life Sciences, Henan Normal University, Xinxiang, Henan, China. [7] Faculty of Science, Leiden University, Leiden, The Netherlands. [8] School of Mathematics and Statistics, Xi'an Jiaotong University, Xi'an, Shaanxi, China. [9]These authors contributed equally: Xiaofei Yang, Shenghan Gao, Li Guo. ✉email: kaiye@xjtu.edu.cn

Metabolic gene cluster (MGC) is a special genetic architecture well known in microorganisms involved in both primary and secondary metabolisms. While MGCs are relatively uncommon in plants, recent studies have discovered over 30 plant secondary metabolites whose biosynthetic pathways are encoded by MGCs, such as thalianol[1], noscapine[2], α-tomatine[3], momilactones[4], DIMBOA (2,4-dihydroxy-7-methoxy-1, 4-benzoxazin-3-one)[5] and taxadiene[6] etc. Furthermore, genomic mining using algorithms such as plantiSMASH[7] and PhytoClust[8] have identified many putative plant MGCs[9]. As more plant genomes being sequenced and analyzed, the list of putative plant MGCs is expected to grow, although how many of these MGCs encode functional metabolic pathways awaits investigation. Functionally related genes in plant genomes are typically unlinked. However, MGCs are exceptional raising questions why they are necessary and widely distributed across multiple kingdoms. One hypothesis for cluster formation is co-inheritance arguing that colocalization of genes prevents losing key pathway genes due to recombination events[10]. Another hypothesis is co-expression suggesting that colocalized genes share common promoters and *cis*-regulatory elements, ensuring a spatially and temporally coordinated expression of biosynthetic enzymes and thus maximizing the yield of end products[11].

Despite discovering plant MGCs, our knowledge is limited regarding the mechanisms and history of their formation, maintenance, and diversification. Formation of microbial MGCs is often through relocation or duplication of native genes[12,13], or horizontal gene transfers[14,15]. By contrast, plants are limited to vertical gene transmission, and thus must have their own as-yet-unknown mechanisms of gene cluster formation. Understanding the evolutionary history of plant MGCs can explain why many secondary metabolites are exclusive to plants of specific lineage or species. For example, opium poppy remains the sole natural source for opiates used to extract morphine for painkilling drugs[16], while the anti-cancer drug paclitaxel (trade name Taxol) mainly exists in the bark of *Taxus* trees[17]. Additionally, some secondary metabolites have apparently undergone convergent evolution in distantly-related plant lineages[18]. Driven by genetic variations, evolution occurs upon selection on traits that benefit organisms under special environment. Therefore, genetic variations and natural selection expectedly play critical roles in the independent evolution of plant MGCs, giving plants a plethora of structurally diverse secondary metabolites of specific bioactivities to support their sessile lifestyle.

Although ample instances of new genes driving the evolution of biosynthetic pathway have been well documented[19–21], how a full cascaded metabolic pathway evolved among different lineages is largely unknown. Recent studies suggest that plant gene clusters may have evolved through the recruitment of genes from elsewhere in the genome via duplications and neofunctionalization[22,23], genomic rearrangement[24,25], transposons[26], and other unknown mechanisms. Three main evolution models of pathway have been proposed during the last decades, including the forward, backward, and patchwork model. The forward model depicts the evolution of the pathway by recruiting enzymes catalyzing forwardly from earlier steps to later steps[27,28], while the backward model describes the pathway evolved by acquiring enzymes in a backward fashion, e.g., from later steps to earlier steps[29]. The patchwork model suggests metabolic pathways are assembled by duplicating genes encoding enzymes reacting with diverse substrates[30,31]. An evolutionary model for MGC-encoded plant metabolic pathway is overall lacking due to quite limited comparative analysis of multiple chromosome-scale plant genomes that are notoriously difficult to assemble. Without taking genomic synteny and large-scale rearrangement events into account, evolutionary analysis of plant gene cluster is likely prone to misleading conclusions on the evolution of pathway.

Opium poppy (*Papaver somniferum* L.) is a medicinal plant producing various benzylisoquinoline alkaloids (BIAs) such as morphine and noscapine[32]. BIA biosynthetic pathway is one of the most completely elucidated in plants, with key enzymes identified and functionally validated[32–34]. We and others previously reported that the biosynthetic pathways of morphinan and noscapine were primarily controlled by a BIA gene cluster of 584 kb in *P. somniferum* genome[24,25], representing one of the largest gene clusters encoding secondary metabolic pathways in eukaryotes. The BIA gene cluster includes a 10-gene cluster for noscapine biosynthesis[2] and five genes (*STORR*[35–37], *SALSYN*, *SALAT*, *SALR*, and *THS*) for morphinan biosynthesis, and also several genes of unknown functions. Genes encoding the biosynthesis of precursor (S)-reticuline (*TYDC, 4OMT, 6OMT, BBE* etc.) as well as converting thebaine to morphine (*CODM, COR,* and *T6ODM*) are unclustered and dispersed in the genome. The morphinan biosynthetic pathway encoded by a single MGC yet producing two distinct subclasses of BIAs, morphinan and noscapine[24,32,35,38–40], presents an ideal model to study the complex evolutionary history of plant MGCs in early-diverging eudicots, given that the BIAs have experienced both natural selection and artificial selection by a human being.

In this work, we de novo assemble two chromosome-level genomes of *P. rhoeas* (common poppy) and *P. setigerum* (Troy poppy) and improve the previous draft *P. somniferum* HN1 genome[24] to gain insights into the evolution of BIA gene cluster among *Papaver* species. The three *Papaver* species are diverse in terms of morphinan and noscapine production level, and WGD events, therefore providing fresh insights into the evolution of the BIA biosynthetic pathway. Through comparative genomic analysis, we infer the evolutionary history of the morphinan biosynthetic pathway and BIA gene cluster which is most likely explained by the punctuated patchwork model.

## Results

**Quantification of noscapine and morphinans in three *Papaver* species**. We evaluated BIA productions of three *Papaver* species by quantifying the noscapine and morphinans (thebaine, codeine, and morphine) in capsules using the HPLC-MS method (Supplementary Methods 1 and 2). *P. somniferum* HN1 cultivar ($2n = 22$)[24] (Supplementary Fig. 1) accumulated the highest amount of morphinan and noscapine (Supplementary Fig. 2). *P. setigerum* DCW1 ($2n = 44$) (Fig. 1a and Supplementary Fig. 1)[41,42] produced no noscapine, and its production of thebaine, codeine, and morphine were only 6.6%, 50%, and 5.5% of what was produced by *P. somniferum* HN1, respectively (Supplementary Fig. 2). *P. rhoeas* YMR1 cultivar ($2n = 14$)[43,44] (Supplementary Fig. 1) produced a trace amount of morphinan and noscapine (Supplementary Fig. 2 and Supplementary Method 3).

**Genome assembly and annotation**. To assemble *P. setigerum* and *P. rhoeas* genomes, we used a combination of sequencing technologies including Oxford Nanopore (ONT) long reads, ONT ultra-long reads, Illumina paired-end reads, and high-throughput chromosome conformation capture (Hi-C) sequencing reads (Supplementary Data 1 and Supplementary Methods 4, 5). The genome assemblies of *P. setigerum* and *P. rhoeas* were highly contiguous with 97.6% and 87.9% of genome contigs anchored to chromosomes by using Hi-C scaffolding, respectively (genome assembly size of 4.6 and 2.5 Gb, scaffold N50 values of 211.2 and 329.4 Mb, contig N50 values of 65.6 and 5.3 Mb) (Table 1 and Supplementary Figs. 3–6, Supplementary Data 2, Supplementary Method 6). Facilitated by Hi-C scaffolding, 92.4% of original *P. somniferum* sequences[24] were anchored on 11 chromosomes with a scaffold N50 of 249.6 Mb, much-improved assembly contiguity

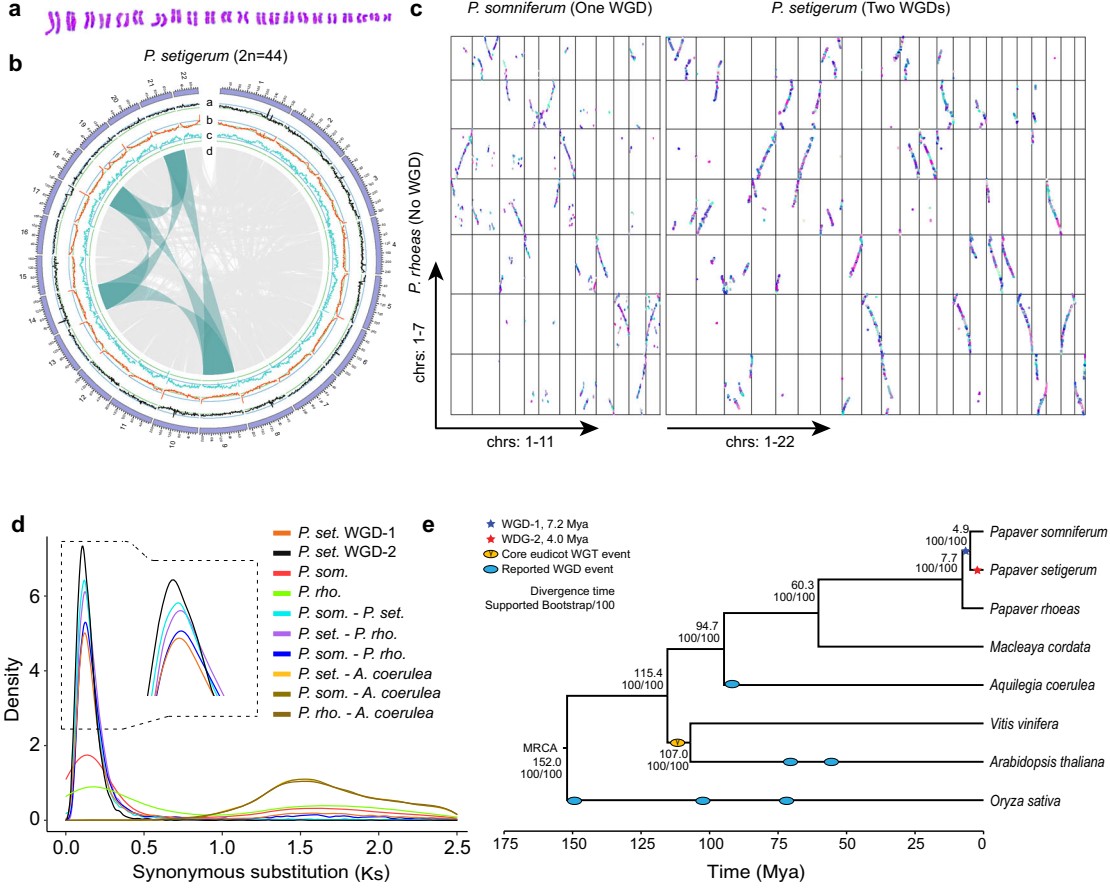

**Fig. 1 Genome features of three *Papaver* species. a** Karyotyping shows *P. setigerum* is diploid (2*n* = 44). **b** Intraspecies synteny analysis reveals two rounds of whole-genome duplications (WGD) in *P. setigerum* genome. The tracks *a*, *b*, and *c* of the circos plot represent the distribution of gene density, repeat density, and GC density, respectively (calculated in 2 Mb windows). Track *d* shows genome-wide syntenic blocks detected by MCScanX[87] where an example of duplications involving four chromosomes as a result of two WGDs was highlighted. Bandwidth is proportional to syntenic block size. **c** Dotplots of interspecies syntenic blocks reveal the 1:2:4 syntenic block ratio of *P. rhoeas*, *P. somniferum*, and *P. setigerum*. Each dot indicates a syntenic gene pair detected by MCScanX. Synteny blocks were colored by MCScanX. **d** Synonymous substitution rate (Ks) density distributions of syntenic paralogs and orthologs, with colored lines representing comparisons among four plant species: *P. set.* (*P. setigerum*), *P. rho.* (*P. rhoeas*), *P. som.* (*P. somniferum*) and *A. coerulea* (*Aquilegia coerulea*)[89]. For *P. setigerum*, only the reciprocal best matches among the syntenic gene pairs were considered as the pairs from WGD-2 while other syntenic gene pairs were grouped as the pairs from WGD-1. **e** Inferred maximum likelihood phylogenetic tree by RAxML[94]. Divergence timings and the supported bootstrap values were labeled on the tree. Estimated WGD-1 and WGD-2 timings and the reported whole-genome triplication (WGT)/WGD timings were superimposed on the phylogenetic tree. MRCA most recent common ancestor, Mya million years ago. Source data underlying **b**–**e** are provided as a Source Data file.

over the original version[24] (Table 1 and Supplementary Fig. 7, Supplementary Data 2, Supplementary Method 6). After reassignment of chromosome IDs, we found the BIA gene cluster was located at chr4 rather than chr11. Benchmarking Universal Single-Copy Orthologs (BUSCO)[45] evaluation revealed the high genome completeness (92.8–95.3%) (Table 1, Supplementary Fig. 8). We annotated 41,407 to 106,517 protein-coding genes and 12,429–23,109 non-coding RNAs for the three *Papaver* species (Table 1 and Supplementary Figs. 9, 10, Supplementary Method 7). Repetitive elements make up more than 70% for each *Papaver* genome, and over 50% of them are long terminal repeat retrotransposons (LTRs) (Table 1 and Supplementary Fig. 11, Supplementary Method 7).

**Whole-genome duplication events in three *Papaver* species.**
Synteny analysis revealed two rounds of WGDs in *P. setigerum* but no WGD in *P. rhoeas*, and confirmed the reported single WGD in *P. somniferum*[24] (Fig. 1b, c and Supplementary Figs. 12–16, Supplementary Method 8). In addition, a collinearity

analysis of three *Papaver* species with grape (*Vitis vinifera*)[46] and ancestral eudicot karyotype genome[47] (Supplementary Figs. 17 and 18) confirmed a lack of the whole-genome triplication (γ event) occurred in core eudicots in *Papaver* genus.

We next investigated whether *P. somniferum* and *P. setigerum* shared any WGD. Using the estimated divergence time of around 7.7 million years ago (Mya), which is consistent with TimeTree[48], and mean synonymous substitutions per synonymous site (Ks) of syntenic gene pairs between *P. somniferum* and *P. rhoeas*, we estimated the mutation rate as $8.1 \times 10^{-9}$ synonymous substitutions per site per year for *Papaver*. Based on this rate and Ks distribution, the *P. somniferum* WGD occurred at around 7.2 Mya, within the range of previous estimation[24] (Fig. 1d and Supplementary Data 3, Supplementary Method 8). Phylogenomic analysis of eight angiosperms using 48 single-copy orthologs identified by OrthoFinder[49] indicated *P. setigerum* and *P. somniferum* diverged at around 4.9 Mya (Fig. 1e and Supplementary Method 8), following the divergence of their common ancestor from *P. rhoeas* at around 7.7 Mya. Since the *P. somniferum* WGD predated the divergence of *P. somniferum* and

**Table 1 The statistics of assemblies and annotations of three Papaver genomes.**

| | P. setigerum | P. rhoeas | P. somniferum |
|---|---|---|---|
| Contig | | | |
| Total number of contigs | 553 | 3,273 | 61,801 |
| Assembly size (MB) | 4590.03 | 2541.84 | 2709.32 |
| Contig N50 (MB) | 65.57 | 5.29 | 1.74 |
| Contig N90 (MB) | 13.37 | 0.61 | 0.12 |
| Largest contig (MB) | 178.78 | 39.02 | 13.77 |
| Scaffold | | | |
| Total number of scaffolds | 381 | 237 | 55,380 |
| Assembly size (MB) | 4590.12 | 2542.27 | 2712.53 |
| Scaffold N50 (MB) | 211.16 | 329.41 | 249.6 |
| Scaffold N90 (MB) | 155.08 | 33.61 | 172.47 |
| Largest scaffold (MB) | 329.11 | 361.76 | 328.07 |
| BUSCO for genome | 94.50% | 92.80% | 95.30% |
| GC content | 36.88% | 37.94% | 37.28% |
| Number of gaps | 172 | 3,036 | 6,421 |
| Annotation | | | |
| Number of protein-coding genes | 106,517 | 41,470 | 55,316 |
| Supported by RNA-seq or homologs | 100% | 100% | 100% |
| Supported by protein family | 70.56% | 70.14% | 70.05% |
| Repeat density | 71.55% | 73.92% | 76.68% |
| Number of ncRNA | 23,109 | 12,429 | 12,636 |

*P. setigerum* (4.9 Mya), this WGD is essentially the same event as the first WGD (WGD-1) in *P. setigerum* which has further underwent a lineage-specific WGD (WGD-2) dated around 4.0 Mya (Fig. 1d, e and Supplementary Data 3).

**Ancestral genomes and accelerated non-random post-WGD rearrangements**. Rearrangements following WGD events are common in plant genome evolution. We are curious about what genome structural changes led to the present-day karyotypes of three *Papaver* species. We adopted the computational strategy proposed by Sankoff et al.[50,51] and developed a bottom-up workflow to reconstruct pre- and post-WGD-1 ancestors for three *Papaver* species (Supplementary Figs. 19, 20 and Supplementary Method 9). Ancestral genome reconstruction relies on accurate genome assembly, and potential misassembly confounds the reconstruction and downstream analysis. Based on the high-quality genome assemblies of three *Papaver* species, the reconstructed pre- and post-WGD-1 ancestral genome had six and eleven protochromosomes, respectively (Fig. 2a and Supplementary Fig. 21). Compared to pre-WGD-1 ancestral genome, *P. rhoeas* likely needed at least five chromosomal fissions and four chromosomal fusions to reach its current structure of seven chromosomes (Fig. 2a and Supplementary Fig. 21). By contrast, *P. setigerum* experienced a much more complex evolutionary history involving two rounds of WGDs and post-WGD rearrangements that finally shaped its karyotype of 22 chromosomes. Shared with *P. somniferum*, *P. setigerum* likely underwent at least 11 chromosomal fissions and 12 chromosomal fusions after WGD-1, and then at least 20 chromosomal fissions and 20 chromosomal fusions following a lineage-specific WGD-2 (Fig. 2a).

We observed a non-random distribution of chromosomal fissions and chromosomal fusions associated with the transformation of the reconstructed pre-WGD-1 protochromosomes to modern chromosomes (Fig. 2). Three chromosomes (chr7, chr16, and chr19) in *P. setigerum* were significantly enriched with fusion events, while chr21 was significantly depleted of them (*p*-value < 0.05, *z*-test, Fig. 2a). Similarly, in pre-WGD-1 ancestor,

protochromosome 4 was significantly enriched with fission events, whereas protochromosome 2 was significantly depleted of them (*p*-value < 0.05, *z*-test, Fig. 2b). In addition, the numbers of WGDs and fission events needed to shape the current karyotypes in three *Papaver* species have a superlinear correlation (Fig. 2b), indicating that post-WGD genome rearrangements might have been accelerated. WGD plays a significant role in the plant secondary metabolism evolution[52] inspiring us to investigate the function of post-WGD diploidization in *Papaver* species. The genes around the shuffling breakpoints in *P. somniferum* were enriched in KEGG (Kyoto Encyclopedia of Genes and Genomes) pathways related to isoquinoline and indole alkaloid biosynthesis, as well as the metabolism of amino acids, such as tyrosine and phenylalanine, whereas breakpoint-vicinity genes in *P. setigerum* were enriched with plant-pathogen interactions and environmental adaptation (Supplementary Fig. 22). These results suggest a post-WGD diploidization might have played a part in the optimization of the biosynthetic pathways of alkaloids such as morphinans in the ancestor of *P. somniferum* and *P. setigerum*.

**Recruitment of new genes to BIA gene cluster locus**. Although the chromosomal shuffling events affected isoquinoline alkaloid metabolism, we did not observe their impact on the BIA gene cluster (Fig. 2a and Supplementary Data 4, 5, Supplementary Method 10). Therefore, we compared three *Papaver* genomes to identify additional structural variation events potentially leading to the formation of the BIA gene cluster. *STORR* encodes a fusion protein of a cytochrome P450 and an oxidoreductase (Supplementary Fig. 23), and enables the gateway reaction towards morphinan biosynthesis[35–37]. If the deletion of the intergenic region between pre-fusion modules is the sole event causing *STORR* fusion, one should expect collinearity between loci encompassing *STORR* and its pre-fusion modules. However, such collinearity was observed in neither *P. somniferum* nor *P. setigerum* (Supplementary Figs. 12 and 24), intriguing us to hypothesize that *STORR* evolution may involve additional translocation besides the fusion event with unknown order, the so-called "fusion, translocation" (FT) event.

We then systematically examined two genomic loci implicated in the translocation event of the three species and traced the evolutionary history of *STORR*. We defined the donor loci as genomic regions syntenic with the *P. somniferum* regions containing the pre-fusion modules, and the recipient loci as genomic regions collinear with the *P. somniferum* regions harboring the *STORR* locus (Fig. 3a and Supplementary Figs. 25, 26). In addition, donor loci carrying the pre-fusion modules represent the prior state of the FT event, while other donor loci denote the post-state. As for recipient loci, the ones carrying *STORR* represent the post-state, while the others represent the prior state (Fig. 3a and Supplementary Figs. 25, 26). We found all four types of loci appeared exactly once in *P. somniferum*, but twice in *P. setigerum*, while only the prior states of donor and recipient loci were observed once in *P. rhoeas* (Fig. 3a and Supplementary Figs. 25, 26, Supplementary Methods 11), supporting our hypothesis.

Considering the phylogeny and WGD history among the three species, we proposed a parsimonious evolutionary model to illustrate a burst of genomic rearrangements giving the birth of *STORR* to the current BIA cluster (Fig. 3b). The ancestor of three species contains the pre-fusion modules at donor loci and an empty recipient locus, which is preserved in *P. rhoeas*. After divergence from *P. rhoeas*, WGD-1 at around 7.2 Mya resulted in two copies of pre-fusion modules, of which one copy was converted to *STORR* by a FT event and inherited by *P. somniferum*. After divergence from *P. somniferum*, *P. setigerum*

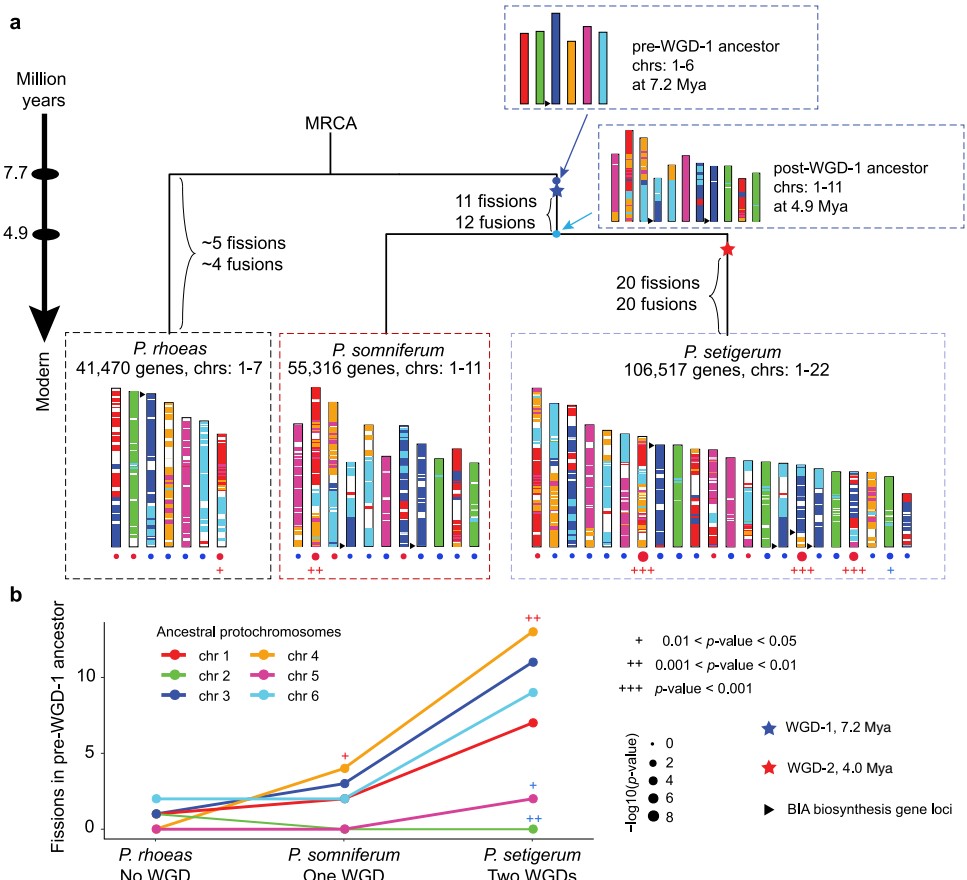

**Fig. 2 Ancestral genomes and accelerated non-random post-WGD rearrangements. a** Both ancestral and modern genomes were illustrated with a six-color code corresponding to protochromosomes of pre-WGD-1 ancestor. Under each modern chromosome, red circles indicate enriched fusions, while blue circles indicate depletion of fusions. MRCA most recent common ancestor, WGD whole-genome duplication, Mya million years ago. **b** Number of fissions on each pre-WGD-1 ancestor protochromosomes to shape three modern genomes. The *p*-value of enrichment and depletion were marked as red and blue "+", respectively, *p*-value is calculated by *z*-test. The *p*-values for protochromosome 4 are 0.034 and 0.005 in *P. somniferum* and in *P. setigerum*, respectively; the *p*-value for protochromosome 2 is 0.004 in *P. setigerum*; the *p*-value for protochromosome 5 is 0.02 in *P. setigerum*. Source data are provided as a Source Data file.

underwent a lineage-specific WGD-2 at around 4.0 Mya, yielding two copies of *STORR* at recipient loci and two copies of pre-fusion module at donor loci (Fig. 3b). Based on comparative analysis of three *Papaver* genomes, current computational strategies, and parsimonious assumption, our *STORR* evolutionary model represents a working hypothesis. The inclusion of sequencing data from additional species may lead to an updated evolutionary model or alternative hypotheses.

We next investigated the evolutionary history of the remaining genes in the BIA gene cluster. Unlike previous analyses focusing on gene trees[53], we integrated evidence of multiple sources from synteny, phylogeny, protein sequence alignments, and WGD (Fig. 4a and Supplementary Figs. 27–36, Supplementary Methods 11, 12) and found that four genes (*PSSDR1, CYP82X1, CYP719A21,* and *PSMT1*) in noscapine branch and two (*SALSYN* and *SALR*) in morphinan branch were already present in MRCA of three *Papaver* species (Fig. 4b). Three genes (*STORR, SALAT,* and *THS*) in the morphinan branch were assembled before *P. somniferum* diverged from *P. setigerum*, while the assembly of noscapine branch was completed in *P. somniferum* via adding six new genes (*PSCXE1, CYP82X2, PSAT1, PSMT2, CYP82Y1,* and *PSMT3*) through lineage-specific duplications (Fig. 4b and Supplementary Figs. 27–35, Supplementary Method 11). Specifically, we inferred five genes including *PSCXE1, PSAT1, PSMT3, SALAT,* and *THS* may be assembled into the BIA gene cluster by

putative dispersed duplications (the gene and their inferred original copy are non-syntenic and not adjacent) while *CYP82X2* was likely generated from a tandem duplication of *CYP82X1* (Fig. 4b and Supplementary Figs. 27–35). Considering the difficulty of distinguishing dispersed duplication from old tandem duplication with gene deletions, we do not rule out the possibility that *PSCXE1, PSAT1, PSMT3, SALAT,* and *THS* were assembled into BIA gene cluster by ancient tandem duplications with follow-up gene deletions (Supplementary Fig. 37).

**Clustered BIA genes are co-regulated and evolved in a coordinated manner.** Assembly of 15 genes encoding two distinct pathways into a compact gene cluster is striking. The genetic components of the BIA pathway should allow the biosynthesis of morphinan and noscapine albeit at their original loci before clustering, raising questions on the necessity of gene clustering in evolution. In fact, besides the BIA gene cluster, the phenomenon of gene clustering has been reported for multiple plant secondary metabolic pathways[54]. This increasingly common theme suggests that plants, with a sessile lifestyle, are under selection pressure to evolve a special genetic architecture to facilitate their adaptation to environmental stimuli[55], consistent with our gene family analysis (Supplementary Data 6–8). Recently, we and others have reported that gene clustering likely enables the coordinated

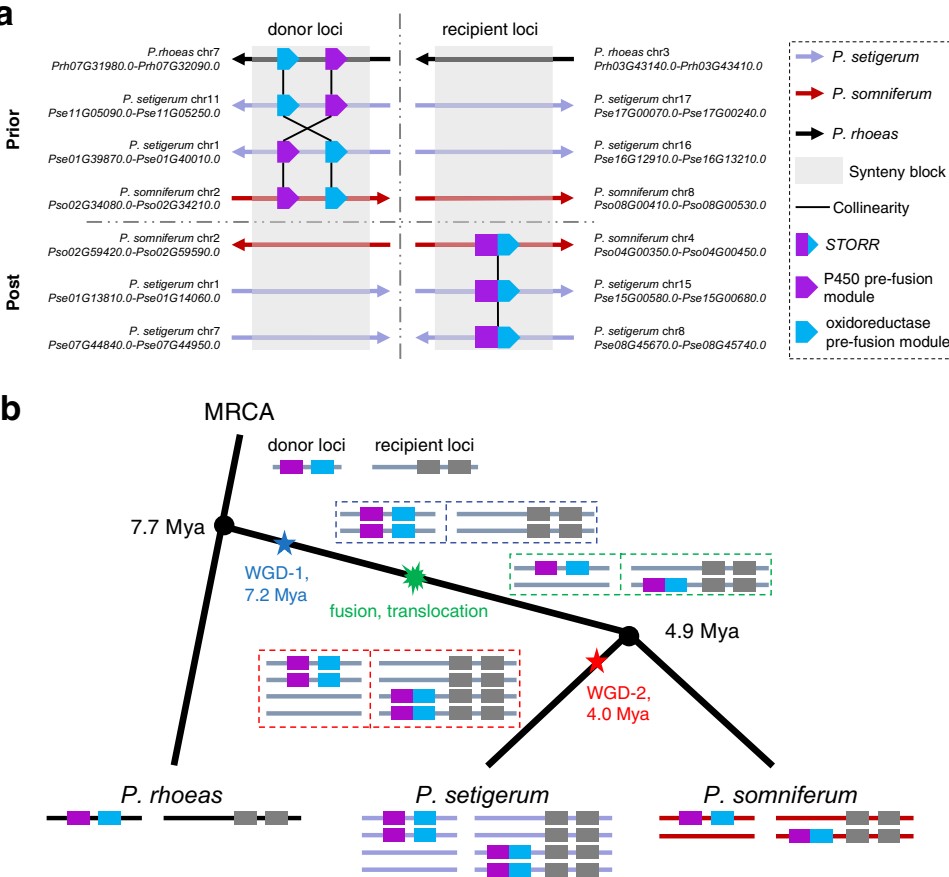

**Fig. 3 Evolutionary history of *STORR*. a** The collinearity of genes in the donor loci and the recipient loci showing both prior and post states of FT event in three *Papaver* species supporting the "fusion, translocation" (FT) event leading to *STORR* formation at BIA gene cluster. The directions of arrows indicated the chromosome from 5′ to 3′. The dash lines indicate the boundary between donor loci and recipient loci, and the boundary between prior states and post states. The detailed diagram with the genomic context was shown in Supplementary Fig. 25. **b** A proposed evolutionary model for the birth of *STORR* at BIA gene cluster raised from a series of whole-genome duplications (WGD), "fusion, translocation" events. Purple and blue colors denote the cytochrome P450 and oxidoreductase modules. MRCA most recent common ancestor, WGD whole-genome duplication, Mya million years ago. Source data underlying **a** are provided as a Source Data file.

expression of metabolic enzymes for BIA biosynthesis in opium poppy stem and root tissues[24,56]. However, it remains unclear how *Papaver* species achieve the coordinated BIA gene expression by evolving this gene cluster. To approach this question, we dissected the tissue-specific expression profiles of the BIA gene cluster and its closest paralogs within *P. somniferum* and *P. setigerum* (Supplementary Method 13). For *STORR* gene-related loci, we found that the donor locus had low expression levels across all tissues among three species while the recipient locus exhibited high expression in the stem where morphinan biosynthesis primarily occurs[24] (Fig. 5a and Supplementary Fig. 38). Expression pattern of the other eight new genes at the BIA gene cluster and their ancestral copies suggests that except for a putative tandem duplicated gene *CYP82X2*, all were duplicated from putative ancestral copies at remote origin loci of largely low and non-tissue-specific expression to the BIA gene cluster displaying a high and coordinated expression in stem (Supplementary Fig. 38). The fact that genes at recipient prior locus were also expressed in stems (Fig. 5a and Supplementary Fig. 38) indicated that new genes arrived at the BIA gene cluster were perhaps pre-equipped with a stem-specific promoter or regulatory elements (Supplementary Data 9 and 10).

We further sought potential epigenetic factors contributing to the co-expression of BIA gene cluster. Hi-C data showed *P. somniferum* noscapine branch and morphinan branch were in

two different chromatin blocks physically interacting likely through a chromatin loop as shown by Hi-C contact maps (Supplementary Fig. 39 and Supplementary Method 14), indicating simultaneous regulation of both pathway branches via an epigenetic regulatory mechanism. Although *P. setigerum* lineage-specific WGD-2 created a second copy of the morphinan branch, only one copy is highly expressed in stem. The highly expressed copy on chr15 has significantly more chromatin contacts than the lowly expressed copy on chr8 ($p$-value $= 6.18e-4$, Fig. 5b, c and Supplementary Fig. 40). Although coding regions of five genes involved in morphinan biosynthesis remain intact in both copies in *P. setigerum*, the promoter regions for five genes in the lowly expressed copy were eroded but were overall maintained by the highly expressed copy (Fig. 5d and Supplementary Fig. 41). This is unexpected because the chance of promoter erosion occurring at either copy should be almost equal unless the locus-specific promoter conservation is favorably selected. Thus, such findings indicated that the observed physical clustering of co-expressed and co-regulated genes is under strong positive selection for a beneficial trait, which keeps the cluster evolved in a coordinated manner.

**Total expression of all gene copies contributes to morphinan production.** Copy number variation of BIA biosynthetic genes

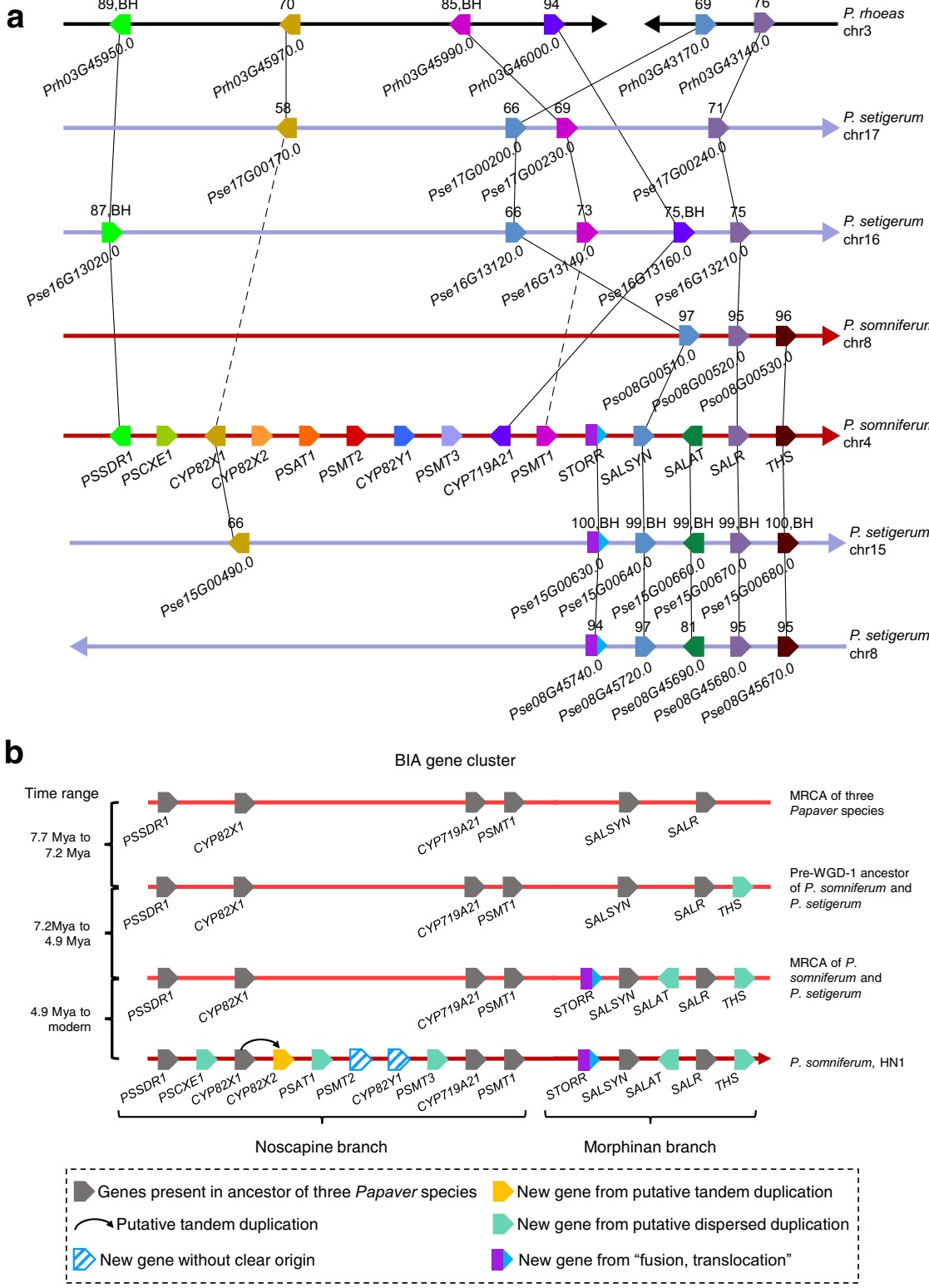

**Fig. 4 Evolutionary history of benzylisoquinoline alkaloid (BIA) gene cluster. a** The syntenic relations at BIA gene cluster in three *Papaver* species. Solid lines denote the syntenic relations detected by MCScanX, and the dash lines denote the top BlastP hits. Different colors denote different genes. The gene name or ID is labeled for each gene. The numbers above the genes represent sequence identity levels from BlastP. The directions of arrows indicate the chromosome from 5′ to 3′. BH best hit. **b** Summary of BIA gene cluster evolution. Putative tandem duplication as a structural variation event producing identical adjacent segments while the other non-locally duplicated ones are "putative dispersed duplication". MRCA most recent common ancestor, WGD whole-genome duplication, Mya million years ago. Source data underlying **a** are provided as a Source Data file.

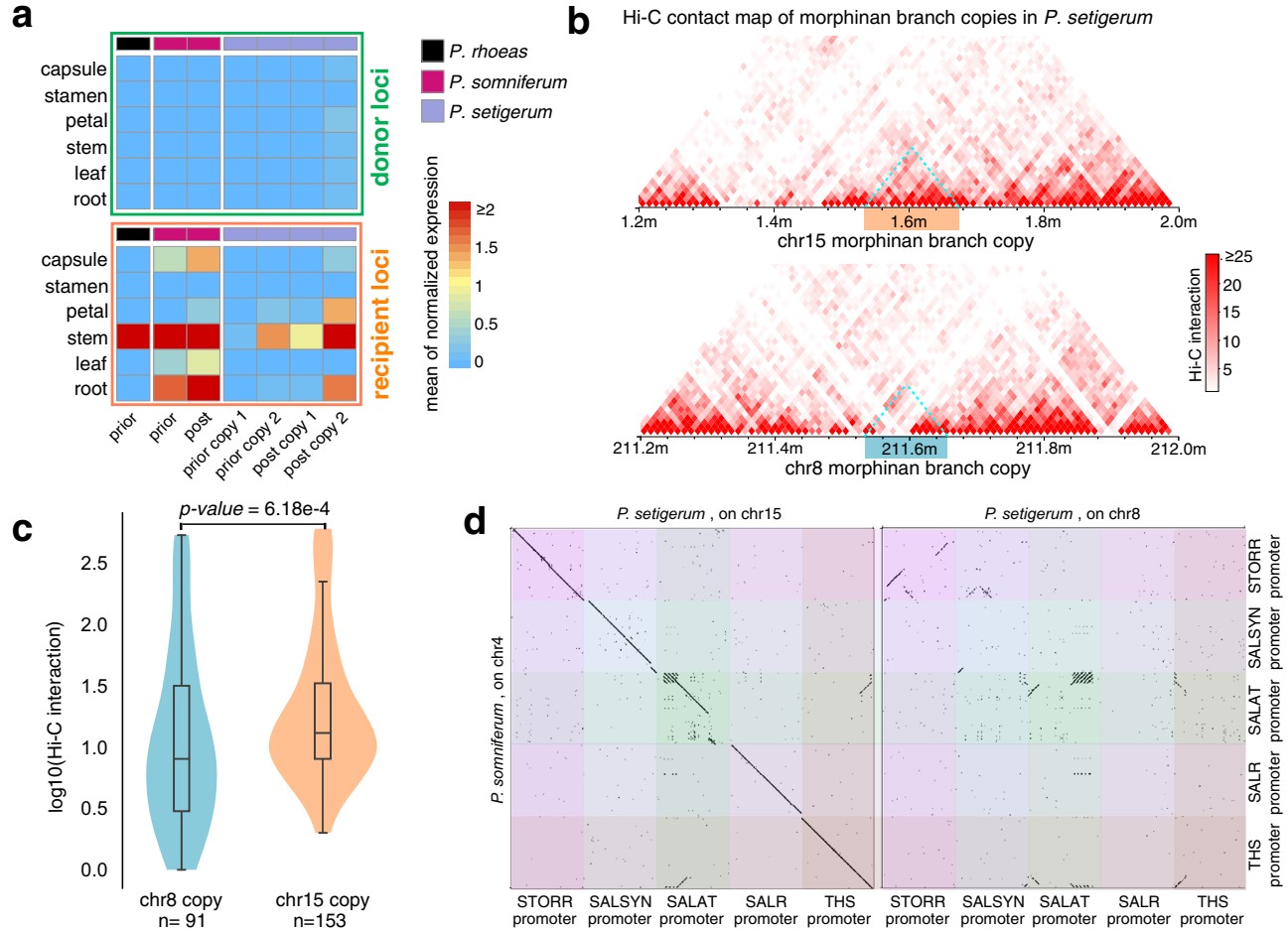

**Fig. 5 Clustered BIA genes are co-regulated and evolved in a coordinated manner. a** The heatmap of the average normalized expression level of genes at the donor and the recipient loci related with *STORR* in six tissues of three *Papaver* species. Details of the gene expressions related to BIA genes were shown in Supplementary Fig. 38. **b** Hi-C interaction heatmap of regions including two copies of morphinan gene cluster on chr15 (top) and chr8 (down) of *P. setigerum*, shown at a resolution of 10 kb. The morphinan branch gene regions are marked with colored boxes. **c** The comparison of the interactions between two morphinan gene copies in *P. setigerum*. The *p*-value is calculated by a two-sided Wilcoxon rank-sum test. For the boxplot, the centerline, median; box limits, upper and lower quartiles; whiskers, data range. *n* is the number of bins. **d** Dotplots of *STORR, SALSYN, SALAT, SALR*, and *THS* promoter sequences between *P. somniferum* (one copy) and *P. setigerum* (two copies). Promoter is defined as 2 kb upstream region of the transcription start site. Source data are provided as a Source Data file.

has been positively correlated with the production of morphinans in *P. somniferum* cultivars in a previous genomic study[56], indicating a potential gene dosage effect. However, we found the extra copy or copies of *STORR, SALSYN, SALAT,* and *SALR* in *P. setigerum* due to WGD-2 did not lead to more abundant production of morphinans compared to *P. somniferum* (Fig. 6 and Supplementary Fig. 2). Moreover, the eroded promoters of one copy of the morphinan gene cluster in *P. setigerum* caused low expression, while the intact promoter of the second copy maintained high expression of genes (Supplementary Method 13). Therefore, the copy number is not the only factor affecting BIA productions, and gene expression may also play an important role. We systematically compared the copy number and total expression of all copies of BIA genes encoding enzymes for the morphinan biosynthetic pathway in the three species. We observed that four genes encoding key enzymes, including *STORR, SALAT, THS,* and *T6ODM*, were missing in the corresponding syntenic blocks of *P. rhoeas* YMR1 (Fig. 4, Supplementary Fig. 42), probably leading to its negligible amount of three morphinans compared to *P. somniferum* HN1 (Fig. 6 and Supplementary Fig. 2). By contrast, compared to *P. somniferum* HN1, *P. setigerum* DCW1 carries twice as many copies of *STORR*,

*SALSYN, SALR,* and *SALAT*, and equal copy number of *THS* and *T6ODM*, but fewer copies of *COR* and *CODM* catalyzing the final step to codeine and morphine production (Fig. 6 and Supplementary Fig. 42, Supplementary Data 5). The pure difference of gene copy numbers between the two species could not explain the large gap in their morphinan productions. Thus, we compared the total expression of all copies of each morphinan branch gene in the stem where morphinan biosynthesis primarily takes place[24,56], and found their expression in *P. somniferum* HN1 was at least twice as high as that in *P. setigerum* DCW1 (Fig. 6 and Supplementary Fig. 42, Supplementary Data 5). As the morphinan biosynthesis occurs in a cascade fashion, we expect the total expression of all copies of each gene and its copies along the pathway to yield an amplified difference in the gross production of the final products, which is exactly what we observed in the two *Papaver* relatives (Fig. 6). As such, these results suggest the morphinan production is attributed to the total expression of all copies of key genes besides their copy numbers. Finally, distinct copy numbers for *CODM, T6ODM,* and *COR* have been observed among three *Papaver* species (Fig. 6 and Supplementary Fig. 42) and among cultivars of *P. somniferum*[56], suggesting that dosage of these genes is possibly under positive selection.

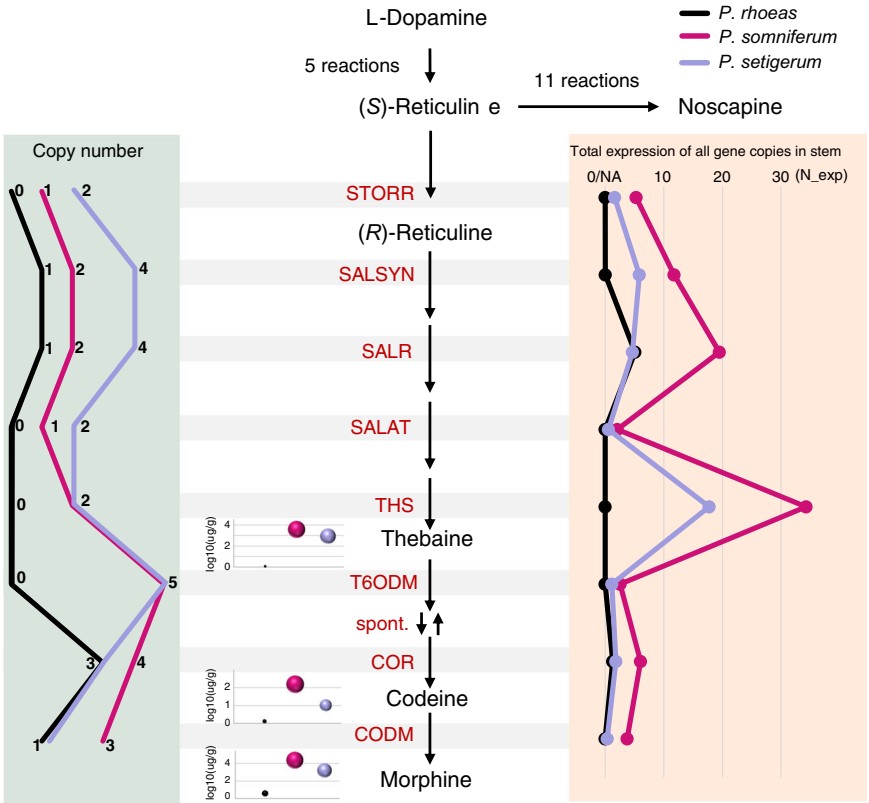

**Fig. 6 Copy number and expression of genes on morphinan biosynthetic pathway.** Correlation of morphinan production with copy number and expression of morphinan biosynthesis genes in three *Papaver* species. Copy numbers of morphinan biosynthesis pathway genes were shown in a line plot on left. Production levels of thebaine, codeine, and morphine were shown in bubble plots, and the detail numbers were shown in Supplementary Fig. 2. The total expression of all gene copies in the stem was shown in a line plot on right. N_exp normalized expression. Source data are provided as a Source Data file.

## Discussion

Gene clustering, a frequent genetic paradigm in microbial metabolic pathways, has long been thought as a rarity in higher eukaryotes such as land plants. Until recently, with the burst of plant genome sequencing projects and advancement of bioinformatic tools, biosynthetic pathways for over 30 plant natural products are known to be encoded by gene clusters[55], including *A. thaliana* (e.g., thalianol)[25], opium poppy (e.g., morphinan)[2,24], maize (e.g., DIMBOA)[57], rice (e.g., momilactones)[4], and tomato (e.g., α-Tomatine)[3], etc. Our studies on opium poppy, and recent examples shown by other groups[56] have demonstrated that the formation and diversification of biosynthetic gene clusters in plants have been made possible largely by gene gain and loss through structural variations[58].

In this study, the contribution of large-scale genome variations, e.g., WGD and subsequent genome rearrangements, to the rise and divergence of biosynthetic pathways is well illustrated by the reconstructed evolutionary history of *STORR* and other key BIA genes. Three high-quality *Papaver* genomes and their distinct WGD status enabled us to infer the origins and steps leading to the current BIA gene cluster. We showed that duplication was a major factor contributing to the formation of the BIA gene cluster. The origin and putative type of the duplications were inferred based on synteny, WGD, and protein sequence alignments analysis of three *Papaver* genomes. Alternative explanations of the BIA gene cluster formation such as "old tandem duplications" are equally possible but shall be tested with more sequencing data and genome analyses. Interspecies comparison of the morphinan production revealed that total expression of gene copies of BIA genes contributed to the morphinan production. Consistently, the importance of gene copies for the BIA

biosynthetic genes was also shown by an intraspecies comparison of nine *P. somniferum* cultivars[56]. Therefore, gene copies are pivotal to both intraspecies and interspecies variation of morphinan production in *Papaver*.

*Papaver* species produce diverse types of natural products besides morphinans and noscapine. For example, Menéndez-Perdomo et al. summarized 44 authentic BIAs in *P. somniferum*, including salutaridine, papaverine, and narcotoline[59], and Grauso et al. summarized 128 organic compounds isolated from *P. rhoeas*, including leucine and rhoeadine[60]. The fact that biosynthetic pathways of the non-morphinan and non-noscapine compounds are poorly understood in *Papaver* species limited our further investigation of their metabolic pathways in this study. Thus, we focused on the evolutionary steps leading to the efficient production of the morphinans and noscapine in this interspecies comparison. Expectedly, *P. rhoeas* which produces a trace amount of morphinans and noscapine lacks a bona fide BIA gene cluster. We found that *P. rhoeas* has a primitive metabolic pathway leading to the formation of morphinan precursors such as dopamine, 4-Hydroxyphenylacetaldehyde, and (*S*)-Norco-claurine encoded by *TYDC* (*PrhUNG23530.0*) and *TryAT* (*Prh05G30560.0*), *NCS* (*PrhUNG12800.0*), respectively. Given the phylogenetic relationships of the three *Papaver* spp., it suggests that an ancient repertoire of unregulated enzymes with promiscuous reactivities might be already present in *Papaver* to allow the production of morphinans at minimal levels. WGD-1 apparently energized the ancestral genome and triggered chromosomal rearrangements for boosting the production levels of beneficial metabolites. As a result, a set of new genes originated from fusions and gene duplications were recruited to a locus pre-equipped with desired regulatory elements, leading to the

formation of the BIA gene cluster. The formation of bifunctional STORR enzyme as a result of gene fusion channels metabolic flow preferentially into the morphinan pathway. Recruitment of subsequent enzymes for both morphinan and noscapine pathways as they were branching included neither forwardly nor backwardly evolving progression of evolutionary steps, rejecting both hypotheses of forward pathway evolution[61] and retrograde evolution of biochemical pathway[29]. Instead, once a relatively large repertoire of enzymes becomes available after WGD-1, duplicated promiscuous enzymes can be readily recruited from various genomic regions into a locus pre-equipped with tissue-specific regulatory elements to maximize flux through morphinan and noscapine pathways, as elegantly explained by the patchwork model, which describes the way of recruitment of enzymes in a pathway like a patchwork manner[30,31,62]. There are extensive instances supporting the patchwork model in bacteria. For example, the pathway for degradation of pentachlorophenol, in *Sphingomonas chlorophenolica*[63] and metabolic pathway for 2CNB degradation in *Pseudomonas stutzeri*[64]. By contrast, an evolutionary model for metabolic pathways in eukaryotes is largely unknown[54]. We presented evidence that the evolution of the BIA biosynthetic pathway encoded by a metabolic gene cluster, could be well explained by the patchwork model.

With some exceptions, metabolic gene clustering often enables a coordinated expression of the biosynthetic genes in a tissue-specific manner. We showed in this study that post-WGD genome rearrangement played a major role in achieving this tissue-specific co-expression of the morphinan gene cluster for *P. somniferum* and *P. setigerum*. We showed that new genes were recruited from lowly expressed genomic loci to the pre-configured locus with a stem-specific expression. It remains a mystery why such a low entropy of tightly packed gene cluster ever exists and evolves coordinative, as it is common to find genes at various genomic loci encoding the same pathway, functionally linked via gene co-regulation. In fact, unclustered biosynthetic genes are common in plants that somehow are co-expressed in specific tissues. One example is the flame lily (*Gloriosa superba*) colchicine biosynthetic pathway encoded by genes showing no apparent genomic clustering but strongly co-expressed in seeds and corms[65]. For opium poppy, genes such as *CODM* and *T6ODM* encoding the tailoring enzymes in morphinan biosynthesis are also unclustered but co-expressed with clustered core genes encoding the morphinan pathway even though located on different chromosomes[24,56]. Among plant metabolic gene clusters, the genetic architecture of a core gene cluster plus several peripheral or satellite genes is rather common[9,55]. The gene clusters can be continuous without any intervening genes such as avenacin (oat) gene cluster[34], while some have genes of unknown function separating the cluster genes such as thalianol[25], DIMBOA[57], and morphinan gene cluster[2,24]. There are also biosynthetic pathways such as tomatine, morphine, and cucurbitacin C encoded by a core cluster with satellite genes or cluster[55]. The morphine biosynthetic pathway is composed of a core cluster coupled with several peripheral genes (in trans) or satellite gene groups[24]. Recent studies also show that collinearity exits within a single gene cluster such as the noscapine gene cluster where the cluster is organized in modules that correspond to early, middle and late steps of the pathway[2]. The diverse architecture of metabolic gene clusters shows the evolutionary history of gene clustering for secondary metabolic pathways is complex and species-specific, involving complicated genomic variations and selection processes. Yet how plant biosynthetic genes, either clustered or dispersed in genome, are co-regulated at both epigenetic and transcriptional levels remains largely unknown and needs further investigation. We presented substantial evidence that epigenetic regulations and a non-random

erosion of *cis*-elements within the gene cluster on chr8 in *P. setigerum* underlies the efficient biosynthesis of BIA. The fusion protein STORR couples two reactions, forming the core of a so-called "metabolon" together with co-expressed subsequent morphinan biosynthetic enzymes, to maximize the efficiency of morphinan production but minimize diffusion of toxic intermediate products. The lately formed noscapine pathway joined the locus, perhaps exploiting the optimized regulatory elements and chromatin-level regulation, further expanding the production line of the secondary metabolite factory.

Finally, as we and others have reported the structural variants such as copy number variants within morphinan biosynthetic genes[24,56], opium poppy genome as well as its morphinan gene cluster are not static and probably still evolving given the natural and human artificial selection process. Thus, it would be interesting to observe how biosynthetic pathways continue to evolve due to human selection, and whether biosynthetic genes that are clustered but did not co-express, and genes that co-expressed but non-clustered will have a converged or diverged evolutionary trajectory in future opium poppy crops.

## Methods

**Plant materials, DNA, and RNA isolation**. *P. setigerum* variety DCW1, *P. rhoeas* variety YMR1, and *P. somniferum* variety HN1 were grown in Azalea pots in a growth chamber with 16 h of light located at Xi'an Jiaotong University Laboratory of BioData Sciences. Genomic DNA was extracted from fresh leaves (the four uppermost ones) harvested from 6-weeks-old seedings of three *Papaver* species. For Illumina and regular Oxford Nanopore (ONT) sequencing, high molecular weight (HMW) genomic DNA was prepared by the CTAB method and purified with QIAGEN® Genomic kit (Cat#13343, QIAGEN). For ultra-long ONT sequencing, ultra-high molecular weight DNA was extracted by the SDS method[66] handled gently to sustain the length of DNA. For transcriptome sequencing (RNA-seq), Total RNA was extracted from the six tissue types on the first day of anthesis: root, leaves, stem (the 2-cm-long part just underneath the capsule), capsule, petal, and stamens, using TRIzol reagent (TIANGEN). RNA integrity was determined using regular agarose gel electrophoresis, Nanodrop (ThermoFisher Scientific), and Agilent 2100 Bioanalyzer (Agilent Technologies). RNA sample of high quality (OD260/280 within range [1.8, 2.2], OD260/230 ≥ 2.0, RIN ≥ 8, >1 μg) was used to construct the sequencing library.

**Genome and transcriptome sequencing**. DNA and RNA sequencing in this study were performed using sequencers at Nextomics Inc. (Wuhan, China). For ONT sequencing, about 4 and 10 μg high molecular weight DNA was used for regular and ultra-long library construction, respectively. About 700 ng (regular) and 800 ng (ultra-long) DNA libraries were sequenced on a Nanopore PromethION sequencer. A total of ten and eight ONT cells were used for *P. setigerum* and *P. rhoeas*, respectively. For Illumina paired-end sequencing, a total of 1.5 μg DNA was used to construct the library and the libraries were sequenced by using the Illumina NovaSeq platform to generate 150 bp paired-end reads with insert size around 400 bp. Hi-C library construction was performed according to a 3C protocol established for maize[67]. Briefly, 5 g fresh leaves were fixed with 1% formaldehyde solution in MS buffer (10 mM potassium phosphate, pH 7.0, 50 mM NaCl; 0.1 M sucrose), and homogenized with liquid nitrogen, and then subjected to nuclei isolation and enrichment. Chromatin was digested for 16 h with 400 U *Hin*dIII restriction enzyme (New England Biolabs) at 37 °C. DNA ends were labeled with biotin and DNA ligation was performed by adding T4 DNA ligase (New England Biolabs, USA) and incubation at 16 °C for 4–6 h, followed by reverse cross-linking using proteinase K (Promega, USA). Purified DNA was fragmented to a size of 300–500 bp, and DNA fragments labeled by biotin were finally separated on Dynabeads® M-280 Streptavidin (Life Technologies, USA). Hi-C libraries were controlled for quality and sequenced on an Illumina Novoseq sequencer. For RNA-seq, a total amount of 1 μg RNA was used for library construction with the TruSeq RNA Library Preparation Kit (Illumina, USA). The RNA-seq libraries were sequenced on an Illumina NovaSeq platform and paired-end reads of 150 bp were generated.

**Genome assembly**. To obtain chromosome-scale genome assemblies, all sequencing reads were first subjected to quality control. The ONT reads were assembled into primary contigs by using NextDenovo (v2.2)[68] software, upon which we next performed three rounds of polishing with ONT reads and another round of polishing using Illumina paired-end reads by Nextpolish (v1.2.0)[69] to yield high-quality contigs. Redundant sequences in the polished contigs were removed by purge_dups[70]. Then, breakhic (v1.1) (https://github.com/wtsi-hpag/scaffHiC) was used to identify assembly breakpoints of polished contigs by screening paired Hi-C reads. Finally, 3d-DNA (v180922)[71] pipeline was used to reorder and anchor contigs into scaffolds and chromosomes. To assess genome completeness, we

applied BUSCO (Benchmarking Universal Single-Copy Orthologs) (v3) analysis using the plant early release version (v1.1b1, release May 2015)[45]. In addition, we aligned the Illumina paired-end reads to the final genome assemblies, and detected SNP and Indels by GATK (v4.1.8)[72] to evaluate the base accuracy of the assemblies.

**Genome annotation.** We used Repbase[73] and the species-specific de novo repeat library constructed by RepeatModeler (vopen-1.0.8) to annotate the repeat DNA sequences in three *Papaver* species. RepeatMasker (vopen-4.0.7) was applied to annotated the repeat elements. In addition, we applied LTR_Finder (v1.1)[74], LTRharvest (v1.5.9)[75], and LTR_retriever (v2.8.5)[76] to detect LTR elements. Protein-coding genes were predicted using the MAKER2 pipeline (v2.31.8)[77] integrating evidence-based and ab initio gene predictors. The evidence includes Swiss-Prot (downloaded in Jan. 2020), protein sequences of *A. thaliana*[78], *Beta vulgaris*[79], and *Vitis vinifera*[46], and transcripts assembled by Trinity (v2.1.1)[80]. Three ab initio gene predictors include AUGUSTUS (v3.3)[81], SNAP (v2006-07-28)[82], and Gene-Mark_ES (v3.48)[83]. Predicted genes were evaluated in terms of whether they were supported by transcript or protein homolog, the annotation edit distance (AED), and the exon AED (eAED). We kept the predicted genes with AED < 0.5, eAED < 0.5, and ones supported by transcript or protein homologs. The function domains of the protein-coding genes were annotated by using interProScan with default parameters[84]. Non-coding RNAs (ncRNAs) were annotated using cmscan from INFERNAL (v1.1.2)[85] package based on Rfam database (v14.1)[86].

**Whole-genome duplication identification.** The syntenic analysis was performed by MCScanX[87] with default parameters from top-five BlastP hits. Within each *Papaver* genome, the proportion of genes with WGD/segmental duplication types and the widespread and well-maintained copy numbers of the syntenic blocks indicate no WGD in *P. rhoeas*, single WGD in *P. somniferum*, and two WGDs in *P. setigerum*. In addition, comparison among three *Papaver* genomes, between ancestral eudicot karyotype (AEK)[47] and each *Papaver* genome, as well as between *Vitis vinifera*[46] and each *Papaver* genome provides other evidence for confirming the number of WGDs in three *Papaver* genomes.

**Phylogenomic analysis and divergence time estimation.** We applied Ortho-Finder (v2.3.4)[49] to detect single-copy orthologs from the three *Papaver* genomes and other five angiosperm species including the monocot *Oryza sativa*[88], *Aquilegia coerulea*[89], *Macleaya cordata*[90], *Arabidopsis thaliana*[91], and *Vitis vinifera*[46]. Then, MAFFT (v7)[92] was applied to align the detected single-copy ortholog pairs and the conserved sites were extracted by Gblocks (v0.91b)[93] with default parameters. The maximum likelihood phylogenomic tree was constructed by using RAxML (v8.2.12)[94] with 100 bootstraps. Based on the neutral theory and molecular clock[95], synonymous substitution rate (Ks) measures the divergence time. The divergence times between species were estimated using the Penalized likelihood (PL) method and parameter of "setsmoothing = 1000" with r8s v1.8[96], based on the constructed phylogenetic tree and the fixage times of monocot-dicot split time (152 Mya, http://timetree.org/)[48], the constrain taxon time of *Aquilegia–Papaver* (127.9–139.4 Mya, http://timetree.org/)[48], and the constrain taxon time of *A. thaliana* and *V. vinifera* (107–135 Mya, http://timetree.org/)[48]. We estimated the *P. rhoeas* and *P. somniferum* diverged time at around 7.7 Mya, consisting of timetree website[48] (http://timetree.org/) reports. Similarly, we estimated the divergence time of *P. somniferum* and *P. setigerum* as 4.9 Mya.

**Timing of whole-genome duplications.** To estimate the timing of the WGD event in *P. somniferum* and *P. setigerum*, Ks values of *P. somniferum* syntenic block genes and Ks values of *P. setigerum* syntenic block genes were calculated respectively using YN model in KaKs_Calculator (v2.0)[97]. For *P. setigerum*, we considered the reciprocal best matches among the syntenic gene pairs as the pairs from WGD-2 (the second WGD event) while other syntenic gene pairs were grouped as the pairs from WGD-1 (the first WGD event). The Ks value distributions were then fitted to a mixture model of Gaussian distribution using the Mclust R package[98]. We identified components associated with WGD peaks and calculated their mean and standard deviation of Ks values. To time the WGDs in three *Papaver* species, we estimated the average evolutionary rate for Papaveraceae based on the estimated divergence time of 7.7 Mya and the mean Ks value (0.12) of *P. somniferum–P. rhoeas* syntenic gene pairs. We calculated the synonymous substitutions per site per year (*r*) for Papaveraceae equaling 8.08e−9 (*r* = Ks/2T). The *r* value and the mean WGD Ks value were applied to time the WGDs in *P. somniferum* and *P. setigerum* based on *T* = Ks/2r.

**Ancestral genome reconstruction.** The pre- and post-WGD-1 ancestors were inferred according to a workflow with three stages. In the first stage, we built the syntenic relations among *P. setigerum*, *P. somniferum*, and *P. rhoeas*. The ortholog gene pairs were detected by BlastP and MCscanX[87]. Next, a graph was built based on the ortholog gene pairs, and the graph components were detected as ortholog gene groups (that is putative protogenes, pPGs). We defined a pPG, consisting of four genes from *P. setigerum*, two genes from *P. somniferum*, and one gene from *P. rhoeas*, as a core pPG based on the corresponding numbers of WGD. Finally, DRIMM-Synteny[99] were performed on the core pPG to identify the non-overlapping (NO) synteny blocks in the three *Papaver* species. We filtered blocks

without satisfying 4:2:1 ratio in *P. setigerum*, *P. somniferum*, and *P. rhoeas*. And then, pPGs except for core pPGs were fill in NO synteny blocks. In stage two, the ancestor protochromosomes were reconstructed by solving GMP (genome median problem)[50,100] and GGHP (guided genome halving problem)[101] in the scenario of genomes with multiple WGD events based on the detected NO synteny blocks. We modeled GMP and GGHP in scenario with multiple WGDs as a block matching optimization (MO) problem based on SCoJ (single cut or join) genomic distance[102], and solved this problem by an integer programming method. In the third stage, we inferred the possible order of pPGs based on the previously reconstructed ancestor protochromosomes and NO syntenic blocks. We built a directed weighted pPG graph for each syntenic block. A topological sorting method with a greedy strategy was performed on each graph to get possible gene orders. Compare to MGRA[103], which is the core program in Murat et al.'s research[47] and is can only apply to the block ratio satisfied 1:1:1 and not able to reconstruct ancestral genome for the evolutionary scenarios with WGD[104], our pipeline adopted MO to match block copies in *Papaver* species by minimizing SCoJ distance to reduce the block ratio, enabling transformed problems in *Papaver* species (with two recent WGDs, one shared and one lineage-specific) to traditional GMP and GGHP. Our workflow can also be applied to similar evolutionary scenarios of *Papaver*. We are still exploring the solutions for other complex evolutionary scenarios and eventually will provide a generalized and user-friendly framework for all possible evolutionary scenarios. Our workflow is based on the accuracy of genome assembly. But now even with the cutting-edge sequencing data and widely used assembly methods, assembly errors are inevitable[105]. The potential misassembly may affect the reconstruction and downstream analysis. Therefore, computational evaluation and experimentally validation of genome assemblies are important to obtain more reliable results.

**Gene expression analysis.** The cleaned RNA reads were aligned against the assembled genome using Hisat2 (v2.2.1)[106] and transcripts were discovered and quantified by Stringtie (v2.1.4) and Ballgown[107], respectively, with default parameters. We measured the gene expression level by TPM (transcripts per million). To compare the TPMs between different species, we normalized the TPM by calculating *z*-score in each species. For each species, we first calculated the mean and standard deviation values of all TPMs. Then we calculated $TPM_z$ of each TPM as $TPM_z = (TPM − mean)/sd$, where the mean is the mean value of all TPMs, and sd is the standard deviation of all TPMs. Furthermore, we calculated the normalized TPM as $TPM_n = TPM_z + min\_TPM_z$ to make the normalized TPM nonnegative, where $min\_TPM_z$ is the minimize value of all $TPM_z$.

**Hi-C data analysis.** Juicer (v1.6)[108] and Tadtools (v0.76)[109] were used to calculating Hi-C contact matrix and generating the heatmap. The chromatin loops were detected by HICCUPS[110].

**Reporting summary.** Further information on research design is available in the Nature Research Reporting Summary linked to this article.

## Data availability

Data supporting the findings of this work are available within the paper and its Supplementary Information files. A reporting summary for this Article is available as a Supplementary Information file. The Oxford Nanopore, Illumina paired-end, Hi-C, transcriptome sequencing data generated in this study have been deposited in the NCBI Sequence Read Archive (SRA) database under accession code PRJNA720042 and the National Genomics Data Center under accession code PRJCA004217. This study used previously published RNA-seq data under accession number GSE111119[24]. The genome assembly data are available at the Genome Warehouse in National Genomics Data Center under accession number GWHAZPI00000000, GWHAZPH00000000, and GWHAZPJ00000000 for *P. rhoeas* genome, *P. setigerum* genome and *P. somniferum* genome, respectively. The genome annotations of three *Papaver* genomes are available from GitHub [https://xjtu-omics.github.io/Papaver-Genomics]. The used genomes and annotations of *Oryza sativa*[88] [http://plants.ensembl.org/Oryza_sativa/Info/Index], *Arabidopsis thaliana*[91] [http://plants.ensembl.org/Arabidopsis_thaliana/Info/Index], and *Vitis vinifera*[46] [http://plants.ensembl.org/Vitis_vinifera/Info/Index] are downloaded from EnsemblPlants database. The used genomes and annotations of *Aquilegia coerulea*[89] are downloaded from Phytozome database [https://phytozome-next.jgi.doe.gov/info/Acoerulea_v3_1]. The used genomes and annotations of *Macleaya cordata*[90] are downloaded from the GenBank database under accession number GCA_002174775.1. Source data are provided with this paper.

## Code availability

The ancestral genome reconstruction code has been deposited to https://github.com/XJTU-YeLab/IAG[111] and https://doi.org/10.5281/zenodo.5528515. The analysis scripts have been deposited to https://github.com/XJTU-YeLab/Papaver-Genomics/tree/main/analysis_scripts[112] and https://doi.org/10.5281/zenodo.5528517.

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

## Acknowledgements

We thank Ian A. Graham, Yi Li, Wen Wang, and Zemin Ning for helpful discussions on genome analysis, and Jing Hai, Langchong He, Shengli Han, and Xumei Wang for administrative and technical support. X.Y., K.Y., and L.G. are supported by the National Science Foundation of China (62172325, 32125009, 32070663, 31970317), the Key Construction Program of the National '985' Project, and the Fundamental Research Funds for the Central Universities.

## Author contributions

K.Y. designed and supervised research. L.G. and Y.J. collected materials for sequencing. X.Y., S.G, P.J., and T.X. performed the genome assembly, assembly evaluation, genome annotation, and synteny analysis. K.Y., X.Y., S.G., and L.G. interpreted the WGD events. X.Y. and T.X. performed the gene expression analysis. X.Y. and B.W. performed phylogenomic analysis. S.G., X.Y., J.L., and J.S. performed ancestral genome reconstruction. Y.J. and J.Z. performed the karyotyping of three species. X.Y. and Y.C. performed the Hi-C data analysis. K.Y., X.Y., L.G., and S.G. interpreted the evolution of the BIA gene cluster. X.Y., K.Y., S.G., and Y.C. prepared figures and tables. K.Y., X.Y., L.G., and S.G. wrote the paper. All authors read and proved the final version of this manuscript.

## Competing interests

The authors declare no competing interests.
