## [Peer Review File · Nature Communications]

Three chromosome-scale Papaver genomes reveal punctuated patchwork pathway evolution leading to morphinan and noscapine biosynthesisREVIEWER COMMENTS

Reviewer #1 (Remarks to the Author):

The authors have presented a comparative genomic study of three species of poppy using new genome assemblies to study how genes involved in alkaloid metabolism have evolved over time. This is quite an interesting area of research and it's good to see the new genome assemblies. Figure 4 is the most important contribution of this paper and on this piece of evidence alone, this paper merits eventual publication -- but this figure has some problems and could be substantially clarified (see comment #5). I think greater care should be taken to not overstate the significance of any results and to take a balanced view of what they mean. It is more interesting to just show us the data and allow the data to speak for themselves, rather than trying to build elaborate stories about how something might have happened when the evidence isn't really conclusive supporting a given sequence of rearrangement/duplication/fusion events (e.g. comment #2). I find the authors are often seeming to push a little too hard on the importance or strength of their conclusions, perhaps as they are worried about being perceived as sufficiently novel/impactful for Nature Communications? This is detrimental to the paper. Also, the question of cluster evolution has been considered extensively by other papers and if this is the aim of this paper, more connection should be made to this literature (see comments 7 & 8). A major question in the evolution of gene clusters is how commonly they have evolved from tandem duplication vs. rearrangement. It is critical that this aspect be very rigorously assessed. I think with some substantial re-working this manuscript could be suitable for publication, but it would really need to push less on promoting one particular story and take a much more balanced view of the data. I would appreciate if the authors approached their data with a bit more hesitancy and acknowledgement of the areas of uncertainty -- this is fine! Having two new genomes provides lots of scope for assessing how rearrangements happened and is sufficiently novelty to merit publication in Nature Communications, but only if this is discussed in a more balanced and careful way.

Major comments:

1. I found it very difficult to understand the results of the paper without going to the supplementary materials. I think this paper is much more suitable for a longer and more in-depth format in a specialty journal. Most of the analyses are descriptive with little in the way of hypothesis testing -- the reconstructions of orthology relationships don't conclusively support one model of gene evolution over another, as this would require sequencing of more species.

2. The previous paper by Guo et al (Science 2018) focused extensively on evolution of STORR. It is unclear what is contributed by this further analysis -- can the authors more clearly describe what this shows that was not known previously? I found the paragraphs and figures on STORR particularly confusing and unclear. The model of "Fusion-Translocation" seems like one of many possible sequences of events and this seems more like evolutionary story-telling than hypothesis testing. In fact, given the figures as they are presented, it seems less likely than an alternative hypothesis. As I understand it, here is the presented FT-hypothesis:

- two parent genes leading to STORR were present on same chromosome prior to WGD
- WGD occurred in ancestor of setigerum and somniferum (not rhoeas)
- some "Fusion, translocation (FT)" event happened (which is not described but says it may involve some translocation event)

The authors claim the evidence in Figure 3 and Figure S17 supports this claim. But I fail to see how this is more parsimonious than simply having some deletion between the intergenic regions of the "pre-fusion" modules? This would be SO much clearer if the paralogs to the pre-fusion modules within somniferum were shown on Figure 4. Where are they? I can only assume they must be on some other

chromosome, which would actually be much clearer support for the "translocation" part of the story than the convoluted Figure 3 and accompanying explanation. Alternatively, expanding Figure 3 to show the chromosomal context more clearly might be useful. It's clear from Figure 4 that there are translocations involved because the noscapine cluster got fused with the morphinan cluster. I generally feel like this manuscript spends a lot of words trying to prove hypotheses that are not particularly complicated, and that are not really "provable" anyway with the given data. We won't be able to understand conclusively how rearrangements happened, but showing Figure 4 makes things as clear as they can be. By contrast, Figure 3, which only shows STORR with no surrounding loci, is opaque. I think the manuscript could be considerably condensed with a lot less discussion about STORR -- it was already previously studied quite extensively by Guo et al. 2018. Currently the entire section on "recruitment of new genes to BIA gene cluster locus" is only discussing STORR, which is really less interesting than the other stuff shown on Figure 4. With a good figure, this entire section could be condensed to a single sentence.

3. This paper uses a new and untested method to reconstruct the fissions and fusions in evolution. There is not a lot analysis presented about how the new "MO Solving" framework should work or testing of this on simulated/known data. It is good to see innovation within this paper, so I don't mean to knock this, but I would like to know whether the results are robust to possible errors there. What if you just counted breakpoints? This seems much simpler and less prone to inference errors -- and it is known that such reconstruction is exceedingly error prone/lacks information for strong inference.

4. Line 189: While I agree that transposable elements may have facilitated rearrangements as those involved in STORR evolution, Figure S19 doesn't actually show anything interesting or provocative there so it should not be listed as evidence somehow in support of this idea. Unless it is showing some pattern that is not immediately obvious, in which case this should be explained more clearly.

5. Figure 4: This is an amazing figure! The clear comparison among species is really the novelty that is introduced by this paper. It's very interesting to show that parts of this cluster exists in the other outgroups from *somniferum*. But, I'm concerned the authors are attempting to over-explain the results shown here and that this figure doesn't go far enough. How can you be so confident that there is only one tandem duplication involved here, and the rest are all dispersed duplications? This would seem to require some contrasting of the %identity between all copies present on the same chromosome vs. present on different chromosomes. This could be clarified by extending Figure 4 to show where the paralog copies are residing for the "dispersed duplicates". Given that some of these putative dispersed duplicates have gene names that are pretty close to each other, this needs to be clearly established. For example:

- PSMT3 is "dispersed" but it's only 1 gene away from PSMT1. This could be tandem duplication of PSMT1 + CYP719A21 followed by deletion.

- THS is classified as "dispersed" but it's syntenic with the paralogs of SALSYN and SALR on chr8 in *somniferum*, so where did the duplication happen?

I would like to see some clear phylogenetic study of which genes are the closest relatives of these putative "dispersed" duplications. One way to show this would be to have a heatmap with percentage identity between all genes in the noscapine and morphinan clusters, as well as their putative paralogs -- then you could clearly show which ones were more likely to have arisen from dispersed duplication. This is absolutely critical to the findings of this paper, because tandem duplication is relatively common and dispersed duplication is relatively rare and unlikely to land a gene exactly in a cluster. If 5 genes were dispersed duplicates, this is REALLY important to solidly demonstrate and establish (show that they could not parsimoniously be tandem duplicates).

6. Concerted evolution has a very different meaning than implied on line 214. Please consider some other terminology.

7. The discussion about cluster evolution needs to be improved. The study has not cited another paper published in Nature Communications on this subject that covers similar questions (evolution of clustering) and identified more extensive gene clustering, as well as patterns of differential expression (Li et al. 2020, Nat Comm; <https://doi.org/10.1038/s41467-020-15040-2>) and also focused on testing hypotheses about the evolution of these clusters. Some comparison of results is warranted, at the bare minimum. What did they conclude? How do their conclusions contrast with the conclusions here? The current study has more extensive data with new genome assemblies, so it should be clarified whether the new results contradict or are concordant with previous results. Also, the paper should cite some of the other extensive work on cluster evolution in other species -- for example by Osbourn, Nutzmann, and colleagues (many others as well). The paper should discuss the various hypotheses and evaluate more clearly what kind of data are needed to test them. If this is not done, then I don't see this paper as being suitable for publication -- connections to existing literature are critical.

8. Figure 5A. The text on lines 221-228 seems important but is unclear -- it is stated that the genes "arrived at the recipient locus perhaps pre-equipped with stem-specific promoter or regulatory elements" but this really doesn't make sense -- if that were the case, wouldn't the donor loci also have the same pattern? How would they be pre-equipped if there wasn't a similar pattern in the donor locus? It seems like the "recipient prior" loci also have the same pattern in *somniferum*, so wouldn't it seem more parsimonious that the new loci took on the expression patterns of their local genomic environment? In any case, this is very murky. Li et al. (2020) found that some genes within a cluster had low patterns of co-expression. Do you find evidence that contradicts their claims? Why are the other downstream morphine production genes not clustered?

9. The Hi-C data does not seem conclusive. Hi-C is very noisy and trying to identify loops based on slight enrichment seems vague at best. I can see that it uses the tool HICCUPS but how confident can we be that this is correct? Is there any statement possible about the likelihood of fit to this model over others? Looking visually at the figures, I would have guessed other areas to be enriched and plotted other domains as being TADs/loops, etc. Unless there is some kind of rigorous analysis of this (more replication, ideally), it should be acknowledged as highly speculative. I would like to see how chromatin loops have evolved in these different species but this would require more data. This is another area where I feel the authors are pushing too hard on establishing novelty and not enough on careful interpretation.

11. It would be useful to have a final summary somewhere of the genes involved in the morphinan/noscapine clusters and whether they are syntenic, tandem duplicates, or rearranged.

10. The manuscript needs extensive language editing throughout.

Minor comments:

- Fig S14-S15: I don't see the value in synteny painting. It is somewhat interesting that it can be done, but basically all it shows is that there are many rearrangements. This does not lend itself to quantification or hypothesis testing.

- What tests are being used for the enrichment one line 130? Is this chi-sq test of all simultaneous? Does it correct for chromosome length?

- Fig 1C: What do the colours on the dot plots mean?

- Figure 4b: the two colours of blue for "new gene from dispersed duplication" and "fusion translocation" are easy to confuse. Change the colour scheme.

- Figure 5A. It should be stated in the figure that this is for STORR, as it's only clear in the text.

Reviewer #2 (Remarks to the Author):

In this study, Yang et al. generated chromosome-level genomes of three different poppy species that display extremes in noscapine and morphinan productions to study the evolution of these biosynthetic pathways. Cutting edge technologies were used to sequence and assemble high-quality sequences. These allow for the investigations of different whole-genome duplication events from around 7 million years ago. The authors have used different pieces of evidence to support their claims on the ancestral genomes and their evolution, rearrangements and recruitments of new genes to BIA gene cluster locus. Intriguingly, 15 genes involved in the two distinct pathways were assembled into such a compact gene cluster on one single chromosome 4 in opium poppy. Of particular interest is the investigation of the formation of the fusion STORR enzyme through fusion translocation events, possibly thanks to transposable elements. This article could be of great interest to audiences who are interested in natural product pathway evolution.

Some minor comments:

- This manuscript lacks an abstract and introduction.
- Line 54, page 4: what are the levels of noscapine in *P. setigerum* and morphine levels in *P. rhoeas*? Can the authors provide a figure or a table in the supplementary regarding the noscapine and morphine profiles of these three species?
- Please define patchwork model more clearly.
- It is unclear whether the formation of the fusion STORR enzyme happens prior to or after the formation of the morphinan gene cluster. The author might want to add a statement regarding this event in the main text.
- Fig. 4: the label of new genes from dispersed and new genes from "fusion, translocation" are quite similar. It is suggested that the authors choose different colours.
- It is interesting to see that new genes that arrived at the recipient locus might be pre-equipped with the stem-specific promoter or regulatory elements. With the high sequence quality that the authors have, it would be interesting to know what these promoters and regulatory elements would be. Can the authors provide more information regarding this?
- Can the authors elaborate more on the potential epigenetic regulatory mechanism for simultaneously regulating genes from the two branches? It would significantly increase the novelty of the work and would be of great interest for the audience who read this article to have deeper information regarding these mechanisms.
- What are the levels of identity of the genes from these different species?
- Is this patchwork model common in other biosynthetic pathways? Can the authors comment on this fact on the reported clusters so far?

Reviewer #3 (Remarks to the Author):

Reviewers comment for Nature communication article-

In the present study under review, titled, "Three chromosome-scale *Papaver* genomes reveal punctuated patchwork pathway evolution leading to morphinan and noscapine biosynthesis", authors reported whole-genome assembly for two *Papaver* species, namely *P. setigerum* and *P. rhoeas*, and worked to improve previously published genome for *P. somniferum*. These three species represent extreme in the sense of morphine and noscapine synthesis. The authors then used these three genome assemblies to reconstruct the ancestral genome for *Papaver* species and derived a hypothesis for the evolution of BIAs in plants. The authors proposed the role of whole-genome duplication, followed by chromosomal rearrangement, fusion, and translocation as key events towards the innovation of secondary metabolites. The study is well written, and all figures are of good quality. I also need to mention here that overall, the manuscript is super-condensed and sometimes hard to

follow, especially with the transition from ancestral genome construction towards the proposed hypothesis of the role of fusion and translocation towards the evolution of specialized metabolites. I highly recommend to consider expanding the explanation, simplification of text, and improving the flow of the argument with more detail in order to attract general and specialized readers towards this study. One of the weaknesses of this study is the generality, which is entirely missing. The offered method for ancestral genome construction, the whole hypothesis of evolution of specialized metabolites, and all discussions are *Papaver* specific. I feel that the authors have not attempted to draw comparisons with other studies or to draw a general point-of-view based on their results and other high-quality genomes that have also offered different approaches towards the evolution of specialized metabolites. This is one of the significant weaknesses of this study that I feel must be taken care of.

While going through this study, I find several questions unanswered. I am listing these in the same flow as these are described in the manuscript.

Major comments-

1. I am wondering as why authors choose to use *P. somniferum* genome that they improved in this study while a recent publication has already shown a much-improved genome assembly of *P. somniferum* (<https://www.nature.com/articles/s41467-020-15040-2#Sec25>). In this study, authors showed significant improvement in overall genome assembly compared to what were published in the Science article (Guo et al., 2018). Furthermore, the contig N50, which I personally believe is one of the best criteria to judge a complete and well-ordered genome, is significantly better in Li. et al, 2020 (7.6 Mb compared to 1.74 Mb of this *P. somniferum* genome reported in this study). To me, not including that genome for comparative analysis is not a good idea, and I wonder what authors have the reason to not include it in this study. I am not sure if using this genome changes any of the hypothesis that authors have proposed in this study.
2. Authors described different levels of morphine and noscapine across the three species (Supplementary Figure 2). As the entire manuscript premise is about evolution of the specialized metabolites biosynthesis using these three plants, I feel that looking at overall metabolites and metabolite intermediates to morphine, noscapine across these species will be important. I am particularly curious as how about the intermediates from the earlier steps of biosynthesis pathways of key metabolites and their levels across these three species. Probably such quantification would allow to associate evolution of genes with presence/absence of the metabolites. If these metabolites/intermediates have already been reported and described for these plant species in the past, would be nicer to include as part of discussion and authors views on association of presence/absence of genes with the metabolites. If metabolites are not identified yet, authors should consider identifying and quantifying these intermediates.
3. I am curious about the difference between the different level of contiguity among the genome assemblies reported in this study. Based on my perception, I expected *P. sentigerum* genome to have more difficulties in terms of getting assigned to the scaffolds while *P. rhoeas* to be the easiest based on the genome size, number of chromosomes and so on. But this is just opposite, and 97.6%, 92.4% and 87.9% of contigs for *P. setigerum*, *P. somniferum* and *P. rhoeas*, respectively. I wanted to ask authors their opinion on this. I am assuming that this has to do with raw reads quality (ONT raw read N50), and some other factors. If this could be discussed in the supplementary method section, that would be helpful for people who may want to read this study to get inspiration for a genome assembly project.
4. This is a minor point, but in Table 1, authors reported number of protein coding genes across these three species. The numbers are quite comparable between *P. somniferum* and *P. rhoeas*, while the former have undergone whole genome duplication while the latter has not. While we know that whole genome duplication is further followed by massive genome readjustment, including deletion and rejection of gene families and so on, I am curious to know as what genes specifically were gained and lost across these species. Are they somehow being favored for BIA biosynthesis? What kind of processes were lost in *P. somniferum*? Authors have briefly described it in section 10 of supplementary information, but I feel that a direct comparison between these three species in terms of understanding gene gain and loss would be interesting as they do have a contrasting levels of BIAs despite being closely related. I also feel that including or expanding this aspect in the discussion will make the

discussion section better to understand evolution of BIAs as emerging through this study.

5. Authors mentioned in the method and supplementary method section that they used `purge_dup` to exclude any redundancies from the genome assembly. I think that its important to provide parameters used. As `purge_dup` may also exclude some of the genes that may be real and not artifacts, the used cutoff threshold needs to be mentioned. It may be a good idea if authors could provide all scripts and parameters for different stages of analysis/assembly/ comparative genomics as a github repository as they have provided for reconstruction of ancestral genome.

6. This is just a minor comment. For phylogenetic tree, authors have used eight angiosperms. I feel that including more plant species for the phylogenetic tree by including plants from different lineages and from different time of evolution would be nicer to provide a relative view of the evolution of Papaver species with others. This is simply my preference, and I leave this up to the authors to agree or disagree.

7. I am not sure if I have understood Fig1d. I know that none of the Papaver species have undergone whole genome triplication, hence no peak around 1.5~2 Ks. Normally, we expect a peak that corresponds to a whole genome duplication as probability of sharp increase in Ks value increased with it, hence, I was expecting two peaks for *P. setigerum*, while single peak for *P. somniferum* (this is in-line with what we have observed in case of Arabidopsis, which shows three peaks, one that represents whole genome triplication while rest of the two peaks corresponds to the two whole genome duplication events). If authors do not split *P. setigerum* genome as WGD1 and WGD2 set, how does the peak looks like?

8. Much of this article, including hypothesis and interpretations for comparative genome analysis replies on the constructed ancestral genome of Papaver species. At the beginning of section, "Ancestral genomes...", author says, "We developed a novel bottom-up framework to reconstruct pre- and post-WGD-1 ancestors of Papaver genomes". I have followed work mainly from Jerome Salse, and I wonder how this method is different from what his group have been using and have proposed before? A comparison in terms of what difference between this approach and what previously have been used and providing further discussion on advantages and novelty would be very important.

9. Authors provided github repository for the codes used in this study, and they say in the text as well as in the description of github repository that this code is suitable for reconstruction of ancestral genome of Papaver species. I wonder if the codes could be used for other plant species. Does this code be applicable for including and comparing different plant species and to derive ancestral genomes for a certain lineage? Authors need to clarify this in the text, and if this is specific to Papaver species, why? This is a topic that would be of wider audience interest, and if addressed would be able to make this paper even stronger and would add general appeal.

10. Authors reported six and eleven protochromosomes for the pre- and post-WGD-1 event. A widely believed concept using reconstructed ancestral genomes using species known to have undergone minimal rearrangements proposed seven protochromosomes and described as ancestral eudicot genome having seven chromosomes while ancestral monocot genomes having five chromosomes pre-whole genome duplication (Murat et al., 2017). I am curious as how these reconstructed genomes are different from what authors have reported here. Authors have compared AEK with these plants, but how about the reconstructed genome. A comparison is essential as a lot of hypothesis and interpretations depends on this aspect as well. Also, when authors would predict the origin of ancestral genomes of Papaver species (I mean around what Millions of years ago on the time scale).

11. Authors have described chromosomal fissions and chromosomal fusions associated with the transformation of pre-WGD-1 protochromosomes to modern chromosomes. Given that the genome continuity is vastly different between these species, on what basis we can assign a confidence score for the detection of fusion event sites, which could very well be due to misassembly. Another point, in a recent study on Chromosome assembly of *Ophiorrhiza pumila*, authors experimentally validated genome (assembly gap as 21). In this study, authors identified orientational error across the assembly gap, which were not identified using Bionano and Hi-C datasets. My question is this: How authors could order or talk about the fusion event sites and the shuffling breakpoints when one may very well question the correctness of contigs orientation within assigned scaffolds, and when the percentage of contigs assigned to chromosomes ranges between 87.9% to 97.6%?

12. Authors have very elegantly described the origin of STORR in the section "Recruitment of new

genes to BIA gene cluster locus". However, as a reader, I would be interested if similar events and similar processes have ever been reported in other plant species. In terms of situation in other plants, is there any report which have observed such scenarios, or this is the first result that have observed means of evolution?

13. Role of transposons in the evolution of BIAs in these plant species are not explored. This is another weak point that I wish authors should work. I agree that the synteny analysis and translocation of fusion protein does make sense, but for me, its hard to believe that the transposons have not played any major role when almost 3/4th of the entire genome of all these species are constituted of transposons. A old study published in PNAS on wild tobacco and comparative genome analysis showed role of transposons towards evolution of nicotine, and that study showed that incorporation of specific transposons to the promoter regions promoted or disrupted expression of key genes. I believe that an entire new section on how transposons distribution across promoter and genes of these species, and its interpretation needs to be explored. I disagree that WGD alone could derive such a fascinating metabolic pathway, and including other players are important to make this study complete. Authors have one sentence in page 15, line number 188, and a supplementary Fig. 19, but I find it not described enough and missing interpretations. A more detailed discussion about authors interpretation on role of TEs in these species would be very helpful.

14. I wanted to ask about the Hi-C experiment that authors used to describe epigenetic factors in the last section. I did my best to find out number of replicates that were used for this interpretation but could not. I hope that authors used replicates in the Hi-C experiment, and only then, the interpretations were derived. I will not like to believe this data if replicates are not included for the interpretation.

15. Since expression of genes involved in the BIAs biosynthesis are known to be tissue specific, and this is what authors also reported by saying that the epigenetic factor played a role in getting these genes highly expressed in a tissue specific manner. Do authors have any reference to support their point other than the comparative HiC data for these three species? What I mean is that it would be clearer if authors could show Hi-C data for tissues known to have no expression of BIA biosynthesis associated genes and metabolites, and tissues that show highest accumulation of BIAs and expression of genes. Those comparisons will be better to identify interactions at chromatin level (even if this is done for *P. somniferum*), which then could be explored across these three species to derive the conclusion that authors have made here.

16. I feel that discussion need a comprehensive overview of current study standing with what has been done previously. For example, previously published study on *P. somniferum* (<https://www.nature.com/articles/s41467-020-15040-2#Sec25>) reported copy-number variations for key enzymes towards biosynthesis of BIAs across producing species. Authors should discuss as what they think about those key enzymes described in that study, and if possible, evolutionary scenarios for them. Further, standing of this study with other specialized metabolic pathway is needed to provide a more general point of view on the evolution of secondary metabolism. I could find several interesting aspects being presented in the supplementary information, which should be included in the discussion section.

These are few of the minor comments and are not necessary in the order they appear.

Minor comments-

1. Scale bar is missing in the Supplementary Fig 1.
2. All Hi-C figure panels that authors have shown across all supplementary figures are not clear and difficult to get a sense of quality of the genome. Supplementary Fig. 4b, 6b, and 7b are not clear at all, and ideally should have, axis labels.
3. Please provide the number of assembly gaps in the final genomes of these species.
4. For figure where authors have reported synteny between these species as well as AEK (dot plot), it would be helpful if authors could highlight the synteny. I really liked the way they visualized their results in *P. somniferum* paper (Science, 2018) in Supplementary Fig. S9C and D.
5. Fig2a, when would be the expected time of emergence of pre-WDG-1 ancestor along the time scale.

Reviewer #4 (Remarks to the Author):

The authors of the article "Three chromosome-scale *Papaver* genomes reveal punctuated patchwork pathway evolution leading to morphinan and noscapine biosynthesis" provide the genomes of *Papaver rhoeas* and *Papaver setigerum* and by focusing on the evolutionary fate of genes involved in benzyloisoquinoline alkaloid (BIA) biosynthesis they also provide very interesting insights into how a cascaded pathway evolves. They conclude, that BIA biosynthesis follows the patchwork model: metabolic pathways assemble via the recruitment of primitive, promiscuous enzymes that can react with a wide range of chemically related substrates. To completely follow this conclusion, it is necessary to elaborate in more detail on the phytochemical space of *P. rhoeas*, *P. setigerum* and *P. somniferum*. *P. rhoeas* resembles the ancestral state of the three species most, as it did not experience a whole genome duplication events, which often accelerates evolution. Which type of BIAs are produced in *P. rhoeas*? (see specific comments). Does *P. rhoeas* possess a primitive metabolic pathway that enables the production of noscapine and/or morphinan derivatives? I also suggest to the authors to elaborate more on the biosynthetic steps. Possibly provide a schematic representation of the noscapine and morphinan pathway and as a phytochemist, I miss the chemical structure of at least the major BIAs, e.g. noscapine and morphine. These structures illustrate best the differences/similarities of the major BIAs, and the metabolic steps that "separate" them.

The genome sequencing and data analysis is profound. I only can't completely follow the reconstruction of the ancestral genomes, which might be due to my limited expertise in this field. Still, I would recommend to revise this part, as the authors want to address a broad audience.

Also – I am not a native speaker - but I would recommend language/linguistic editing of the manuscript. I highlighted some of my language/grammar concerns in the specific comments. Summarizing, I believe the data generated in this work is very valuable and interesting to the scientific community, but some points should be revised – see my specific comments.

Specific comments:

Page 3:

Line 22: "While single variant could create a new gene" – this is very unvague, what is meant exactly?

Line 25: "various numbers of whole genome duplications" – this phrase is a bit exaggerated, please be more precise -one WGD shared by *P. somniferum* and *P. setigerum*, and a second only in *P. setigerum*.

Line 27: "nonrandom distribution towards innovation of secondary metabolism" – in *P. somniferum*! See results part, page 10, line 119. In *P. setigerum* – genes in breakpoint vicinity were enriched in plant-pathogen interactions.

Page 4:

Line 40: please elaborate a bit more on the morphinan and noscapine alkaloids – as mentioned above. Biosynthesis, chemical structure, occurrence in the three *Papaver* species.

Line 47: "and compared them with *P. somniferum* genome".

Line 52: Chapter "Genome assembly and annotation" - describing the pattern of morphine and noscapine accumulation in the model species *P. somniferum*, *P. rhoeas* and *P. setigerum* better fits to the introduction (see above). Also, check the sentence and cite references! And the cross reference to Fig. 1a is a bit misleading at this point (line 55), as the Fig. 1. does not illustrate alkaloid accumulation. Which, by the way, could be helpful. For example, the "alkaloid-type" could be included in Fig. 1e.

Page 5:

Line 59: Check the sentence please. "For *P. setigerum* and *P. rhoeas* , the final assemblies with high contiguity were:"?

Page 7:

Starting with line 82 – 94: I can't exactly follow the rationale in this paragraph. How has the divergence time and the WGD time been estimated? The 7.7 Ma divergence time – where does it come

from?

Page 10 – associated to Fig.2: In a previous study (“The opium poppy genome and morphinan production”), the BIA gene cluster was described to be located on Chromosome 11. As far as I understood, the sequence data from this former analysis was re-analyzed in the actual study. Please elaborate why chromosome of BIA gene cluster changed. Have there been large re-assignments?

Page 11:

Line 141: These results suggest a post-WGD diploidization may have driven the innovation of the alkaloid – morphinan - biosynthetic pathways in ancestor of *Papaver somniferum* and *Papaver setigerum*! Please be more specific! Although *P. somniferum* and *P. setigerum* share WGD-1, only *P. somniferum* evolved the most complex alkaloid bouquet. Is the diploidization after WGD-1 – mostly important?

Page 14:

Line 195: check cross reference to Fig. 4a, it seems to me, that Fig.4b is correct.

Elaborate on the “dispersed” duplications. Are these small-scale duplications? Not result of WGD? How did they “arrive” in the gene cluster?

Page 18:

Line 215 to 217: Please check the sentence.

Page 19:

Line 223: Check the sentence – all were duplicated from ancestral copies at remote donor loci of largely low and non-tissue-specific expression compared to the BIA gene cluster that displays a high and coordinated expression in the stem.

Line 225: “recipient prior locus”? What do you mean exactly here? The cluster on chromosome 4 in *P. somniferum*? In Fig. 5a – is the expression of all BIA cluster genes shown, or only of STORR? The donor and recipient loci – do they refer to Fig. 3a?

Line 227: Do you mean that the recipient locus was perhaps pre-equipped with stem-specific promoters? I think this could be written clearer – check the sentence.

Line 233: the WGD created a second copy.

Line 238: coding regions of five genes involved in noscapine biosynthesis (?)

Page 22:

Line 259: *P. rhoeas* produces trace amounts of morphinans and noscapine – in the introduction, only noscapine was mentioned. Please specify! Also, in *P. rhoeas* four genes of the noscapine biosynthesis are present – PSSDR1, CYP82X1, PMT1, and CYP719A21 – all on Chromosome 4. Are they co-expressed? Maybe this is the “core” pathway of noscapine biosynthesis, which was optimized according to the patchwork model?

Line 263. If morphinans and noscapine are both already present in *P. rhoeas*, which most likely resembles the ancestral state, then these compounds were not innovated after the WGD-1, but selective forces could have acted to enhance noscapine and morphinan biosynthesis. Also, only one gene originated from a fusion (STORR), the other were recruited to the locus after gene duplications.

Line 266: STORR channels metabolic flow (not flows).

Page 23:

Line 271: How many of the duplicated genes in the BIA cluster (Fig. 4b) resulted from the WGD-1?

Line 277: “It remains a mystery...” check sentence. Also, nobody “places” genes of the same pathway in separate loci... This is to simplified “slang”. In evolution, selection acts and shapes enzymes, a pathway, the localization of enzymes on chromosomes...

Line 280: Please be more precise. A nonrandom erosion of cis-element of the gene cluster on chromosome 8 in *P. setigerum* was shown, this gene cluster resulted from WGD-2, thus a duplicated cluster was downregulated.

Line 284: What proof is available that noscapine pathway evolved after morphinan pathway? Again- in *P. rhoeas*, noscapine is present (according to the statement in the introduction), I would assume, that noscapine biosynthesis is old in *Papaver*.

Page 24:

Line 290: plants grown in (?) a growth chamber

Line 293: ONT sequencing – introduce abbreviation – move from line 302 to 293.

Page 27:

Line 352: introduce ancestral eudicot karyotype – AEK

Page 28: Divergence time estimates – calibration with fossil records would strengthen the divergence time estimates.

Line 368: Explain your approach more precise. I assume, within species comparisons of syntenic block were performed.

Line 376 & 379: Papaverance !? -

Figures:

Fig.1c: Please include color code/legend that was used to color the syntenic blocks. I assume its Ks?

Fig.1d:

What should the close up illustrate? And its very hard to differentiate between the colored lines representing syntenic orthologs of *P.som.* – *P. set.* and *P. set.* – *P.rho.* in the figure. Please adjust the colors.

Figure2: The graphic figure legend is a bit un-ordered. I miss the explanation of the color code - red indicates enriched fusions, while blue indicates depletion of fusions.

Figure 3:

In general, Fig. 3 is rather complex, which makes it difficult to follow. Maybe it would make the figure more intuitive, if the two colinear regions (the donor loci and the recipient loci) could be highlighted in different colors? The dashed lines _ . . _ . . are confusing, what do they exactly mark? I also don't understand, why the *P. setigerum* chromosomes are not grouped together. In Fig. 3b, the grey boxes in the recipient loci – are they necessary? Graphic legend Fig. 3a: the blue color of STORR is hardly to distinguish from the oxidoreductase pre-fusion module, anyway -would it not be more correct to give the same symbol for STORR in the graphic legend as in the figure itself (lilac rectangle together with blue arrow)?

Fig. 4: The color code is confusing in in panel b – new gene from “fusion, translocation” same color as new gene from dispersed duplication. “dispersed duplication” – are paralogs present somewhere else in the genome?? Please explain what you mean and please elaborate the origin of this genes in more detail in the text!

Extended Data Fig. 3: Please explain the abbreviation GMP, and pPG. In general, the figure is not very intuitive. The figure should be able to stand alone. I would advice to revise it.

Extended Data Fig. 4: Please include color code/legend that was used to color the syntenic block. And could you elaborate it bit on the findings, that we can get from the dotplots in the figure legend? E.g. in 4b – *P. somniferum* plotted against the post WGD-1 ancestor – chromosome 1 shows a syntenic block ration of 1:2, or in other words: a syntenic blocks in chromosome 1 *P. somniferum* “matches” chromosome 1 and chromosome 6 in the post-WGD-1 ancestor. What is the explanation? Small-scale duplication?

With kind regards!

Elisabeth Kaltenegger

Thank you for all your valuable comments. We have provided a response letter addressing all the issues raised by the reviewers. For clarity, all reviewer comments or quoted contents are in italicized fonts. A point-to-point response to each comment is provided in normal fonts. References to revised manuscript contents are also provided where needed. Please notice that the Figure or Supplementary Figure/Table IDs in the response letter are the new IDs in the revised manuscript.

REVIEWER COMMENTS

Reviewer #1 (Remarks to the Author):

The authors have presented a comparative genomic study of three species of poppy using new genome assemblies to study how to genes involved in alkaloid metabolism have evolved over time. This is quite an interesting area of research and it's good to see the new genome assemblies. Figure 4 is the most important contribution of this paper and on this piece of evidence alone, this paper merits eventual publication -- but this figure has some problems and could be substantially clarified (see comment #5). I think greater care should be taken to not overstate the significance of any results and to take a balanced view of what they mean. It is more interesting to just show us the data and allow the data to speak for themselves, rather than trying to build elaborate stories about how something might have happened when the evidence isn't really conclusive supporting a given sequence of rearrangement/duplication/fusion events (e.g. comment #2). I find the authors are often seeming to push a little too hard on the importance or strength of their conclusions, perhaps as they are worried about being perceived as sufficiently novel/impactful for Nature Communications? This is detrimental to the paper. Also, the question of cluster evolution has been considered extensively by other papers and if this is the aim of this paper, more connection should be made to this literature (see comments 7 & 8). A major question in the evolution of gene clusters is how commonly they have evolved from tandem duplication vs. rearrangement. It is critical that this aspect be very rigourously assessed. I think with some substantial re-working this manuscript could be suitable for publication, but it would really need to push less on promoting one particular story and take a much more balanced view of the data. I would appreciate if the authors approached their data with a bit more hesitancy and acknowledgement of the areas of uncertainty -- this is fine! Having two new genomes provides lots of scope for assessing how rearrangements happened and is sufficiently novelty to merit publication in Nature Communications, but only if this is discussed in a more balanced and careful way.

RESPONSE: Thanks for your interests on our work and indeed Figure 4 shows the BIA gene cluster evolutionary history and show the detail evidences of each gene in Supplementary Figures 22-30 in the revision manuscript. In addition, we improved the previous draft genome of *P. somniferum* and *de novo* assembled two additional *Papaver* genomes, *P. setigerum* and *P. rhoeas*. We evaluated the genomes and found they were high quality and in chromosomal-level. We did extensive synteny analysis on these three genomes, and decoded the evolutionary history of these three *Papaver* species. We found two rounds of whole genome duplication (WGD) events with one being shared by *P. somniferum* and *P. setigerum* at around 7.2 Ma and another being *P. setigerum* specific at around 4.0 Ma. Moreover, thanks for the three chromosomal-level genomes, we decoded the evolutionary history of BIA gene cluster and found it can be explained by the

patchwork model.

We summarized the BIA gene cluster evolutionary history in Figure 4 and show the detail evidences of each genes in Supplementary Figures 22-30. Thanks for your comments, and we have revised the Figure as your suggestion. In addition, we constructed the identity heatmap of all genes in noscapine and morphinan clusters, as well as their close paralogs (Figure R4) and added it as a new Supplementary Figure (Supplementary Figure 31). The identity patterns father supports our conclusion of the evolution of genes in BIA genes cluster.

We have revised our manuscript to tone down our statement. Based on our data, we did extensive analysis to reach the conclusions, *e.g.* the evolutionary history of *STORR* and the BIA gene cluster. In our opinion, these results are the reasonable inference and stories based on the data.

Compare to previous paper, *e.g.* Guo, 2018, *Science*; Li *et al.* 2020, *Nat Comm*, we rescaffold *P. somniferum* draft assembly using Hi-C data from the same cultivar (HN1), and *de novo* assembled two additional *Papaver* species. Based on the three high-quality genomes, we investigated the evolutionary history of BIA gene cluster and reveals novel insights into how BIA gene cluster was formed and evolved among different species, which is unexplored by previous paper. We have revised the discussion to link the available researches on gene cluster evolution.

We investigated the origin of genes in BIA gene cluster by integrating evidence of multiple sources including synteny, phylogeny and WGD. We have provided Supplementary Figures 22-30 and Supplementary Tables 13-15 in the revision manuscript to show the original results. We summarized the results in Figure 4. From our results, we found 6 genes in the BIA gene cluster were already presented in the MRCA of three *Papaver* species. During the evolution, nine genes were assembled to this gene cluster. Except *STORR* was resulted from "fusion, translocation" events, we have inferred that *PSCXE1*, *PSAT1*, *PSMT3*, *SALAT*, and *THS* were resulted from "dispersed duplication", and *CYP82X2* was resulted from "tandem duplication". Moreover, two genes, *PSMT2* and *CYP82Y1*, was not obtained a clear origin. We defined the "tandem duplication" as the one genes' origin is adjacency with it, and others defined as the "dispersed duplication". We also tried our best to find the TEs associated with BIA related genes, and marked them on Supplementary Figures (Supplementary Figures 21-30), *e.g.* Supplementary Figure 21 for TEs related with *STORR*, Supplementary Figure 22 for TEs labelling related with *SALAT*, Supplementary Figure 23 for TEs labelling related with *THS*. However, about 3/4th *Papaver* genome are constituted of repetitive elements and half of them are transposons elements (TE), such enrichment of TE makes no clear patterns of TE in the formation of BIA gene cluster.

We have revised our manuscript as suggested, and provided a point-by-point response to your questions.

Major comments:

1. *I found it very difficult to understand the results of the paper without going to the supplementary materials. I think this paper is much more suitable for a longer and more in-depth format in a specialty journal. Most of the analyses are descriptive with little in the*

way of hypothesis testing -- the reconstructions of orthology relationships don't conclusively support one model of gene evolution over another, as this would require sequencing of more species.

RESPONSE: Thanks for your comments. As you suggested, we have revised our manuscript by adding more details. We apologize for any unclarity in our manuscript, which we intended to write in a way that keeps the main manuscript concise and provides abundant supporting evidence in the supplementary materials. In this way, the readers may grasp the main conclusions of the main text, while those who are interested in more technical details are directed to read the supplementary materials.

We agree with you that sequencing more species could bring more intermediate states into the analysis, probably allowing us to capture a finer picture of evolutionary events for each gene in the cluster. However, based on the cutting-edge sequencing data and our extensive analysis of three high-quality *Papaver* genomes, we believe the comparison among them already yields sufficient evidence that the so called "patchwork model" is a decent working model to explain the evolutionary history of benzylisoquinoline alkaloid (BIA) gene cluster. That being said, we expect that the evolutionary history of BIA gene cluster would be updated after more *Papaver* species being sequenced, which will be our future research direction. We have revised the manuscript in a more open-minded manner, and tuned down our conclusion about the BIA gene cluster evolutionary model.

2. *The previous paper by Guo et al (Science 2018) focused extensively on evolution of STORR. It is unclear what is contributed by this further analysis -- can the authors more clearly describe what this shows that was not known previously?*

RESPONSE: In our previous *Science* paper (Guo *et al.* 2018), we, for the first time, reported a chromosome-level genome assembly of the *P. somniferum*, from which a large gene cluster harboring 15 genes involved in the BIA biosynthetic pathway was discovered. In addition, we also showed opium poppy underwent a whole genome duplication at around 7.8 million years ago that could contribute to the clustering of BIA genes. However, due to a lack of genome sequences for additional *Papaver* species, in the 2018 paper we were not able to get a clear picture of the evolutionary steps leading to the assembly of the entire gene cluster formed.

In this current manuscript (Yang *et al.*), we significantly improved this draft genome assembly by incorporating Hi-C data and increased the chromosome anchor rate from 81.6% to 92.4%. This has effectively allowed us to correct assembly errors in the original assembly and enabled the investigation of the complex events involved in evolution of BIA gene cluster. For example, regarding the evolution of *STORR*, we identified the pre-fusion module of *STORR* in 2018 *Science* paper (Figure 2D in Guo *et al.* 2018). Using *P. somniferum* genome only, we inferred the "*STORR* and its closest paralogs show amino acid sequence identity of 75 and 82% for the P450 and oxidoreductase modules, respectively, which suggests that the duplication leading to the *STORR* gene fusion occurred earlier than the WGD event" (Guo *et al.* 2018). Given only one genome at that time, we could only rely on the sequence identity information to draw the conclusion, which was the best explanation at that time. As you correctly pointed out in Comment

#1 that sequencing more genomes will improve the evolutionary model, we assembled another two *Papaver* genomes, and did an extensive syntenic analysis among these three species. We found that the "duplication" in the *Science 2018 STORR* evolutionary model "duplication leading to the *STORR* gene fusion occurred earlier than the WGD event" is actually the WGD shared by *P. setigerum* and *P. somniferum* (Figure 3a, Supplementary Figure 20 in the revision manuscript). We come up with an updated hypothesis of *STORR* evolution, e.g., a "fusion, translocation" event after WGD leading to *STORR* formation at BIA gene cluster (Figure 3, Supplementary Figure 20 in the revision manuscript). We also know now that *P. rhoeas* lacks WGD and doesn't have a *STORR* gene as well as several key BIA genes in this species.

In short, in this manuscript by Yang *et al.*, we corrected the original model published by Guo *et al.* for evolution of BIA genes including *STORR* (Figure 3, Supplementary Figure 20 in the revision manuscript) and other morphinan biosynthesis genes (Figure 4), thanks to the availability of three high quality *Papaver* genomes. Besides, we reconstructed the evolutionary history of three *Papaver* genomes by inferring the ancestral karyotype, showing extensive chromosome fissions and fusions leading to the current karyotypes of three *Papaver* species.

I found the paragraphs and figures on STORR particularly confusing and unclear. The model of "Fusion-Translocation" seems like one of many possible sequences of events and this seems more like evolutionary story-telling than hypothesis testing. In fact, given the figures as they are presented, it seems less likely than an alternative hypothesis. As I understand it, here is the presented FT-hypothesis:

- two parent genes leading to *STORR* were present on same chromosome prior to WGD
- WGD occurred in ancestor of *setigerum* and *somniferum* (not *rhoeas*)
- some "Fusion, translocation (FT)" event happened (which is not described but says it may involve some translocation event)

*The authors claim the evidence in Figure 3 and Figure S17 supports this claim. But I fail to see how this is more parsimonious than simply having some deletion between the intergenic regions of the "pre-fusion" modules? This would be SO much clearer if the paralogs to the pre-fusion modules within *somniferum* were shown on Figure 4. Where are they? I can only assume they must be on some other chromosome, which would actually be much clearer support for the "translocation" part of the story than the convoluted Figure 3 and accompanying explanation. Alternatively, expanding Figure 3 to show the chromosomal context more clearly might be useful. It's clear from Figure 4 that there are translocations involved because the noscapine cluster got fused with the morphinan cluster. I generally feel like this manuscript spends a lot of words trying to prove hypotheses that are not particularly complicated, and that are not really "provable" anyway with the given data. We won't be able to understand conclusively how rearrangements happened, but showing Figure 4 makes things as clear as they can be. By contrast, Figure 3, which only shows *STORR* with no surrounding loci, is opaque. I think the manuscript could be considerably condensed with a lot less discussion about *STORR* -- it was already previously studied quite extensively by Guo *et al.* 2018. Currently the entire section on "recruitment of new genes to BIA gene cluster*

locus" is only discussing STORR, which is really less interesting than the other stuff shown on Figure 4. With a good figure, this entire sentence could be condensed to a single sentence.

RESPONSE: We appreciate your comments and suggestions. The evolutionary model of BIA genes we proposed in this manuscript is derived from a thoughtful explanation of our observation from the multi-genome comparison. Based on thorough analysis of three high-quality genomes, we rejected the evolution model of *STORR* presented in *Guo et al, Science, 2018* and come up a “fusion, translocation” model. Here, we will present a brief explanation of why our proposed model is the more likely hypothesis and why the hypothesis of "simply having some deletions" conflicted with what the data shows.

First, if *STORR* formation is caused by a "simply some deletions between the intergenic regions", the *STORR* flanking genes must have synteny relationships with the flanking genes of pre-fusion modules (Figure R1). However, we did not detect any such syntenic relations between the pre-fusion model loci and *STORR* loci (the donor loci and recipient loci in our manuscript) (Figure 3, Extended Data Figure 1, Supplementary Figures 19, 20 in the revision manuscript), rejecting the "simply some deletions between the intergenic regions " hypothesis.

Figure R1. The genomic synteny relations of *STORR* formation by simple deletion of intergenic region, which is not observed in our data.

Second, since the "deletion" hypothesis does not stand, we proposed the “fusion, translocation” model based on the three chromosomal-level genome comparison, the supporting evidences as following:

- Lack of synteny between the *STORR* loci and pre-fusion module loci in *P. somniferum* and *P. setigerum* (Figure 3, Extended Data Figure 1, Supplementary Figures 19, 20 in the revision manuscript).
- Synteny block detected in donor loci among three species (Figure 3a, Supplementary Figure 20 in the revision manuscript).
- Synteny block detected in recipient loci among three species (Figure 3a, Supplementary Figure 20 in the revision manuscript).

- Absence of pre-fusion module at donor loci of post state in *P. somniferum* and *P. setigerum*, while presence of pre-fusion module at donor loci of prior state in three species (Figure 3a, Supplementary Figure 20 in the revision manuscript).
- Absence of *STORR* at recipient loci of prior state in three species, while presence of *STORR* at recipient loci of post state in *P. somniferum* and *P. setigerum* (Figure 3a, Supplementary Figure 20 in the revision manuscript).

Figure 3a. The simple diagram of synteny relations related with *STORR* loci, and we removed the genome context to make it clearer

Supplementary Figure 20. The whole picture of synteny relations related with *STORR* in three *Papaver* species with genome context.

As for the section "Recruitment of new genes to BIA gene cluster locus", we believe "FT" event leading to *STORR* at BIA gene cluster is one of the most significant findings in this work. To our knowledge, this phenomenon is the rare in gene cluster evolution. Moreover, the translocation of *STORR* from loci of largely low and non-tissue-specific expression to the BIA gene cluster displaying a high and coordinated expression in stem is striking. Thus, we mainly focus on *STORR* in this section.

3. This paper uses a new and untested method to reconstruct the fissions and fusions in evolution. There is not a lot analysis presented about how the new "MO Solving" framework should work

or testing of this on simulated/known data. It is good to see innovation within this paper, so I don't mean to knock this, but I would like to know whether the results are robust to possible errors there.

RESPONSE: Thanks for your comments. In our work, we built a novel bottom-up workflow to reconstruct pre- and post-WGD-1 ancestors based on three high quality *Papaver* genomes. The workflow is based on and improved previous computational theories, proposed by David Sankoff (Sankoff D., *et al*, *COCOON* 1997; Zheng C. *et al*, *Evol Bioinform*, 2006) and Pedro Feijao (Feijão P. *et al*, *TCBB*, 2011), with parsimonious assumption on the ancestor genome reconstruction field. We did not use the available methods, e.g. MGRA (Alekseyev M. *et al*, *Genome Res.* 2009; Avdeyev P. *et al*, *J Comput Biol.* 2016), since they cannot handle the specific evolutionary scenario of *Papaver* species where there is one shared WGD (WGD-1) and a lineage specific WGD (WGD-2). Matching optimization is a strategy to circumvent the shared WGD influence by minimizing SCoJ genomic distance (Feijão P. *et al*, *TCBB*, 2011).

We simulated the evolutionary scenario from top to bottom with two WGDs consistent with three *Papaver* species (Figure R2a) under infinite sites assumption (IS), which means a mutation does not occur at the same locus more than once during evolution and is commonly used in evolutionary studies (Aganezov S. *et al*, *Genome Res.* 2020). We simulated block sequences with some random block adjacencies change between each species. Here, we required that the endpoints involved in changes do not overlap to satisfy IS assumption. And then, we applied our model to reconstruct each middle species, e.g. Species 2, Species 3, and Species 5 in Figure R2a. We repeated 200 times and found that Species 2 (simulated pre-WGD-1 ancestor) and Species 3 (simulated post-WGD-1 ancestors) can be reconstructed with 100% block adjacency accuracy, and Species 5 (simulated pre-WGD-2 ancestor) can be correctly reconstructed with average 99.68% block adjacency accuracy (Figure R2b). This result indicated the accuracy and robustness of our framework under IS assumption.

Next, we evaluated the block adjacency reliability for three ancestors in real *Papaver* evolutionary scenarios. We found all block endpoints in the reconstructed pre-WGD-2 and post-WGD-1 ancestors satisfied IS assumption. We inferred that the block adjacency reliability of both pre-WGD-2 and post-WGD-1 ancestors were 99.68% (pre-WGD-2 ancestor is 99.68% and post-WGD-1 ancestor is 99.68%×100%) based on the simulated results under IS assumption. We adjusted the block adjacency reliability by accumulated multiplication bottom-to-up. However, the pre-WGD-1 ancestor has 11.67% non-IS block endpoint. So, we simulated the pre-WGD-1 ancestor reconstruction under non-IS assumption 1000 times with non-IS block ratio from 0 to 100% (Figure R2c). We used quadratic polynomial to fit the correlation between non-IS endpoint rate and endpoint adjacency inconsistency rate, and obtain the fitting curve with R^2 of about 0.99. Finally, we estimated the reconstructed endpoint adjacency inconsistency rate of pre-WGD-1 ancestor being 5.70%. Therefore, the adjacency reliability for this ancestor is 94.0% (99.68%×100%×(1-5.7%)). So, pre-WGD-1 ancestor may have two endpoint adjacencies inconsistency ((1-94%)*36=2.16).

We have revised Supplementary Note 9 in Supplementary Materials. We added Figure R2 as Supplementary Figure 17 and added Supplementary Table 8 in the revision manuscript.

Figure R2. Evaluation for reconstructed ancestral protochromosomes in simulated *Papaver* scenario. **a.** Simulated *Papaver* evolutionary scenario. The stars are whole genome duplication events (WGDs). The small points indicated the ancestors. The big circles are species in evolution trees. **b.** Reconstructed adjacency consistency with infinite sites assumption for 200 repeat tests. Reconstructed Species 2 and Species 3 (represent pre- and post-WGD-1 ancestors) can be correctly reconstructed in 200 times. Reconstructed Species 5 (pre-WGD-2 ancestor) can be reconstructed with average 99.68% block adjacency consistency compared with simulated result in 200 times. **c.** Quadratic polynomial fitting the relationship between non-IS block endpoints rate and adjacency inconsistency rate for reconstructed Species 2 in non-IS simulation.

What if you just counted breakpoints? This seems much simpler and less prone to inference errors -- and it is known that such reconstruction is exceedingly error prone/lacks information for strong inference.

In some simple cases, counting breakpoint seems to be an easy way to reconstruct ancestor. However, WGDs in *Papaver* species created multi-copies of syntenic blocks. The reconstructed ancestor must include all block endpoints from three *Papaver* species (Figure R3a). In addition, the ancestral genome reconstruction is a global optimization process rather than local optimization, while simple counting breakpoints is a local optimization strategy. Here, we proposed two examples to show the difference between breakpoints counting and our method (Figure R3). The first one shows that if we only count and select the breakpoints with the most occurrences by a greedy strategy, some endpoints will be isolated leading to errors in reconstruction (Figure R3a). Therefore, the reconstruction process should have constraints to ensure every endpoint be considered. Secondly, if we just count breakpoints, the final reconstruction may include errors. For example, there are four blue-red breakpoints and three blue-green breakpoints (Figure R3b). If we just count breakpoints, the final ancestral connection will be blue-red due to the higher count (Figure R3b). However, in our bottom-to-up process, we are able to get the accurate ancestral connection with blue-green and the intermediate stats (e.g. Species 5 and Species 3). If the final state is blue-red, there are at least three shuffling events to reach the final states. However, if the final state is blue-green, only two shuffling events are required to reach the final states. Therefore, we think the ancestral genome as blue-green is the correct one and the breakpoint counting seems not the best method.

In summary, we added the endpoint constraints in integer programming model to avoid isolated endpoints and make all endpoints in final reconstructions. And our ancestor reconstruction followed a bottom-to-up process based on MO framework which can avoid fall into local optimum. Although we are confident about our ancestor reconstruction pipeline and its associated results, if reviewer #1 and the editor insist our ancestral genome reconstruction is not robust, we are willing to remove the content related to it.

Figure R3. Two examples illustrate the drawback of breakpoint counting. **a.** Isolated endpoints. Black points are block endpoints and line are adjacencies **b.** Reconstruction mistake for breakpoint counting. The colored rectangles represent endpoints and arrows with two different colored rectangles represent different adjacencies (break points).

4. *Line 189: While I agree that transposable elements may have facilitated rearrangements as those involved in STORR evolution, Figure S19 doesn't actually show anything interesting or provocative there so it should not be listed as evidence somehow in support of this idea. Unless it is showing some pattern that is not immediately obvious, in which case this should be explained more clearly.*

RESPONSE: Thanks for your comments. We detected the DNA transposable elements around *STORR* and pre-fusion module loci in three *Papaver* species (Figure S19, which is updated as Supplementary Figure 21 in the revision manuscript). *Papaver* species are rich in transposable elements, and we detected 17 types of transposable elements distributed around the donor and recipient loci. We speculated that these TEs may have been involved in the genome rearrangement around these loci. However, we do not have solid evidence about which TE contributed to the

STORR formation. Therefore, to be cautious, we did not make overstatement about the role of TEs in the *STORR* formation.

5. *Figure 4: This is an amazing figure! The clear comparison among species is really the novelty that is introduced by this paper. It's very interesting to show that parts of this cluster exists in the other outgroups from somniferum. But, I'm concerned the authors are attempting to over-explain the results shown here and that this figure doesn't go far enough. How can you be so confident that there is only one tandem duplication involved here, and the rest are all dispersed duplications? This would seem to require some contrasting of the %identity between all copies present on the same chromosome vs. present on different chromosomes. This could be clarified by extending Figure 4 to show where the paralog copies are residing for the "dispersed duplicates". Given that some of these putative dispersed duplicates have gene names that are pretty close to each other, this needs to be clearly established. For example:*

- *PSMT3 is "dispersed" but it's only 1 gene away from PSMT1. This could be tandem duplication of PSMT1 + CYP719A21 followed by deletion.*
- *THS is classified as "dispersed" but it's syntenic with the paralogs of SALSYN and SALR on chr8 in somniferum, so where did the duplication happen?*

I would like to see some clear phylogenetic study of which genes are the closest relatives of these putative "dispersed" duplications. One way to show this would be to have a heatmap with percentage identity between all genes in the noscapine and morphinan clusters, as well as their putative paralogs -- then you could clearly show which ones were more likely to have arisen from dispersed duplication. This is absolutely critical to the findings of this paper, because tandem duplication is relatively common and dispersed duplication is relatively rare and unlikely to land a gene exactly in a cluster. If 5 genes were dispersed duplicates, this is REALLY important to solidly demonstrate and establish (show that they could not parsimoniously be tandem duplicates.

RESPONSE: Thanks for your comments. We investigated the origin of genes in BIA gene cluster by integrating evidence of multiple sources including synteny, phylogeny and WGD. We have provided Supplementary Figures 22-30 and Supplementary Tables 13-15 in the revision manuscript to support our conclusion.

For *CYP82X2*, we found the reciprocal best *BLASTp* hit is *CYP82X1* with sequence identity of 58% (Supplementary Figure 29, Supplementary Table 14 in the revision manuscript). Since the two genes are adjacent to each other in the genome, we reason that *CYP82X1* or *CYP82X2* could arise from a tandem duplication event.

For *PSMT3*, we found the best *BLASTp* hit is *Pso05G43960.0* with sequence identity of 80%, located on chr5 (Supplementary Figure 28, Supplementary Table 14 in the revision manuscript). The sequence identity between *PSMT3* and *PSMT1* is as low as 30.32%. In addition, we detected the synteny relations of *Pso05G43960.0*, and found one syntenic pair in *P. somniferum*, four syntenic pairs in *P. setigerum*, and one syntenic pair in *P. rhoeas*, indicating *Pso05G43960.0* was present in the MRCA of three *Papaver* species. Taken together, it indicates that *PSMT3* was

duplicated from *Pso05G43960.0* by a dispersed duplication event.

THS's non-syntenic best *BLASTp* hit is *Pso04G09740.0* located 34Mb downstream of BIA gene cluster with percentage identity of 67% (Supplementary Figure 23, Supplementary Table 14 in the revision manuscript). In addition, we detected six synteny pairs of *Pso04G09740.0* in three species (one in *P. somniferum*, four in *P. setigerum*, and one in *P. rhoeas*), indicating *Pso04G09740.0* is present in MRCA of three species. These results suggest *THS* was likely duplicated from *Pso04G09740.0* by a dispersed duplication event. And for *SALSYN*, and *SALAT*, we have shown that they were already present in MRCA of three species.

For other gene classified as "dispersed duplication", e.g. *PSATI*, *PSCXE1* and *SALAT*, they have similar scenarios as *THS* and *PSMT3*. The evidence is summarized in Supplementary Figures (*PSATI*: Supplementary Figure 25, *PSCXE1*: Supplementary Figure 26, *SALAT*: Supplementary Figure 22 in the revision manuscript).

As your kindly suggested, we have constructed the identity heatmap of all genes in noscapine and morphinan clusters, as well as their close paralogs (Figure R4), we added this figure as Supplementary Figure 31 in our revised manuscript. This new figure strengths our conclusion.

In summary, six genes in BIA gene cluster including *PSSDR1*, *CYP82X1*, *CYP719A21*, *PSMT1*, *SALAYN*, and *SALR* already existed in the MRCA of three *Papaver* species. Five genes, including *PSCXE1*, *PSATI*, *PSMT3*, *SALAT*, and *THS*, were assembled into BIA gene cluster by "dispersed duplication". *CYP82X2* was tandem duplicated from *CYP82X1*, and *STORR* was created by a "fusion, translocation" event.

Figure R4. The heatmap of identity between genes related with BIA gene cluster.

6. *Concerted evolution has a very different meaning than implied on line 214. Please consider some other terminology.*

RESPONSE: We have revised the title as " Clustered BIA genes are co-regulated and evolved in a concerted manner " in our manuscript at line 243 on Page 16.

7. *The discussion about cluster evolution needs to be improved. The study has not cited another paper published in Nature Communications on this subject that covers similar questions (evolution of clustering) and identified more extensive gene clustering, as well as patterns of differential expression (Li et al. 2020, Nat Comm; <https://doi.org/10.1038/s41467-020-15040-2>) and also focused on testing hypotheses about the evolution of these clusters. Some comparison of results is warranted, at the bare minimum. What did they conclude? How do their conclusions contrast with the conclusions here? The current study has more extensive data with new genome assemblies, so it should be clarified*

whether the new results contradict or are concordant with previous results.

RESPONSE: Thanks for your comments. We totally agree that a comparison between our work and the work by Li *et al.* 2020 *Nature communications* you mentioned should be conducted. In fact, the analysis conducted by Li *et al.* only focused on *P. somniferum* cultivars, while our work comprehensively compared genomes of three *Papaver* species. Therefore, many of our findings in genome evolution of *Papaver* genus in this manuscript (Yang *et al.*) are simply out of reach for Li *et al.* due to the differences of plant materials and computational analysis involved. That said, we would like to point out several common and different findings in the two papers as you rightly suggested.

Here are the main findings or conclusions from Li *et al* paper: 1. A rescaffolded genome assembly of *P. somniferum* based on our draft genome (Guo, *et al. Science*, 2018) was achieved using Hi-C data, improving the proportion of contigs anchored to chromosomes; 2. Gene clustering for BIA biosynthesis genes and co-expressed in stems and root tissues; 3. Co-expression of BIA genes is correlated with alkaloid contents in *P. somniferum* tissues; 4. Copy number variation is found in BIA biosynthesis genes of *P. somniferum* cultivars, well correlated with BIA accumulation in these cultivars. Now we would like to make a pairwise comparison of these conclusions between Li *et al* paper and our current manuscript.

Regarding the first, second and third conclusions, we have published these conclusions in our *Science* 2018 paper, and Li *et al.* improved our draft genome assembly using Hi-C (generated from a completely different cultivar PS7) and echoed our finding of the BIA gene cluster, but how BIA gene cluster was formed and evolved in *Papaver* genus is unexplored, simply due to a lack of genome sequences for additional *Papaver* species. In this manuscript, our Hi-C data were generated from the same cultivar that was used for the initial genome assembly published on *Science* 2018, therefore, we chose not to use the Li *et al.* assembly, and trust our own improved genome assembly of *P. somniferum*, because we believe that it is best to rescaffold the genome using Hi-C data of the same cultivar.

Regarding the fourth conclusions, what Li *et al* did nicely is to include resequencing data for nine *P. somniferum* cultivars and detected copy number variants. They showed that copy numbers of morphinan biosynthetic genes are positively correlated with morphinan accumulation in these cultivars. In our analysis of three *Papaver* genomes, we noticed there are different copy numbers of morphine biosynthesis genes in *P. rhoeas*, *P. somniferum* and *P. setigerum* (Figure R5). The least copy numbers (zero or one) found in *P. rhoeas* account for the low accumulation of morphinans, whereas *P. somniferum* that has more copy numbers than *P. rhoeas* does accumulates morphine abundantly (Figure R5, Supplementary Figure 2). We also found that *P. setigerum* has more copy numbers than *P. somniferum* for many morphine biosynthesis genes (Figure R5). However, the production of morphinans is lower than *P. somniferum* (Figure R5, Supplementary Figure 2), given that the accumulated expression levels of the *P. setigerum* copies are lower than the corresponding genes in *P. somniferum* (Figure R5). Therefore, our results show that the accumulated expression levels, in addition to the copy numbers, are well correlated with morphinan production in different *Papaver* species, updated the conclusions in various *P. somniferum* cultivars by Li *et al.* paper. We added the Figure R5 as Figure 6 in our revised manuscript and added a section "Accumulated gene expression contributes to morphinan

production" in the revised manuscript.

In summary, given the genome data from two extra *Papaver* species, our study reveals novel insights into how BIA gene cluster was formed and evolved among different species, which could not be accessed by using a single species analysis in Li *et al.* paper.

We have made revision to our discussion in the manuscript on Page 22-23.

Figure R5. Copy number and expression of genes on morphine biosynthetic pathway. Correlation of morphinan production with copy number and expression of morphinan biosynthesis genes in three *Papaver* species. Copy numbers of morphinan biosynthesis pathway genes were shown in a line plot on left. Production levels of thebaine, codeine, and morphine were shown in bubble plots. The accumulated gene expressions in stem were shown in a line plot on right. N_exp: normalized expression.

Also, the paper should cite some of the other extensive work on cluster evolution in other species -- for example by Osbourn, Nutzmann, and colleagues (many others as well). The paper should discuss the various hypotheses and evaluate more clearly what kind of data are needed to test them. If this is not done, then I don't see this paper as being suitable for publication -- connections to existing literature are critical.

RESPONSE: As you suggested, we have revised our manuscript, and added discussion with the previous work, including Li *et al* 2020, and other cluster evolution related work. The discussion has been revised as the following on Page 22-23 line 355-364.

8. Figure 5A. The text on lines 221-228 seems important but is unclear -- it is stated that the

*genes "arrived at the recipient locus perhaps pre-equipped with stem-specific promoter or regulatory elements" but this really doesn't make sense -- if that were the case, wouldn't the donor loci also have the same pattern? How would they be pre-equipped if there wasn't a similar pattern in the donor locus? It seems like the "recipient prior" loci also have the same pattern in *somniferum*, so wouldn't it seem more parsimonious that the new loci took on the expression patterns of their local genomic environment?*

RESPONSE: We think this comment is probably due to some misunderstandings of our statement "arrived at the recipient locus perhaps pre-equipped with stem-specific promoter or regulatory elements". The statement means the recipient locus carrying some cis-regulatory elements (CREs) that will enable transcription of genes (donor loci) jumping into or around the recipient locus. Therefore, under this premise, the recipient loci prior and post have a similar high and stem-specific expression (Figure 5a, Extended Data Figure 6). We wouldn't expect the donor loci (prior or post) have a similar expression pattern with the recipient loci (Figure 5a, Extended Data Figure 6).

This statement is supported by what we observed from the data. We found genes at donor loci (either prior or post) showed low and non-tissues-specific expression pattern (Figure 5a and Extended Data Figure 6), suggesting the lack of stem-specific CREs in donor loci. By contrast, we found genes at recipient loci no matter prior or post showed high and stem-specific expression pattern (Figure 5a and Extended Data Figure 6), suggesting stem-specific CREs were equipped at recipient loci no matter prior or post statuses. These results suggested that BIA related genes were assembled from loci without stem-specific CREs (donor loci) to loci pre-equipped stem-specific CREs (recipient loci). And the pre-equipped stem-specific CREs give the BIA related genes a high and stem-specific gene expression in post recipient loci.

In any case, this is very murky. Li et al. (2020) found that some genes within a cluster had low patterns of co-expression. Do you find evidence that contradicts their claims?

RESPONSE: In the paper we published on *Science* (Guo et al. 2018), we showed the 15 morphinan biosynthesis genes within the BIA gene cluster are co-expressed in stem and root. However, BIA gene cluster also contains genes that have no known roles in biosynthetic pathways of BIA. We showed that these non-morphinan genes have low expression patterns (Supplementary Table S15 in Guo et al. 2018 *Science*). Li et al. 2020 reported the same findings in their genome assembly of *P. somniferum*, consistent with our paper (Guo et al. 2018 *Science*). It is unclear how there are two different co-expression patterns for morphinan and non-morphinan related genes within the BIA gene cluster. The gene cluster is probably still evolving.

Why are the other downstream morphine production genes not clustered?

RESPONSE: We have the same question. Unlike the BIA gene cluster, these downstream genes (*CODM*, *COR*, and *T6ODM*) are located on different chromosomes, where they also went through several tandem duplications. In our *Science* paper in 2018, we found these downstream genes have high expression in capsule and stem, whereas the BIA gene cluster has strong co-expression in

stem and root. Considering these observations, and the fact that the two groups of genes are involved in different stages of the biosynthetic pathway, we speculate that opium poppy has evolved a modular structure of the morphinan biosynthetic pathway with different tissue-specificity of local regulation. This has allowed majority of the pathway genes (leading to thebaine) in a gene cluster, and the rest of the pathway genes (converting thebaine to codeine and morphine) in a different genomic location. By separating the two modules genomically, it may help achieving a highly efficient and finely regulated accumulation of morphinans in the designated tissue types. Another possibility is that the opium poppy is still dynamically evolving as Li *et al.* (2020) and others pointed out, so that these downstream genes could be recruited into the BIA gene cluster in the future through either natural or artificial selection.

9. *The Hi-C data does not seem conclusive. Hi-C is very noisy and trying to identify loops based on slight enrichment seems vague at best. I can see that it uses the tool HICCUPS but how confident can we be that this is correct? Is there any statement possible about the likelihood of fit to this model over others? Looking visually at the figures, I would have guessed other areas to be enriched and plotted other domains as being TADs/loops, etc. Unless there is some kind of rigorous analysis of this (more replication, ideally), it should be acknowledged as highly speculative. I would like to see how chromatin loops have evolved in these different species but this would require more data. This is another area where I feel the authors are pushing too hard on establishing novelty and not enough on careful interpretation.*

RESPONSE: Thanks for your comments. We agree that Hi-C data is noisy in general. In our work, we detected 4,948 loops in *P. somniferum* (Supplementary Table 18 in the revision manuscript), and 1,702 loops in *P. setigerum* (Supplementary Table 19 in the revision manuscript) based on all Hi-C data (merging different replicates for each species to enhance data coverage allowed for detecting loop signals) under the FDR threshold as 0.1 (the default parameter). For the loop related with BIA gene cluster in *P. somniferum*, the FDR values are *fdrBL*: 1.08E-4, *fdrDonut*: 1.08E-5, *fdrH*: 1.98E-4, *fdrV*: 6.95E-6 (we labeled the loop as yellow in Supplementary Table 18 in the revision manuscript).

We have two replicates of Hi-C data in both *P. somniferum* and *P. setigerum*. For *P. somniferum*, we did the loop detection on each replicate. We detected the BIA cluster related loop in replicate 1 (coverage is about 149x) with FDR values being *fdrBL*: 0.0159, *fdrDonut*: 1.63E-3, *fdrH*: 7.64E-4, *fdrV*: 7.18E-4, while the BIA cluster related loop in replicate 2 was not significant probably due to lower coverage (coverage is about 133x) with FDR values being *fdrBL*: 0.157, *fdrDonut*: 0.264, *fdrH*: 0.506, *fdrV*: 0.196 (Figure R6, Supplementary Table 18 in the revision manuscript), suggesting that the Hi-C loop between morphine branch genes and noscapine branch genes lacks robustness. Therefore, we removed the Figure 5b into Supplementary Figure 32 in the revision manuscript and tune down our conclusion accordingly.

We also did the Hi-C interaction comparison between morphinan gene copies on chr15 and that on chr8 of each *P. setigerum* replicate (replicate 1 coverage is about 58x, and replicate 2 coverage is about 85x). We found that the Hi-C interactions on chr15 copy was significantly larger than that on chr8 copy on both replicates (Figure R7), indicating the conclusion about different Hi-C contacts between two copies of morphine branch genes in *P. setigerum* is robust. Therefore,

we added Figure R7 as Supplementary Figure 33 in the revision manuscript.

We agree that more data will be helpful to investigate the intricate chromatin interactions within different *Papaver* species. This will be our future work.

Figure R6. Hi-C interaction heatmap of the region including BIA gene cluster in *P. somniferum*. **a.** The heatmap for replicate 1; **b.** the heatmap for replicate 2; **c.** the heatmap for merged data.

Figure R7. Hi-C interactions of morphinan gene copies in *P. setigerum*. **a.** Hi-C interaction heatmap of regions including two copies of morphinan gene cluster on chr15 (left) and chr8 (right) of *P. setigerum* replicate 1. The morphinan gene cluster regions are marked as orange boxes. **b.** The comparison of the interactions from replicate 1 between two morphinan gene cluster copies in *P. setigerum*. The *p*-value is calculated by Wilcoxon rank-sum test. **c.** Hi-C interaction heatmap of regions including two copies of morphinan gene cluster on chr15 (left) and chr8 (right) of *P. setigerum* replicate 2. The morphinan gene cluster regions are marked as orange boxes. **d.** The comparison of the interactions from replicate 2 between two morphinan gene cluster copies in *P. setigerum*. The *p*-value is calculated by Wilcoxon rank-sum test. **e.** Hi-C interaction heatmap of regions including two copies of morphinan gene cluster on chr15 (left) and chr8 (right) of *P. setigerum* merged replicate. The morphinan gene cluster regions are marked as orange boxes. **f.** The comparison of the interactions from merged replicate between two morphinan gene cluster copies in *P. setigerum*. The *p*-value is calculated by Wilcoxon rank-sum test.

10. It would be useful to have a final summary somewhere of the genes involved in the morphinan/noscapine clusters and whether they are syntenic, tandem duplicates, or rearranged.

RESPONSE: Thanks for your comments. Based on the syntenic and *BLASTp* analysis, we found six genes, including *PSSDR1*, *CYP82X1*, *CYP719A21*, *PSMT1*, *SALSYN*, and *SALR*, were already presented in the MRCA of three *Papaver* species. During the evolution, *PSCXE1*, *PSAT1*, *PSMT3*,

SALAT, and *THS* were assembled into the gene cluster by dispersed duplications; *CYP82X2* was assembled into the gene cluster by tandem duplication; and *STORR* was assembled into the gene cluster by "fusion, translocation" event. For each gene, we provided the evidence of synteny, phylogeny and WGD from Supplementary Figures 22 - 30, and Supplementary Tables 13 – 15 in the revision manuscript. The evolutionary model of the BIA gene cluster is summarized in Figure 4. We summarized the evolutionary history of gene in BIA gene cluster in our manuscript from line 221 to line 233 on Page 13-14.

11. The manuscript needs extensive language editing throughout.

RESPONSE: We have carefully revised our manuscript, and corrected some grammar errors.

Minor comments:

- Fig S14-S15: I don't see the value in synteny painting. It is somewhat interesting that it can be done, but basically all it shows is that there are many rearrangements. This does not lend itself to quantification or hypothesis testing.

RESPONSE: We agree with your opinion on these two Supplementary figures. We have removed the synteny painting plot in Fig S14-S15, which are updated as Supplementary Figures 15 and 16 in the revision manuscript.

- What tests are being used for the enrichment one line 130? Is this chisq test of all simultaneous? Does it correct for chromosome length?

RESPONSE: The enrichment test is z-test, which has been corrected by chromosome length. Thanks for pointing this. We have revised this in our manuscript in the legend of Figure 2.

- Fig 1C: What do the colours on the dot plots mean?

RESPONSE: The color in the dotplot of Figure 1c is generated by *MCSanX* automatically. Different colors indicate different synteny blocks. We have revised this in our manuscript in the legend of Figure 1.

- Figure 4b: the two colours of blue for "new gene from dispersed duplication" and "fusion translocation" are easy to confuse. Change the colour scheme.

RESPONSE: We have revised the color scheme of Figure 4b.

- Figure 5A. It should be stated in the figure that this is for STORR, as it's only clear in the text.

RESPONSE: We have revised the Figure legend of Figure 5a as “The heatmap of mean normalized expression level of genes at the donor and the recipient loci related with *STORR* in six tissues of three *Papaver* species”.

Reviewer #2 (Remarks to the Author):

In this study, Yang et al. generated chromosome-level genomes of three different poppy species that display extremes in noscapine and morphinan productions to study the evolution of these biosynthetic pathways. Cutting edge technologies were used to sequence and assemble high-quality sequences. These allow for the investigations of different whole-genome duplication events from around 7 million years ago. The authors have used different pieces of evidence to support their claims on the ancestral genomes and their evolution, rearrangements and recruitments of new genes to BIA gene cluster locus. Intriguingly, 15 genes involved in the two distinct pathways were assembled into such a compact gene cluster on one single chromosome 4 in opium poppy. Of particular interest is the investigation of the formation of the fusion STORR enzyme through fusion translocation events, possibly thanks to transposable elements. This article could be of great interest to audiences who are interested in natural product pathway evolution.

RESPONSE: Thanks for your positive comments. We are also proud of our work on the genomes of three *Papaver* species. We applied the latest sequencing technologies to obtain high-quality chromosomal-level *Papaver* genomes, and found two rounds of whole genome duplications during about 7 million years. We are excited by this work because it is the closest, we have come in the plant biology community to characterizing the evolutionary history of the largest known metabolic gene cluster in plants producing two important medicinal compounds: morphine and noscapine. We believe our work is fundamentally important to understanding trait innovation, pathway evolution and the role of whole genome duplication during speciation and innovation of new traits.

Some minor comments:

- *This manuscript lacks an abstract and introduction.*

RESPONSE: We have added the “Abstract” and “Introduction” in the manuscript to indicate these two parts.

- *Line 54, page 4: what are the levels of noscapine in *P. setigerum* and morphine levels in *P. rhoeas*? Can the authors provide a figure or a table in the supplementary regarding the noscapine and morphine profiles of these three species?*

RESPONSE: We apologized for the missing information in the main text. We did not detect any

noscapine in *P. setigerum*. Both noscapine and morphine level in *P. rhoeas* are very low. We have revised the Supplementary Figure 2 to show the levels of noscapine, morphine, codeine, and thebaine in three *Papaver* species.

- *Please define patchwork model more clearly.*

RESPONSE: Patchwork model was proposed by Ycas in 1974 (Ycas, *J Theor Biol.*, 1974) and Jensen in 1976 (Jensen, *Annu Rev Microbiol.*, 1976), and is a model of pathway evolution, with ancestral enzymes, with broad specificities and catalyzing classes of reaction, forming a large network of possible pathways, and duplication and specialization of these enzymes accounting for extant pathways. We have revised our manuscript to describe the patchwork model at line 386-397 on Page 24.

- *It is unclear whether the formation of the fusion STORR enzyme happens prior to or after the formation of the morphinan gene cluster. The author might want to add a statement regarding this event in the main text.*

RESPONSE: The BIA gene cluster includes the noscapine gene branch and morphinan gene branch. Based on three species comparison, a "fusion, translocation" after WGD-1 but before the divergence of *P. somniferum* and *P. setigerum* (between 7.2Ma and 4.9Ma) led to the formation of *STORR*. And the formation of noscapine branch in *P. somniferum* was after the divergence of *P. somniferum* and *P. setigerum* (4.9 Ma to modern). That means formation of *STORR* happened prior to the formation of noscapine branch in BIA gene cluster.

There are five genes in the morphinan branch of BIA gene cluster, three of which (*SALR*, *SALSYN*, and *THS*) occurred before *STORR* formation in BIA gene cluster, while the order of *STORR* and *SALAT* remains unknown.

- *Fig. 4: the label of new genes from dispersed and new genes from "fusion, translocation" are quite similar. It is suggested that the authors choose different colours.*

RESPONSE: Thanks for pointing this out. We have revised the Figure 4 using different color codes.

- *It is interesting to see that new genes that arrived at the recipient locus might be pre-equipped with the stem-specific promoter or regulatory elements. With the high sequence quality that the authors have, it would be interesting to know what these promoters and regulatory elements would be. Can the authors provide more information regarding this?*

RESPONSE: For the promoter sequences, we have added a Supplementary Data file of the promoter sequence of genes (2kb upstream region) in BIA gene cluster in *P. somniferum*. For the regulatory elements, we have tried to predict the transcription factor (TF) binding motifs in promoter regions of the BIA related genes by *FIMO* software based on JASPAR database (added

as a Supplementary Table 17). However, to validate these motifs and to associate them with potential transcription factors in *P. somniferum*, a non-model plant, is quite challenging and will need more sequencing data such as Chip-seq for different TF, DNase-seq, DNA methylation sequencing data etc. The ChIP-seq is particularly tricky because the antibodies for specific TFs are unavailable for non-model organisms such as *P. somniferum*. Therefore, at this moment we could not make any solid conclusions about the regulatory elements for these BIA genes. We plan to combine various epigenomic sequencing, regulatory network inference and functional genomics to pinpoint the underlying regulatory mechanisms of the BIA related genes in our future work.

- *Can the authors elaborate more on the potential epigenetic regulatory mechanism for simultaneously regulating genes from the two branches? It would significantly increase the novelty of the work and would be of great interest for the audience who read this article to have deeper information regarding these mechanisms.*

RESPONSE: Thanks for your suggestion. We agree that more analysis of potential epigenetic regulatory mechanism of BIA related genes is very interesting. However, as our response to the last comment, it will require more epigenomic sequencing data, regulatory network inference and functional genomic analysis to pinpoint the epigenetic regulatory mechanisms for the BIA genes. For non-model organisms such as *Papaver* species, this is a challenging task and will require careful investigation in the future. In this work, we focus on comparative genome analysis of three *Papaver* species, and the epigenetic and transcriptional regulation of BIA genes will be our future work.

- *What are the levels of identity of the genes from these different species?*

RESPONSE: In Figure 4a and Supplementary Figures 22-30 in the revision manuscript, we labeled the identity as numbers of each gene compare to BIA related genes in *P. somniferum*. We made a table and a heatmap to show the identity distribution (Figure R4, Table R1), and the dominate identity level is larger than 91%.

Table R1. The gene identity of genes related with BIA genes cluster in three *Papaver* species (Figure 4a and Supplementary Figures 22-30)

BIA genes	P. somniferum		P. setigerum		P. rhoeas	
	geneID	Identity (%)	geneID	Identity (%)	geneID	Identity (%)
PSSDR1	Pso04G00230.0	100	Pse16G13020.0	87	Prh03G45950.0	89
PSCXE1	Pso04G00240.0	100	Pse16G13000.0	91		
	Pso04G00200.0	91				
CYP82X1	Pso04G00250.0	100	Pse17G00170.0	58	Prh03G45970.0	70
	Pso04G00260.0	58	Pse15G00490.0	66		
CYP82X2	Pso04G00260.0	100				
	Pso04G00250.0	58				

PSAT1	Pso04G00270.0	100	Pse15G14360.0	70	Prh03G33940.0	68
	Pso04G13170.0	66	Pse08G33790.0	70		
	Pso08G12300.0	69	Pse16G24440.0	69		
Pse17G10180.0			79			
PSMT2	Pso04G00280.0	100	Pse01G40610.0	57	Prh07G33140.0	57
	Pso02G33600.0	58	Pse01G13080.0	57		
	Pso02G60190.0	56	Pse11G05890.0	57		
			Pse07G44350.0	55		
CYP82Y1	Pso04G00290.0	100				
PSMT3	Pso04G00300.0	100	Pse05G48700.0	79	Prh04G36800.0	80
	Pso05G43960.0	80	Pse16G08430.0	80		
	Pso01G11570.0	79	Pse04G11990.0	79		
			Pse20G07920.0	80		
CYP719A21	Pso04G00320.0	99	Pse16G13160.0	75	Prh03G46000.0	93
PSMT1	Pso04G00330.0	99	Pse16G13140.0	73	Prh03G45990.0	85
			Pse17G00230.0	79		
STORR	Pso04G00400.0	100	Pse15G00630.0	100		
			Pse08G45740.0	94		
SALSYN	Pso04G00410.0	100	Pse16G13120.0	66	Prh03G43170.0	69
			Pse17G00200.0	66		
	Pso08G00510.0	97	Pse15G00640.0	99		
			Pse08G45720.0	97		
SALAT	Pso04G00430.0	100	Pse15G00660.0	99		
	Pso04G13170.0	66	Pse08G45690.0	81		
SALR	Pso04G00440.0	98	Pse17G00240.0	71	Prh03G43140.0	76
			Pse16G13210.0	75		
	Pso08G00520.0	95	Pse15G00670.0	99		
			Pse08G45680.0	95		
THS	Pso04G00450.0	99	Pse08G45670.0	95	PrhUNG15120.0	74
	Pso08G00530.0	96	Pse15G00680.0	100		
	Pso04G09740.0	67	Pse15G10610.0	70		
			Pse08G36760.0	57		
	Pso08G09240.0	59	Pse16G21460.0	66		
			Pse17G07600.0	70		

- *Is this patchwork model common in other biosynthetic pathways? Can the authors comment on this fact on the reported clusters so far?*

RESPONSE: Researchers have proposed several models to explain the pathway evolution. For instance, as early as 1945, *Horowitz* proposed the retrograde mode, that means pathways evolve “backwards” from a key metabolite (*Horowitz, PNAS. 1945*). *Granick* proposed the pathway evolution model is that the development of biosynthetic pathways in the forward direction (*GRANICK, Harvey Lect., 1948-1949; GRANICK Ann N Y Acad Sci., 1957*). The third one is the

patchwork model proposed by Ycas in 1974 (Ycas, *J Theor Biol.*, 1974) and Jensen in 1976 (Jensen, *Annu Rev Microbiol.*, 1976).

Patchwork model is one of the most popular pathway evolution models. There are extensive instances supporting patchwork model. For examples, the small-molecule metabolism (SMM) pathway in *E. coli* has been proved as a patchwork model (Rison et al, *Curr Opin Struct Biol.*, 2002), pathway for degradation of pentachlorophenol (PCP), a xenobiotic pesticide, in *Sphingomonas chlorophenolica* (Copley, *Trends Biochem Sci.*, 2000), metabolic pathway for 2CNB degradation in *Pseudomonas stutzeri* (Liu et al, *Appl Environ Microbiol.*, 2011).

The genes involved in a pathway are not always clustered in a gene cluster, especially for Eukaryotes, e.g. plants. Although, many gene clusters have been reported to encode secondary metabolic pathways in plants, including opium poppy, none of them has been linked to the patchwork model. To our knowledge, our work on BIA gene cluster of *Papaver* genus revealed the first gene cluster explained by patchwork model in plants.

We have added the references and the discussions of patchwork related work in our manuscript at line 391-397 on Page 24.

Reviewer #3 (Remarks to the Author):

Reviewers comment for Nature communication article-

In the present study under review, titled, "Three chromosome-scale Papaver genomes reveal punctuated patchwork pathway evolution leading to morphinan and noscapine biosynthesis", authors reported whole-genome assembly for two Papaver species, namely P. setigerum and P. rhoeas, and worked to improve previously published genome for P. somniferum. These three species represent extreme in the sense of morphine and noscapine synthesis. The authors then used these three genome assemblies to reconstruct the ancestral genome for Papaver species and derived a hypothesis for the evolution of BIAs in plants. The authors proposed the role of whole-genome duplication, followed by chromosomal rearrangement, fusion, and translocation as key events towards the innovation of secondary metabolites. The study is well written, and all figures are of good quality. I also need to mention here that overall, the manuscript is super-condensed and sometimes hard to follow, especially with the transition from ancestral genome construction towards the proposed hypothesis of the role of fusion and translocation towards the evolution of specialized metabolites. I highly recommend to consider expanding the explanation, simplification of text, and improving the flow if the argument with more detail in order to attract general and specialized readers towards this study. One of the weaknesses of this study is the generality, which is entirely missing. The offered method for ancestral genome construction, the whole hypothesis of evolution of specialized metabolites, and all discussions are Papaver specific. I feel that the authors have not attempted to draw comparisons with other studies or to draw a general point-of-view based on their results and other high-quality genomes that have also offered different approaches towards the evolution of specialized metabolites. This

is one of the significant weaknesses of this study that I feel must be taken care of. While going through this study, I find several questions unanswered. I am listing these in the same flow as these are described in the manuscript.

RESPONSE: Thanks for your comments. We intended to write in a way that keeps the main manuscript concise and provides abundant supporting evidence in the supplementary materials. In this way, the readers may grasp the main conclusions of the main text, while those who are interested in more technical details are directed to read the supplementary materials. We have revised our manuscript by adding some details.

As for the comment “*the transition from ancestral genome construction towards the proposed hypothesis of the role of fusion and translocation towards the evolution of specialized metabolites*”, we added a few sentences for the transition from ancestral genome reconstruction to the *STORR* formation, and the transition from *STORR* formation to the BIA gene cluster evolution.

As for the comment “*One of the weaknesses of this study is the generality, which is entirely missing*”, we included examples of the patchwork model, the biosynthesis pathway, and the gene clusters in bacterial and plants in the discussion section.

We have revised the manuscript as you suggested, and answered the questions point-to-point.

Major comments-

1. I am wondering as why authors choose to use P. somniferum genome that they improved in this study while a recent publication has already shown a much-improved genome assembly of P. somniferum (<https://www.nature.com/articles/s41467-020-15040-2#Sec25>). In this study, authors showed significant improvement in overall genome assembly compared to what were published in the Science article (Guo et al., 2018). Furthermore, the contig N50, which I personally believe is one of the best criteria to judge a complete and well-ordered genome, is significantly better in Li. et al, 2020 (7.6 Mb compared to 1.74 Mb of this P. somniferum genome reported in this study). To me, not including that genome for comparative analysis is not a good idea, and I wonder what authors have the reason to not include it in this study. I am not sure if using this genome changes any of the hypothesis that authors have proposed in this study.

RESPONSE: Thank you for bringing this up. We did have a couple of considerations for deciding not to use the genome assembly produced by Li *et al.* in this manuscript. Firstly, we published the first draft genome sequence of *P. somniferum* cultivar HN1 in 2018 (Guo *et al.*, *Science*, 2018). After that, Li. *et al* 2020 generated Hi-C data of a different cultivar PS7 and used it to rescaffold our published HN1 contigs to obtain an improved genome assembly (Methods section in Li *et al.* 2020). Given that different cultivars have various amount of indels and structural variations, which would cause artifacts and errors to genome rescaffolding in one cultivar using sequencing data from another cultivar. Therefore, for the best result of genome assembly, in this manuscript we generated Hi-C sequencing data of cultivar HN1 to improve the draft HN1 genome we

published in 2018.

Secondly, we have doubts over the genome assembly statistics such as contig N50 reported in Li *et al.* paper. According to the methods in Li et al. 2020, their strategy for genome assembly is as the following "The sequences of the Guo assembly were broken into segments at gaps of 100 Ns, representing the inter-scaffold gaps of unknown size, and at large gaps of ≥ 1000 Ns. The resulting split genome was comprised of 35,732 resulting segments, which were considered as contigs in the subsequent processing. The lengths of these contigs varied from 132 bp to 38.3 Mb, with the N50 length and the N50 number of the contigs of 7.6 Mb and 104, which provides a draft assembly with sufficient contiguity for making high confidence Hi-C based proximity-guided resc scaffolding". What it says is that Li. *et al* did not assemble genome contigs from scratch, but simply took our published contigs to perform a Hi-C scaffolding. Therefore, the report that they have managed to produce a genome assembly with contig N50 improved from 1.74 Mb (Guo et al., *Science*, 2018) to a reported 7.6 Mb sounds incredible. In fact, we downloaded Li *et al.* genome assembly from NCBI genome repository and the global statistics demonstrates their contig N50 is 1.839 Mb (https://www.ncbi.nlm.nih.gov/assembly/GCA_010119995.1). We also recalculated the contig N50 by *bbmap stats.sh*, which showed that the contig N50 is 1.839 Mb, slightly larger than the 1.74Mb we reported in 2018, but far smaller than 7.6Mb Li *et al.* has reported. Actually, this slightly increased contig N50 from 1.74Mb could be down to the fact that some small contigs were removed (the number of contigs is reduced from 66,578 (Guo, et al. *Science*, 2018) to 9,646 (Li et al. *Nature communications*, 2020)), and which was also reflected from genome size reduction from 2.71Gb to 2.637Gb (Figure R8).

Given these facts and observations, we opted to generate an improved *P. somniferum* HNI genome assembly by ourselves and used it for genome comparisons of three *Papaver* species.

```
(base) [xfyang@mu01 assembly_compare]$ ~/software/miniconda3/envs/bbmap/bin/stats.sh in=GCA_010119995.1_ASM1011999v1_genomic.fna
A      C      G      T      N      IUPAC  Other  GC      GC_stdev
0.3134 0.1866 0.1866 0.3134 0.0005 0.0000 0.0000 0.3732 0.0009

Main genome scaffold total:      11
Main genome contig total:      9646
Main genome scaffold sequence total: 2637.753 MB
Main genome contig sequence total: 2636.327 MB      0.054% gap
Main genome scaffold N/L50:      5/252.943 MB
Main genome contig N/L50:      415/1.839 MB
Main genome scaffold N/L90:      9/190.07 MB
Main genome contig N/L90:      1132/633.21 KB
Max scaffold length:      333.668 MB
Max contig length:      13.77 MB
Number of scaffolds > 50 KB:      11
% main genome in scaffolds > 50 KB: 100.00%

Minimum Scaffold Length      Number of Scaffolds      Number of Contigs      Total Scaffold Length      Total Contig Length      Scaffold Contig Coverage
-----
All      11      9,646      2,637,752,765      2,636,327,527      99.95%
25 KB      11      9,646      2,637,752,765      2,636,327,527      99.95%
50 KB      11      9,646      2,637,752,765      2,636,327,527      99.95%
100 KB      11      9,646      2,637,752,765      2,636,327,527      99.95%
250 KB      11      9,646      2,637,752,765      2,636,327,527      99.95%
500 KB      11      9,646      2,637,752,765      2,636,327,527      99.95%
1 MB      11      9,646      2,637,752,765      2,636,327,527      99.95%
2.5 MB      11      9,646      2,637,752,765      2,636,327,527      99.95%
5 MB      11      9,646      2,637,752,765      2,636,327,527      99.95%
10 MB      11      9,646      2,637,752,765      2,636,327,527      99.95%
25 MB      11      9,646      2,637,752,765      2,636,327,527      99.95%
50 MB      11      9,646      2,637,752,765      2,636,327,527      99.95%
100 MB      11      9,646      2,637,752,765      2,636,327,527      99.95%
250 MB      5      5,166      1,443,459,793      1,442,762,518      99.95%
```

Figure R8. Calculation of Li. *et al.* 2020 genome assembly statistics.

2. *Authors described different levels of morphine and noscapine across the three species (Supplementary Figure 2). As the entire manuscript premise is about evolution of the specialized metabolites biosynthesis using these three plants, I feel that looking at overall metabolites and metabolite intermediates to morphine, noscapine across these species will be important. I am particularly curious as how about the intermediates from the earlier steps of biosynthesis pathways of key metabolites and their levels across these three species. Probably such quantification would allow to associate evolution of genes with presence/absence of the metabolites. If these metabolites/intermediates have already been reported and described for these plant species in the past, would be nicer to include as part of discussion and authors views on association of presence/absence of genes with the metabolites. If metabolites are not identified yet, authors should consider identifying and quantifying these intermediates.*

RESPONSE: We agree that it will be ideal to compare the three species for producing all the various intermediate metabolites. However, this is quite a daunting task and well beyond the objective and scope of this current manuscript. The most challenging part of it is to find trustworthy analytical standards for most of these intermediates. We are not a chemistry research group, and cannot syntheses these intermediate on our own. Due to the regulated substances and the influence of the COVID-19 pandemic, we can only obtain all available standards, *i.e.* Thebaine, Codeine, Morphine, and Noscapine, from domestic market. Therefore, we used HPLC-MS to quantify those four morphinans in the three species.

Regarding the question of “*If these metabolites/intermediates have already been reported and described for these plant species in the past, would be nicer to include as part of discussion and authors views on association of presence/absence of genes with the metabolites. If metabolites are not identified yet, authors should consider identifying and quantifying these intermediates.*”, yes, some metabolic profiling has been reported in literature for several *Papaver* species including *P. rhoeas* and *P. setigerum*. However, we focused on BIA pathways as these are the best understood pathways, for which our genomic analysis can reveal the genetic differences behind these metabolites. For other secondary metabolites, it is challenging to perform similar analysis or discussion as the genetic basis of their biosynthesis is largely unknown.

3. *I am curious about the difference between the different level of contiguity among the genome assemblies reported in this study. Based on my perception, I expected *P. setigerum* genome to have more difficulties in terms of getting assigned to the scaffolds while *P. rhoeas* to be the easiest based on the genome size, number of chromosomes and so on. But this is just opposite, and 97.6%, 92.4% and 87.9% of contigs for *P. setigerum*, *P. somniferum* and *P. rhoeas*, respectively. I wanted to ask authors their opinion on this. I am assuming that this has to do with raw reads quality (ONT raw read N50), and some other factors. If this could be discussed in the supplementary method section, that would be helpful for people who may want to read this study to get inspiration for a genome assembly project.*

RESPONSE: You are right that normally it is more difficult to achieve good contiguity for large and complex genomes than for small and simple genomes. The contiguity of genome assembly is usually affected by multiple factors, such as heterozygosity rates, polyploidy, raw reads quality, repeat content in genome etc. In our study, raw reads quality and genome heterozygosity are two

main reasons on the rates of assignment to the scaffolds. For three *Papaver* species, we have high quality raw reads (Supplementary Table 1), e.g. the ONT raw reads N50 are about 30Kb in both *P. setigerum* and *P. rhoeas*, the Q30 of Hi-C data for *P. setigerum*, *P. rhoeas*, and *P. somniferum* are larger than 91%. The main difference of three *Papaver* species is the heterozygosity rate. *P. setigerum*, despite its large genome size, has a low heterozygosity rate, as shown by a lack of clear heterozygosity peak in the *k*-mer frequency distribution of *P. setigerum* sequencing reads (Supplementary Figure 3). By contrast, despite the relatively smaller genome size, *P. rhoeas* has a high heterozygosity rate of 3.18% as shown by a clear heterozygosity peak in the *k*-mer frequency distribution of *P. rhoeas* sequencing reads (Supplementary Figure 5). Therefore, although the read length, quality of sequencing data and assembly methods for both genomes are similar, the genome assembly contiguity differed a lot. We have revised the supplementary method (Section 6.3 *Assembly evaluation*) by adding the discussion of the reason for different assignment rates in three *Papaver* species.

4. *This is a minor point, but in Table 1, authors reported number of protein coding genes across these three species. The numbers are quite comparable between P. somniferum and P. rhoeas, while the former have undergone whole genome duplication while the latter has not. While we know that whole genome duplication is further followed by massive genome readjustment, including deletion and rejection of gene families and so on, I am curious to know as what genes specifically were gained and lost across these species. Are they somehow being favored for BIA biosynthesis? What kind of processes were lost in P. somniferum? Authors have briefly described it in section 10 of supplementary information, but I feel that a direct comparison between these three species in terms of understanding gene gain and loss would be interesting as they do have a contrasting levels of BIAs despite being closely related. I also feel that including or expanding this aspect in the discussion will make the discussion section better to understand evolution of BIAs as emerging through this study.*

RESPONSE: Thanks for your comments. We performed the syntenic analysis of three *Papaver* species, and found 28,660 genes in *P. somniferum* were syntenic with 19,512 genes in *P. rhoeas* with syntenic depth from one to five (Figure R9a), indicating that 28,660 genes are kept in *P. somniferum* following WGD-1 and diploidization. For any two-species comparison, it is difficult to differentiate gene "gain" and "loss" because gain for one species means loss for the other species, and *vice versa*. Alternatively, we found 21,958 and 26,654 genes are specific to *P. rhoeas* and *P. somniferum* respectively, by comparing *P. rhoeas* and *P. somniferum* genes. We performed the functional enrichment for species-specific genes to understand their functional roles. Based on the functional enrichment analysis, and the *P. somniferum* specific genes were significantly enriched in energy, photosynthesis, and metabolism related pathways, while *P. rhoeas* specific genes were significantly enriched in oxidative phosphorylation, ubiquitin system, ABC transporters related pathways (Figure R9c).

Similarly, we compared *P. somniferum* with *P. setigerum*, and found 41,073 genes in *P. somniferum* were syntenic with 71,398 genes in *P. setigerum* with synteny depth from one to 11 (Figure R9b), indicating that 71,398 genes in *P. setigerum* were related with WGD-2, while 14,241 genes in *P. somniferum* were specific and 35,119 genes in *P. setigerum* were specific based on the

comparison between *P. somniferum* and *P. setigerum*. The functional enrichment showed the *P. somniferum* specific genes were significantly enriched in photosynthesis, ribosome, metabolism related pathways, while *P. setigerum* specific genes were significantly enriched in Spliceosome, metabolism, Endocytosis related pathways (Figure R9d).

We have added a section "8.4 Protein coding gene number comparison based on synteny analysis" the Supplementary material and added Figure R9 as Supplementary Figure 14 in the revision manuscript to explain the difference of protein coding genes in three *Papaver* species.

Figure R9. The synteny depth of *P. rhoeas* versus *P. somniferum* (a) and *P. somniferum* versus *P. setigerum* (b). c. The pathway enrichment of *P. rhoeas* specific genes and *P. somniferum* specific genes based on the comparison between *P. rhoeas* and *P. somniferum*. d. The pathway enrichment of *P. setigerum* specific genes and *P. somniferum* specific genes based on the comparison between *P. setigerum* and *P. somniferum*. We selected the top20 significantly enriched pathways in this figure.

5. Authors mentioned in the method and supplementary method section that they used *purge_dup* to exclude any redundancies from the genome assembly. I think that its important to provide parameters used. As *purge_dup* may also exclude some of the genes that may be real and not artifacts, the used cutoff threshold needs to be mentioned. It may be a good idea if authors could provide all scripts and parameters for different stages of analysis/assembly/comparative genomics as a github repository as they have provided for reconstruction of ancestral genome.

RESPONSE: Thanks for your comments. In our assembly pipeline, we use the default parameters. For *purge_dup* used in *P. rhoeas* genome, the cutoffs parameter is automatically calculated by *calculate* module as "5 34 56 67 112 201". We totally agree your suggestions to provide all scripts and parameters at GitHub. We have uploaded the scripts and parameters used at https://github.com/xjtu-omics/Papaver-Genomics/tree/main/analysis_scripts.

6. *This is just a minor comment. For phylogenetic tree, authors have used eight angiosperms. I feel that including more plant species for the phylogenetic tree by including plants from different lineages and from different time of evolution would be nicer to provide a relative view of the evolution of Papaver species with others. This is simply my preference, and I leave this up to the authors to agree or disagree.*

RESPONSE: Thanks for your kind suggestion. We have carefully selected plant species to compare with *Papaver* species in our phylogenetic analysis. In our opinion, it is critical to include plant species from lineages that are close to *Papaver* spp. in order to get an evolutionary picture of high resolution. However, available genome assemblies of high quality within Papaveraceae are scarce so far, which has limited our options to perform extensive intrafamily phylogeny analysis. Therefore, we included the *Macleaya cordata* (diverged from *Papaver* spp. around 60Ma), the first plant in Papaveraceae that was sequenced and assembled. On the other hand, we chose eight angiosperms from representative families that are commonly used in plant phylogenomic analysis, including monocots (rice), core eudicots (grape, *Arabidopsis* etc.) and an early-diverging dicot (*Aquilegia*). Sampling many species distant from Papaveraceae will probably not reveal any new insights into Papaveraceae evolution because such species have diverged a long time ago from *Papaver* spp. In addition, our selected angiosperms diverged from *Papaver* species at times ranging from 152Ma to 60Ma, covering a wide scale of evolutionary time. Accordingly, we believe our phylogenomic analysis (Figure 1e) provides a fairly good view of the evolutionary relations of *Papaver* species with others.

7. *I am not sure if I have understood Fig1d. I know that none of the Papaver species have undergone whole genome triplication, hence no peak around 1.5~2 Ks. Normally, we expect a peak that corresponds to a whole genome duplication as probability of sharp increase in Ks value increased with it, hence, I was expecting two peaks for P. setigerum, while single peak for P. somniferum (this is in-line with what we have observed in case of Arabidopsis, which shows three peaks, one that represents whole genome triplication while rest of the two peaks corresponds to the two whole genome duplication events). If authors do not split P. setigerum genome as WGD1 and WGD2 set, how does the peak looks like?*

RESPONSE: Thanks for your comments. In Figure 1d, we show the synonymous substitution rate (K_s) density distributions of syntenic paralogs and orthologs. It should be noticed that we split WGD-1 and WGD-2 in *P. setigerum* since the WGD-1 (at 7.2Ma, K_s peak value = 0.115) is very close to WGD-2 (at 4.0Ma, K_s peak value = 0.065). The two close WGD events are reflected by the mixture of two peaks (Figure R10). In our work, we considered that only the reciprocal best

matches among the syntenic gene pairs were considered as the pairs from WGD-2 while other syntenic gene pairs were grouped as the pairs from WGD-1, following the procedure introduced by Liu *et al*, *Cell*, 2020. For Arabidopsis, the WGD-1 (at 67Ma) is far away from WGD-2 (at 50Ma), which was reflected by two distinct peaks.

Figure R10. The K_s distribution of *P. setigerum* without splitting WGD-1 and WGD-2

8. *Much of this article, including hypothesis and interpretations for comparative genome analysis replies on the constructed ancestral genome of Papaver species.*

RESPONSE: Thanks for your comments. We decoded the evolutionary history of BIA gene cluster based on the systematic gene syntenic analysis of three *Papaver* genomes, rather than the reconstructed ancestral genomes. Reconstruction of ancestral genome is the way to decode the karyotyping of *Papaver* species based on the syntenic blocks, while investigating the evolutionary history of BIA gene cluster require the syntenic gene pairs among three *Papaver* species. Therefore, the investigation of the evolutionary history of BIA gene cluster is based on the gene syntenic analysis among three modern *Papaver* species. Sorry for the unclear transition between "Ancestral genomes and accelerated nonrandom post-WGD rearrangements" section to "Recruitment of new genes to BIA gene cluster locus " section. We revised the manuscript at line 173-176 on Page 11 to make the transition clear.

At the beginning of section, "Ancestral genomes...", author says, "We developed a novel bottom-up framework to reconstruct pre- and post-WGD-1 ancestors of Papaver genomes". I have followed work mainly from Jerome Salse, and I wonder how this method is different from what his group have been using and have proposed before? A comparison in terms of what difference between this approach and what previously have been used and providing further discussion on advantages and novelty would be very important.

We have studied Jerome Salse's research (Murat *et al*, *Nat Genet.*, 2017) in details, and found the methods is not able to handle the ancestral genome construction of three *Papaver* species for the following reasons:

- In principle, MGRA, which is the core step in Jerome Salse's research (Murat *et al*, *Nat Genet.*, 2017), is not able to reconstruct ancestral genome for the evolutionary scenarios with WGD (Anselmetti *et al*, *Methods Mol Biol.*, 2018). MGRA was developed by Max A. Alekseyev and Pavel A. Pevzner (Alekseyev *et al*, *Genome Res.*, 2009) to solve genome median problem (GMP) (Sankoff *et al*, *International Computing and Combinatorics Conference*, 1997).
- In Jerome Salse's research, they used three representative species, including grape, cacao and peach, to build ancestral eudicot karyotype (AEK). These three species shared a whole-genome-triplication event (the γ event) at around 100 Ma and never happened another WGD event after that. Since the long time of γ event, the block ratio of three species become 1:1:1 which is suitable for modeling as GMP and solving by MGRA.
- However, the evolutionary scenario of *Papaver* species with a shared WGD by *P. somniferum* and *P. setigerum* at around 7.2Ma and a *P. setigerum* lineage specific WGD at around 4.0Ma. Such a close time scale of these two WGD makes the block ratio as 4:2:1.

To solve this problem, we developed our method for three *Papaver* species with two closed WGD events based on the work of Sankoff (Zheng *et al*, *Evol Bioinform.* 2007) and Pedro Feijão (Feijão *et al*, *TCBB*, 2011). Compared with previous work, we attempted to use block matching strategy to match block copies in related species first by minimizing single cut or join (SCoJ) distance (Feijão *et al*, *TCBB*, 2011) and then relabeled block copies to transform problems in *Papaver* species to traditional GMP and GGHP (Guided Genome Halving Problem). Moreover, we solved the ancestral genomes by integer programming frameworks.

We have revised our method section on reconstruction of ancestral genome in manuscript (Methods Section *Ancestral genome reconstruction*) and Supplementary Materials by adding the comparison as you suggested.

9. *Authors provided github repository for the codes used in this study, and they say in the text as well as in the description of github repository that this code is suitable for reconstruction of ancestral genome of Papaver species. I wonder if the codes could be used for other plant species. Does this code be applicable for including and comparing different plant species and to derive ancestral genomes for a certain lineage? Authors need to clarify this in the text, and if this is specific to Papaver species, why? This is a topic that would be of wider audience interest, and if addressed would be able to make this paper even stronger and would add general appeal.*

RESPONSE: Thanks for your comments. In this research, we designed and implemented the ancestral genome reconstruction according to the evolutionary scenarios with two recent WGDs, one shared and one lineage specific, of three *Papaver* species in this study. Our pipeline can also be applied to similar evolutionary scenarios. Besides the GMP and *Papaver* evolutionary scenarios, we are exploring the solutions for other complex evolutionary scenarios and eventually we will provide a generalized framework for all possible evolutionary scenarios. We have revised the method description in our manuscript and updated the readme at GitHub.

10. Authors reported six and eleven protochromosomes for the pre- and post-WGD-1 event. A widely believed concept using reconstructed ancestral genomes using species known to have undergone minimal rearrangements proposed seven protochromosomes and described as ancestral eudicot genome having seven chromosomes while ancestral monocot genomes having five chromosomes pre- whole genome duplication (Murat *et al.*, 2017). I am curious as how these reconstructed genomes are different from what authors have reported here. Authors have compared AEK with these plants, but how about the reconstructed genome. A comparison is essential as a lot of hypothesis and interpretations depends on this aspect as well. Also, when authors would predict the origin of ancestral genomes of *Papaver* species (I mean around what Millions of years ago on the time scale).

RESPONSE: Thanks for your comments. We strictly followed the definition of Sankoff's guided genome halving problem (GGHP) (Zheng *et al*, *Evol Bioinform*. 2007), which means the ancestral genome is right before the WGD event, and we reconstructed the pre-WGD-1 ancestor at 7.2Ma. We compared pre-WGD-1 ancestor with AEK, and create a dotplot to show the comparison result (Figure R11). We added the Figure R11 as the Supplementary Figure 16g, h in the revision manuscript.

Figure R11. Synteny dotplot between the reconstructed genomes and the ancestral eudicot karyotype (AEK) genome. **a.** pre-WGD-1 ancestor (the *x*-axis) versus AEK (the *y*-axis). **b.** post-WGD-1 ancestor (the *x*-axis) versus AEK (the *y*-axis). Each dot indicates a syntenic gene pair detected by *MCSanX*.

11. Authors have described chromosomal fissions and chromosomal fusions associated with the transformation of pre-WGD-1 protochromosomes to modern chromosomes. Given that the genome continuity is vastly different between these species, on what basis we can assign a confidence score for the detection of fusion event sites, which could very well be due to misassembly. Another point, in a recent study on Chromosome assembly of *Ophiorrhiza pumila*, authors experimentally validated genome (assembly gap as 21). In this study, authors

identified orientational error across the assembly gap, which were not identified using Bionano and Hi-C datasets. My question is this: How authors could order or talk about the fusion event sites and the shuffling breakpoints when one may very well question the correctness of contigs orientation within assigned scaffolds, and when the percentage of contigs assigned to chromosomes ranges between 87.9% to 97.6%?

RESPONSE: We agree that assembly errors will affect the calculation of chromosomal fissions and fusions. In our work, we assembled three *Papaver* species based on the cutting-edge sequencing data and the widely used assembly technologies. We evaluated our assembly from both completeness and base accuracy (Supplementary Section 6.3), and the results indicated the high-quality chromosome level genome assemblies of three *Papaver* species. We are confident of our data, methods, analysis, and the conclusions about the “non random distribution of chromosomal fissions and fusions” and the “superliner correlation between number of WGDs and number of chromosomal rearrangements”. However, we admit that even with the cutting-edge sequencing data and widely used assembly methods, assembly errors are inevitable, especially for genomic regions with large and repetitive segments. Thus, we tune down the conclusion in our result (Section *Ancestral genomes and accelerated nonrandom post-WGD rearrangements*). If the editor and the Reviewer #3 insist that the genome assembly errors may compromise the conclusions about the number of chromosomal fissions and fusions, we are willing to remove the content related to chromosomal fissions and fusions.

12. Authors have very elegantly described the origin of STORR in the section “Recruitment of new genes to BIA gene cluster locus”. However, as a reader, I would be interested if similar events and similar processes have ever been reported in other plant species. In terms of situation in other plants, is there any report which have observed such scenarios, or this is the first result that have observed means of evolution?

RESPONSE: "Fusion" and "translocation" are two main mechanisms to generate new genes (Cardoso-Moreira *et al*, *Methods Mol Biol*, 2012). For example, *Hlip5* has been reported as a result from a novel fusion event between *Coh1* and a *Hlip* domain in *Synechococcus* (Kilian *et al. Mol Plant*, 2008), and *p1-vvD103* is associated with an inter-chromosomal translocation in maize (Wang *et al*, *Genetics*, 2015). However, to our knowledge, *STORR* is the first gene resulted from both fusion and translocation event in plants.

13. Role of transposons in the evolution of BIAs in these plant species are not explored. This is another weak point that I wish authors should work. I agree that the synteny analysis and translocation of fusion protein does make sense, but for me, its hard to believe that the transposons have not played any major role when almost 3/4th of the entire genome of all these species are constituted of transposons. A old study published in PNAS on wild tobacco and comparative genome analysis showed role of transposons towards evolution of nicotine, and that study showed that in-corporation of specific transposons to the promoter regions promoted or disrupted expression of key genes. I believe that an entire new section on how transposons distribution across promoter and genes of these species, and its interpretation

needs to be explored. I disagree that WGD alone could derive such a fascinating metabolic pathway, and including other players are important to make this study complete. Authors have one sentence in page 15, line number 188, and a supplementary Fig. 19, but I find it not described enough and missing interpretations. A more detailed discussion about authors interpretation on role of TEs in these species would be very helpful.

RESPONSE: Thanks for your comments. We totally agree that TEs probably play important roles in the formation of BIA gene cluster. We identified the TEs located around the BIA related genes (Supplementary Figures 21-30 in the revision manuscript). However, as you mentioned that about 3/4th *Papaver* genome are repetitive elements and about half are TEs, making it difficult to associate specific TE with the formation of BIA gene cluster. We have revised our discussion about the role of TEs at line 349-353 on Page 22.

14. I wanted to ask about the Hi-C experiment that authors used to describe epigenetic factors in the last section. I did my best to find out number of replicates that were used for this interpretation but could not. I hope that authors used replicates in the Hi-C experiment, and only then, the interpretations were derived. I will not like to believe this data if replicates are not included for the interpretation.

RESPONSE: Thanks for your comments. We agree that Hi-C data is noisy in general. In our work, we detected 4,948 loops in *P. somniferum* (Supplementary Table 18 in the revision manuscript), and 1,702 loops in *P. setigerum* (Supplementary Table 19 in the revision manuscript) based on all Hi-C data (merging different replicates for each species to enhance data coverage allowed for detecting loop signals) under the FDR threshold as 0.1 (the default parameter). For the loop related with BIA gene cluster in *P. somniferum*, the FDR values are *fdrBL*: 1.08E-4, *fdrDonut*: 1.08E-5, *fdrH*: 1.98E-4, *fdrV*: 6.95E-6 (we labeled the loop as yellow in Supplementary Table 18 in the revision manuscript).

We have two replicates of Hi-C data in both *P. somniferum* and *P. setigerum*. For *P. somniferum*, we did the loop detection on each replicate. We detected the BIA cluster related loop in replicate 1 (coverage is about 149x) with FDR values being *fdrBL*: 0.0159, *fdrDonut*: 1.63E-3, *fdrH*: 7.64E-4, *fdrV*: 7.18E-4, while the BIA cluster related loop in replicate 2 was not significant probably due to lower coverage (coverage is about 133x) with FDR values being *fdrBL*: 0.157, *fdrDonut*: 0.264, *fdrH*: 0.506, *fdrV*: 0.196 (Figure R6, Supplementary Table 18 in the revision manuscript), suggesting that the Hi-C loop between morphine branch genes and noscapine branch genes lacks robustness. Therefore, we removed the Figure 5b into Supplementary Figure 32 in the revision manuscript and tune down our conclusion accordingly.

We also did the Hi-C interaction comparison between morphinan gene copies on chr15 and that on chr8 of each *P. setigerum* replicate (replicate 1 coverage is about 58x, and replicate 2 coverage is about 85x). We found that the Hi-C interactions on chr15 copy was significantly larger than that on chr8 copy on both replicates (Figure R7), indicating the conclusion about different Hi-C contacts between two copies of morphine branch genes in *P. setigerum* is robust. Therefore, we added Figure R7 as Supplementary Figure 33 in the revision manuscript.

15. *Since expression of genes involved in the BIAs biosynthesis are known to be tissue species, and this is what authors also reported by saying that the epigenetic factor played a role in getting these genes highly expressed in a tissue specific manner. Do authors have any reference to support their point other than the comparative HiC data for these three species? What I mean is that it would be clearer if authors could show Hi-C data for tissues known to have no expression of BIA biosynthesis associated genes and metabolites, and tissues that show highest accumulation of BIAs and expression of genes. Those comparisons will be better to identify interactions at chromatin level (even if this is done for *P. somniferum*), which then could be explored across these three species to derive the conclusion that authors have made here.*

RESPONSE: Thanks for your comments. Since *Papaver* species are not model organism, epigenetic datasets, such as DNA methylation, histone modification, and tissue-specific Hi-C data, are still unavailable. In our work, we generated the Hi-C data from leaf successfully. We attempted several times to construct the Hi-C libraries of *P. somniferum* roots and stems. However, libraries construction failed due to the high lignin content in these two tissues. Therefore, we can only analyze leaf Hi-C data in this study, which is rather common in plant Hi-C studies. We are still attempting to construct the root and stem Hi-C libraries. In addition, we are planning to investigate epigenetic factors, such as DNA methylation, histone modifications, to decode the underlying regulatory mechanisms of BIA genes in the future.

16. *I feel that discussion need a comprehensive overview of current study standing with what has been done previously. For example, previously published study on *P. somniferum* (<https://www.nature.com/articles/s41467-020-15040-2#Sec25>) reported copy-number variations for key enzymes towards biosynthesis of BIAs across producing species. Authors should discuss as what they think about those key enzymes described in that study, and if possible, evolutionary scenarios for them. Further, standing of this study with other specialized metabolic pathway is needed to provide a more general point of view on the evolution of secondary metabolism. I could find several interesting aspects being presented in the supplementary information, which should be included in the discussion section.*

RESPONSE: Thanks for your comments. Li *et al.*'s work (Li *et al.* *Nature communications*, 2020) performed intra-species comparison by including resequencing data for nine *P. somniferum* cultivars and indicated the copy number variants of BIA genes are positively correlated with morphinan accumulation in these cultivars. By contrast, we did comparison among three *Papaver* species and revealed that novel insights into how BIA gene cluster was formed and evolved among different species as well as accumulated gene expression besides copy numbers contributed to the morphinan production in *Papaver* species. We have revised our discussion section by adding the comparison between our work and previously published work on *P. somniferum* on Page 22-23.

In addition, we revised the Results section (on Page 14) and Discussion section (on Page 23) to illustrate the evolutionary scenarios of available metabolic pathways, and other genes in the BIA gene cluster on Page 23.

These are few of the minor comments and are not necessary in the order they appear.

Minor comments-

1. *Scale bar is missing in the Supplementary Fig 1.*

RESPONSE: We have revised the Supplementary Figure 1.

2. *All Hi-C figure panels that authors have shown across all supplementary figures are not clear and difficult to get a sense of quality of the genome. Supplementary Fig. 4b, 6b, and 7b are not clear at all, and ideally should have, axis labels.*

RESPONSE: We have revised the Hi-C figures in our supplementary figures.

3. *Please provide the number of assembly gaps in the final genomes of these species.*

RESPONSE: We have added the assembly gaps in table 1.

4. *For figure where authors have reported synteny between these species as well as AEK (dot plot), it would be helpful if authors could highlight the synteny. I really liked the way they visualized their results in *P. somniferum* paper (Science, 2018) in Supplementary Fig. S9C and D.*

RESPONSE: We have revised those figures and added the label to highlight the synteny.

5. *Fig2a, when would be the expected time of emergence of pre-WGD-1 ancestor along the time scale.*

RESPONSE: We have added the time scale of pre-WGD-1 ancestor which is 7.2 Ma just before the WGD-1.

Reviewer #4 (Remarks to the Author):

*The authors of the article “Three chromosome-scale *Papaver* genomes reveal punctuated patchwork pathway evolution leading to morphinan and noscapine biosynthesis” provide the genomes of *Papaver rhoeas* and *Papaver setigerum* and by focusing on the evolutionary fate of genes involved in benzylisoquinoline alkaloid (BIA) biosynthesis they also provide very interesting insights into how a cascaded pathway evolves. They conclude, that BIA biosynthesis follows the patchwork model: metabolic pathways assemble via the recruitment of primitive, promiscuous enzymes that can react with a wide range of chemically related substrates. To completely follow this conclusion, it is necessary to elaborate in more detail on the phytochemical space of *P. rhoeas*,*

P. setigerum and *P. somniferum*.

RESPONSE: Thanks for your interest on our work. Actually, we are not a phytochemical group but a genomics group, focusing on the evolutionary history of BIA gene cluster, encoding the biosynthesis pathway of morphinans, in these three species. We try our best to search the literatures to come up a summary of the phytochemical capabilities of three *Papaver* species, such as *Facchini* group summarized 44 authentic BIAs in *P. somniferum*, including Salutaridine, Papaverine, Narcotoline (Menéndez-Perdomo *et al*, *J Mass Spectrom.* 2021); Grauso *et al* summarized 128 organic compounds isolated from *P. rhoeas*, including Leucine, Rhoeadine (Grauso *et al*, *Phytochem Rev*, 2020). We revised our discussion at line 365-369 on Page 23.

P. rhoeas resembles the ancestral state of the three species most, as it did not experience a whole genome duplication events, which often accelerates evolution. Which type of BIAs are produced in *P. rhoeas*? (see specific comments). Does *P. rhoeas* possess a primitive metabolic pathway that enables the production of noscapine and/or morphinan derivatives? I also suggest to the authors to elaborate more on the biosynthetic steps. Possibly provide a schematic representation of the noscapine and morphinan pathway and as a phytochemist, I miss the chemical structure of at least the major BIAs, e.g. noscapine and morphine. These structures illustrate best the differences/similarities of the major BIAs, and the metabolic steps that “separate” them.

RESPONSE: In *P. rhoeas*, we have only detected trace amount of Noscapine, Thebaine, Codeine and Morphine (Supplementary Figure 2). *P. rhoeas* has a primitive metabolic pathway, leading to the formation of Dopamine and 4-Hydroxyphenylacetaldehyde, the precursor of morphinans (the genes on the pathway are: *PrhUNG23530.0* (TYDC), *Prh05G30560.0* (TryAT)). It also has the NCS (*PrhUNG12800.0*), encoding the enzyme to convert Dopamine and 4-Hydroxyphenylacetaldehyde into (S)-Norcoclaurine, the first step of morphinan biosynthesis. The complete BIA biosynthesis pathway has been published in Hagel *et al. BMC Plant Biology* (2015) 15:227 and we also have a brief version in Guo, *et al. Science* (2018). We revised the introduction to elaborate more and cited early studies on the BIA biosynthesis pathway at line 42-53 on Page 3 but hesitated to extensively describe the previously published results. We also revised Supplementary Figure 2 to add the chemical structure of the major BIAs.

The genome sequencing and data analysis is profound. I only can't completely follow the reconstruction of the ancestral genomes, which might be due to my limited expertise in this field. Still, I would recommend to revise this part, as the authors want to address a broad audience. Also – I am not a native speaker - but I would recommend language/linguistic editing of the manuscript. I highlighted some of my language/grammar concerns in the specific comments. Summarizing, I believe the data generated in this work is very valuable and interesting to the scientific community, but some points should be revised – see my specific comments.

RESPONSE: We have revised the manuscript and figures, including main figures, extended data figures, and supplementary figures, to remove some grammar errors. In addition, we have provided point-to-point responses to your specific comments.

Specific comments:

Page 3:

Line 22: “While single variant could create a new gene” – this is very unvague, what is meant exactly?

RESPONSE: What we meant is that a single genomic variation, *e.g.* duplication, translocation, is able to create a new gene. We have revised this sentence as "While single genomic variation, like duplication and translocation, could create a new gene, it remains enigmatic how a cascaded pathway evolves."

Line 25: “various numbers of whole genome duplications” – this phrase is a bit exaggerated, please be more precise -one WGD shared by *P. somniferum* and *P. setigerum*, and a second only in *P. setigerum*.

RESPONSE: We have revised this sentence as " We observed different numbers of whole genome duplications (WGDs) in three *Papaver* species (zero in *P. rhoeas*, one in *P. somniferum* and two in *P. setigerum*)".

Line 27: “nonrandom distribution towards innovation of secondary metabolism” – in *P. somniferum*! See results part, page 10, line 119. In *P. setigerum* – genes in breakpoint vicinity were enriched in plant-pathogen interactions.

RESPONSE: We have revised this sentence as "nonrandom distribution towards innovation of secondary metabolites and plant-pathogen interactions", and revised the results part on Page 9-10, line 161-163 as "These results suggest a post-WGD diploidization might have driven the innovation of the alkaloid biosynthesis and plant-pathogen interaction related pathways in *Papaver* species."

Page 4:

Line 40: please elaborate a bit more on the morphinan and noscapine alkaloids – as mentioned above. Biosynthesis, chemical structure, occurrence in the three *Papaver* species.

RESPONSE: We have included production levels of BIAs in three *Papaver* species and the chemical structures in Supplementary Figure 2. The complete BIA biosynthesis pathway has been published in Hagel *et al. BMC Plant Biology* (2015) 15:227 and we also have a brief version in Guo, *et al. Science* (2018).

Line 47: “and compared them with *P. somniferum* genome”.

RESPONSE: We have revised this sentence as "To gain insights into the evolutionary history of

the BIA gene cluster among *Papaver* species, we *de novo* assembled two chromosome-level genomes of *P. rhoeas* (common poppy) and *P. setigerum* (Troy poppy) while also improved the previous draft *P. somniferum* HN1 genome."

Line 52: Chapter "Genome assembly and annotation" - describing the pattern of morphine and noscapine accumulation in the model species P. somniferum, P. rhoeas and P. setigerum better fits to the introduction (see above). Also, check the sentence and cite references! And the cross reference to Fig. 1a is a bit misleading at this point (line 55), as the Fig. 1. does not illustrate alkaloid accumulation. Which, by the way, could be helpful. For example, the "alkaloid-type" could be included in Fig. 1e.

RESPONSE: Although it is known that the production levels of BIAs in three *Papaver* species are different, the precise amount was determined in this study (Supplementary Figure 2). Thus, we decided to describe the patterns of morphinan and noscapine accumulation in three *Papaver* species in Results rather than Introduction.

We have revised the morphinan and noscapine accumulation description in section "Genome assembly and annotation" as "Among three *Papaver* species, *P. somniferum* HN1 cultivar (2n=22) accumulates highest amount of morphine and noscapine (Supplementary Figs. 1, 2), while no noscapine but intermediate levels of morphine for *P. setigerum* DCW1 cultivar (2n=44) (Fig. 1a, Supplementary Figs. 1, 2) whereas trace amount of morphinan and noscapine for *P. rhoeas* YMR1 cultivar (2n=14) (Supplementary Figs. 1, 2)". We have corrected the Figure references as you suggested.

Fig. 1e is the phylogenetic tree analysis of three *Papaver* species with other five species. There are at least 40 alkaloids (Grauso et al, *Phytochem Rev*, 2020, Menéndez-Perdomo *et al*, *J Mass Spectrom*. 2021) and their accumulations are largely unknown, especially for the five species for the comparison. We put the production levels and the chemical structures of morphinans and noscapine in Supplementary Figure 2.

Page 5:

Line 59: Check the sentence please. "For P. setigerum and P. rhoeas , the final assemblies with high contiguity were:"?

RESPONSE: We have double checked this sentence and revised it as "The genome assemblies of *P. setigerum* and *P. rhoeas* were highly contiguous with 97.6% and 87.9% of genome contigs anchored to chromosomes by using Hi-C scaffolding, respectively (genome assembly size of 4.6 and 2.5 Gb, scaffold N50 values of 211.2 and 329.4 Mb, contig N50 values of 65.6 and 5.3 Mb)".

Page 7:

Starting with line 82 – 94: I can't exactly follow the rationale in this paragraph. How has the divergence time and the WGD time been estimated? The 7.7 Ma divergence time – where does it come from?

RESPONSE: Based on the neutral theory and molecular clock (Kimura *et al.*, *J Mol Evol.* 1987), synonymous substitution rate (K_s) measures the divergence time. We estimated the divergence time based on phylogenetic analysis (Supplementary Section 8.2) as "The divergence times between species were estimated using the Penalized likelihood method and parameter of "setsmoothing = 1000" with *r8s v.1.843*, based on the constructed phylogenetic tree and the fixage times of monocot-dicot split time (152 Ma, <http://timetree.org/>), constrain taxon time of *Aquilegia-Papaver* (127.9~139.4 Ma, <http://timetree.org/>), and constrain taxon time of *A. thaliana* and *V. vinifera* (107~135 Ma, <http://timetree.org/>). We estimated the *P. rhoeas* and *P. somniferum* diverged time at around 7.7 Ma, consisting with timetree website (<http://timetree.org/>) reports. Similarly, we estimated the divergence time of *P. somniferum* and *P. setigerum* is 4.9 Ma".

Based on the divergence time and K_s distributions, we estimated the WGD time (Supplementary Section 8.3). The method is like "Given the mean K_s value (0.12) of *P. somniferum-P. rhoeas* and their divergence date T (7.7 Ma), we calculated the synonymous substitutions per site per year (r) for Papaveraceae as $8.08e-9$ ($T = K_s / 2r$) (Supplementary Table 6) which was applied to time the WGDs of *P. somniferum* and *P. setigerum*. We dated the WGD in *P. somniferum* ($K_s = 0.116 \pm 0.028$) around 7.2 ± 1.7 Ma, the first WGD in *P. setigerum* ($K_s = 0.115 \pm 0.018$) around 7.1 ± 1.1 Ma, and the second WGD in *P. setigerum* ($K_s = 0.065 \pm 0.017$) around 4.0 ± 1.0 Ma (Fig. 1d, Supplementary Table 6)."

Page 10 – associated to Fig.2: In a pervious study ("The opium poppy genome and morphinan production"), the BIA gene cluster was described to be located on Chromosome 11. As far as I understood, the sequence data from this former analysis was re-analyzed in the actual study. Please elaborate why chromosome of BIA gene cluster changed. Have there been large re-assignments?

RESPONSE: Thanks for your comments. After analysis of the Hi-C data of *P. somniferum*, we found our previous published genome assembly (Guo, *et al*, *Science*, 2018) has several mis-scaffolding (Figure R12a). We used Hi-C data and 3d-DNA software to rescaffold our published contigs to obtain current assembly. We compared the rescaffolded genome assembly to the published genome assembly, and reassigned chromosome IDs based on the genome dotplot (Figure R12b). After reassignment of chromosome IDs, we found the BIA gene cluster were located at chr4 rather than chr11. Based on the genome wide Hi-C contact maps, we can see that the rescaffolded genome assembly has indeed been improved by correcting the mis-assembled contigs in the previous published genome.

Fig. R12. The explanation of chromosome ID changes. **a.** The Hi-C contact map based on previous published genome of *P. somniferum*. **b.** The dotplot of comparison between improved genome and previous published genomes. The green circle labeled the position of BIA gene cluster.

Page 11:

Line 141: *These results suggest a post-WGD diploidization may have driven the innovation of the alkaloid – morphinan - biosynthetic pathways in ancestor of Papaver somniferum and Papaver setigerum! Please be more specific! Although P. somniferum and P. setigerum share WGD-1, only P. somniferum evolved the most complex alkaloid bouquet. Is the diploidization after WGD-1 – mostly important?*

RESPONSE: Thanks for pointing this out. The diploidizations after both WGD-1 and WGD-2 are important for novel traits of *Papaver* species. We have revised the sentence as "These results suggest a post-WGD diploidization might have driven the innovation of the alkaloid biosynthesis and plant-pathogen interaction related pathways in *Papaver* species." at line 162-164 on Page 10.

Page 14:

Line 195: *check cross reference to Fig. 4a, it seems to me, that Fig.4b is correct.*

Elaborate on the “dispersed” duplications. Are these small-scale duplications? Not result of WGD? How did they “arrive” in the gene cluster?

RESPONSE: We have revised that reference of Fig. 4b. Dispersed duplication means the duplicated genes and their original copies are not adjacent. Yes, the dispersed duplications are small-scale duplications compared to WGD and not result of WGD. We tried to investigate the mechanisms of "how did they 'arrive' in the gene cluster", and labeled the annotated transposon elements (TEs) in related with these genes in Supplementary Figures 22-30. The results indicated the TEs likely mediated the duplications.

Page 18:

Line 215 to 217: Please check the sentence.

RESPONSE: We have revised this sentence as “In principle, the genetic components of BIA pathway should allow the biosynthesis of morphinan and noscapine albeit at their original loci before clustering, raising questions on the necessity of gene clustering in evolution.”.

Page 19:

Line 223: Check the sentence – all were duplicated from ancestral copies at remote donor loci of largely low and non-tissue-specific expression compared to the BIA gene cluster that displays a high and coordinated expression in the stem.

RESPONSE: We have revised this sentence as you suggested.

Line 225: “recipient prior locus”? What do you mean exactly here? The cluster on chromosome 4 in *P. somniferum*? In Fig. 5a – is the expression of all BIA cluster genes shown, or only of STORR? The donor and recipient loci – do they refer to Fig. 3a?

RESPONSE: We defined the donor and recipient loci in section "Recruitment of new genes to BIA gene cluster locus". Original, there was one copy of donor and recipient loci. And then, WGD duplicated them. One copy of pre-fusion module jumped from donor loci to one of the recipient loci by a “cut and paste” manner. Thus, two copies created by WGD recorded the states before and after the jumping. We defined the “recipient post locus” as the copy of recipient loci with jumping fragment inserted, while the other untouched recipient locus is “recipient prior locus” (Figure 3a, Supplementary Figure 20 in the revision manuscript). Figure 5 shows the summarized gene expression of donor and recipient loci, referred to Figure 3a. Extended Data Fig. 6 shows the detail gene expressions of all genes related with BIA gene cluster in three species. We have revised the Figure legend of Figure 5.

Line 227: Do you mean that the recipient locus was perhaps pre-equipped with stem-specific promoters? I think this could be written clearer – check the sentence.

RESPONSE: Yes, this is exactly what we meant.

Line 233: the WGD created a second copy.

RESPONSE: Thanks, we have revised that as you suggested.

Line 238: coding regions of five genes involved in noscapine biosynthesis (?)

RESPONSE: We have revised the sentence as "although coding sequence regions of five genes involved in morphinan biosynthesis remain intact in both copies in *P. setigerum*".

Page 22:

Line 259: *P. rhoeas* produces trace amounts of morphinans and noscapine – in the introduction, only noscapine was mentioned. Please specify! Also, in *P. rhoeas* four genes of the noscapine biosynthesis are present – *PSSDR1*, *CYP82X1*, *PMT1*, and *CYP719A21* – all on Chromosome 4. Are they co-expressed? Maybe this is the “core” pathway of noscapine biosynthesis, which was optimized according to the patchwork model?

RESPONSE: We have revised the section “Genome assembly and annotation” of the production as “whereas trace amount of morphinan and noscapine for *P. rhoeas* YMR1 cultivar”. All these four genes on chr3 of *P. rhoeas*. We showed their expression patterns in Extended Data Fig. 6 and found they were lowly expressed and not tissue-specific. We agree with you that *PSSDR1*, *CYP82X1*, *PMT1*, and *CYP719A21* were already present in the MRCA of three species as the core genes of noscapine biosynthesis and the cluster was optimized according to the patchwork model in *P. somniferum* during evolution.

Line 263. If morphinans and noscapine are both already present in *P. rhoeas*, which most likely resembles the ancestral state, then these compounds were not innovated after the WGD-1, but selective forces could have acted to enhance noscapine and morphinan biosynthesis. Also, only one gene originated from a fusion (*STORR*), the other were recruited to the locus after gene duplications.

RESPONSE: This is also our interpretation.

Line 266: *STORR* channels metabolic flow (not flows).

RESPONSE: Thanks. We have revised that as you suggested.

Page 23:

Line 271: How many of the duplicated genes in the BIA cluster (Fig. 4b) resulted from the WGD-1?

RESPONSE: Based on our analysis, seven genes presented at BIA cluster before WGD-1 and eight genes assembled into the BIA cluster after WGD-1 (Figure 4b), all of which are resulted from small-scale genomic rearrangement event, e.g. duplication, translocation, as consequences of WGD-1 rather than directly from WGD-1.

Line 277: “It remains a mystery...” check sentence. Also, nobody “places” genes of the same pathway in separate loci... This is to simplified “slang”. In evolution, selection acts and shapes enzymes, a pathway, the localization of enzymes on chromosomes...

RESPONSE: We have revised the sentence as “It remains a mystery why such a low entropy of tightly packed gene cluster ever exists and evolves concertedly, as it is common to find genes at various genomic loci encoding the same pathway, functionally linked via gene co-regulation.”.

Line 280: Please be more precise. A nonrandom erosion of cis-element o the gene cluster on chromosome 8 in P. setigerum was shown, this gene cluster resulted from WGD-2, thus a duplicated cluster was downregulated.

RESPONSE: Thanks. We have revised the sentence as you suggested.

Line 284: What proof is available that noscapine pathway evolved after morphinan pathway? Again- in P. rhoeas, noscapine is present (according to the statement in the introduction), I would assume, that noscapine biosynthesis is old in Papaver.

RESPONSE: We think that the presence of pathway leading to biosynthesis of a particular compound and emerge of a gene cluster are different. According to patchwork model, the genes in a pathway might exist in a species to enable biosynthesis but scatter in the genome without co-regulation or co-location (gene clustering). In our study, the trace amount of noscapine biosynthesis in *P. rhoeas* indicates the presence of noscapine biosynthesis genes but those genes scatter in the genome, not optimized in a cluster as in *P. somniferum*. Comparing the BIA gene cluster locus of the three species, we found that noscapine branch in BIA gene cluster completed after morphinan branch in BIA gene cluster (Figure 4b) based on the systematic syntenic analysis.

Page 24:

Line 290: plants grown in (?) a growth chamber

RESPONSE: We have revised that as "grown in Azalea pots in a growth chamber".

Line 293: ONT sequencing – introduce abbreviation – move from line 302 to 293.

RESPONSE: We have revised in manuscript.

Page 27:

Line 352: introduce ancestral eudicot karyotype – AEK

RESPONSE: We have revised in manuscript.

Page 28: Divergence time estimates – calibration with fossil records would strengthen the divergence time estimates.

RESPONSE: The calibration with the split time is from timetree (<http://timetree.org/>), which is indeed based on the fossil records.

Line 368: Explain your approach more precise. I assume, within species comparisons of syntenic block were performed.

RESPONSE: We have revised the sentence as “To estimate the timing of the WGD event in *P. somniferum* and *P. setigerum*, *Ks* values of *P. somniferum* syntenic block genes and *Ks* values of *P. setigerum* syntenic block genes were calculated respectively using YN model in *KaKs_Calculator* (v2.0)”.

Line 376 & 379: Papaverance !? -

RESPONSE: We have revised this word.

Figures:

*Fig.1c: Please include color code/legend that was used to color the syntenic blocks. I assume its *Ks*?*

RESPONSE: The color in the dotplot of Figure 1c is generated by *MCScanX* automatically. Different colors indicate different syteny blocks.

Fig.1d:

*What should the close up illustrate? And its very hard to differentiate between the colored lines representing syntenic orthologs of *P.som.* – *P. set.* and *P. set.* – *P.rho.* in the figure. Please adjust the colors.*

RESPONSE: The close up in the figure illustrate the similarity of *Ks* distributions. We have revised the colors of Fig. 1d.

Figure2: The graphic figure legend is a bit un-ordered. I miss the explanation of the color code - red indicates enriched fusions, while blue indicates depletion of fusions.

RESPONSE: We have revised the figure legend.

Figure 3:

*In general, Fig. 3 is rather complex, which makes it difficult to follow. Maybe it would make the figure more intuitive, if the two colinear regions (the donor loci and the recipient loci) could be highlighted in different colors? The dashed lines _ . . _ . . are confusing, what do they exactly mark? I also don't understand, why the *P. setigerum* chromosomes are not grouped together. In Fig. 3b, the grey boxes in the recipient loci – are they necessary? Graphic legend Fig. 3a: the blue color of *STORR* is hardly to distinguish from the oxidoreductase pre-fusion module, anyway -would it not be more correct to give the same symbol for *STORR* in the graphic legend as in the figure itself (lilac rectangle together with blue arrow)?*

RESPONSE: Thanks for your comments. The detail of Figure 3a is shown in Supplementary Figure 20 in the revision manuscript. The colinear regions in donor and recipient loci are marked as gray box. The dashed lines _ . . _ . . indicating the boundary between donor loci and recipient loci,

and the boundary between prior states and post states. Because we grouped chromosomes by Prior and Post states rather than species, *P. setigerum* chromosomes are not clustered together, e.g. chr16 and chr17 in *P. setigerum* are recipient prior loci, while chr15 and chr8 in *P. setigerum* are recipient post loci. We have revised the figure legend of *STORR* in Figure 3a.

Fig. 4: The color code is confusing in in panel b – new gene from “fusion, translocation” same color as new gene from dispersed duplication. “dispersed duplication” – are paralogs present somewhere else in the genome?? Please explain what you mean and please elaborate the origin of this genes in more detail in the text!

RESPONSE: Sorry for the color code issue. We have revised the color code of Figure 4b. From Supplementary Figures 22 to 30 in the revision manuscript, we showed the evidences and potential origin of each gene. In addition, we describe the evidences and origin of each gene in Supplementary section 15 "BIA pathway evolution in three *Papaver* species".

Extended Data Fig. 3: Please explain the abbreviation GMP, and pPG. In general, the figure is not very intuitive. The figure should be able to stand alone. I would advice to revise it.

RESPONSE: Thanks for your comments. GMP is the genome median problem, proposed by David Sankoff in 1997 (Sankoff *et al*, *International Computing and Combinatorics Conference*, 1997). GMP describe the problem of finding ancestral genome structure that all blocks appear in related species with single copy. pPG is the putative protogene, proposed by Florent Murat in 2017 (Murat *et al*, *Nat Genet.*, 2017). pPG means the inferred genes in ancestral genomes. We have revised the figure and legend by adding some detailed explanation.

*Extended Data Fig. 4: Please include color code/legend that was used to color the syntenic block. And could you elaborate it bit on the findings, that we can get from the dotplots in the figure legend? E.g. in 4b – *P. somniferum* plotted against the post WGD-1 ancestor – chromosome 1 shows a syntenic block ration of 1:2, or in other words: a syntenic blocks in chromosome 1 *P. somniferum* “matches” chromosome 1 and chromosome 6 in the post-WGD-1 ancestor. What is the explanation? Small-scale duplication?*

RESPONSE: These dotplots were generated by *MCSanX*, and the color was generated by *MCSanX* automatically. Different colors indicate different syteny blocks. We have revised the figure legend to indicate this. Comparisons between three *Papaver* genomes and two reconstructed ancestors (pre-WGD-1 ancestor and post-WGD-1 ancestor) reveal the differences between the ancestor genomes and the modern *Papaver* genomes. For example, the comparison between post-WGD-1 ancestor and *P. somniferum* show a 2:2 block ratio, indicating both *P. somniferum* and post-WGD-1 ancestor have WGD event. A syntenic block in chromosome 1 of *P. somniferum* “matches” both chromosome 1 and chromosome 6 of the post-WGD-1 ancestor indicating chromosome 1 and chromosome 6 in post-WGD-1 ancestor were a duplicated pair resulted from WGD-1 rather than small-scale duplication.

REVIEWER COMMENTS

Reviewer #1 (Remarks to the Author):

The authors have made a revision that addresses some of the issues raised in the last review. The new figures in the supplementary materials now provide useful information that goes part of the way to assessing whether tandem vs. dispersed duplication is responsible for cluster evolution, but the study really has not clearly demonstrated this VERY important point. It is not suitable for publication until this aspect is clarified and actually tested rigorously (comment #2). In several places, I still feel that the manuscript is putting forward one possible explanation for how things evolved, but not really doing a good job to test among different alternative explanations, as described in the comments below. I think this paper has a lot of REALLY interesting data -- I really like the comparison with the other two poppy species -- but it is still being hampered by these aspects to the point where it is not yet suitable for publication.

LINE NUMBERS refer to the line numbers in the word document included as track changes.

Major comments:

1. The writing needs a lot of work. For example, the first few sentences of the abstract are unclear and halting in their style and not precise in their meaning:
 - line 22: selection operates on things, not selection of, and this implies that evolution only operates through selection on this variation, which is not true/general (selection also acts on SNPs, indels, rearrangements, etc).
 - line 23: what is "single genomic variation"? This is not standard terminology
 - line 25: what does "forwardly or backwardly" mean? What is the patchwork model? This is not self-evident and everything in the abstract should be crystal clear.
 - line 34: what does "evolved coordinately" mean? Again, this is non-standard and unclear.
 - line 34: what are "accumulated expressions". Again, non-standard and unclear.

These problems extend throughout the paper and go far beyond what I can reasonably edit as a reviewer -- the science is interest but it is not presented at the standard of a journal like Nature Communications, and this is not just about whether the language is polished but also whether it is clear in its presentation of ideas. This should not be a barrier to publication when the science is sound, but it is important that it is rendered in the proper style. As an example of how this hampers communication -- the introduction is too short and fails to review the main ideas being discussed. As an example, the patchwork model is only introduced on line 605 and then it simply uses the name of the model as the definition, which is not clear and this is not something well-known to a general audience. One line 608 it is said that the BIA cluster is the first example in plants of the patchwork model, but what about the extensive work done by Nutzman and Osborn (<http://dx.doi.org/10.1016/j.copbio.2013.10.009>) on clustering plant secondary metabolic pathways? Why does previous work not conform? I don't know what the patchwork model is, and I work in this general area of the field. This is just one example but in general I find the scholarship aspect of this paper (clarity of terminology, clarity of concepts, linking with existing literature) is still lacking and this is an important part of academic publication.

2. While the authors have responded to my comments about STORR, they haven't really changed the presentation in a substantial way. I don't think it will be evident to any reader why this shows evidence for STORR evolution as the authors explain. Figure 3 is a cartoon showing a hypothesis, it doesn't present any evidence. Figure S20 shows evidence, but it is so convoluted it is hard to see anything besides "it's complicated". I can accept the author's story about how they think STORR evolved as one possible explanation, but Figure S20 doesn't really make that particularly clear, and

other less-parsimonious explanations are possible (parsimony isn't always == truth when it comes to phylogenetics).

More importantly, the authors have largely ignored suggestions I made to improve Figure 4 and to more clearly identify whether genes arose from tandem duplication or dispersed duplication. I think it is REALLY important to nail this down because tandem duplication is very common and immediately leads to the evolution of a cluster, whereas the chance of a dispersed duplicate fortuitously landing right next to other genes to form a cluster is small --- so seeing evidence of this is very interesting but requires a high burden of proof. So the burden of proof here lies on showing that a gene is a dispersed duplicate (tandem duplicate is the null expectation). While I like this figure 4 as an illustration, I don't think this qualifies as comprehensive evidence about whether these genes arose from old tandem duplications vs. "dispersed duplication" -- the blastp results added to Figure 4 don't really allow us to assess whether a gene arose from a tandem vs. dispersed duplication -- they just show one line of "best blast hit", but this best blast hit could just be due to the WGD if that happened after an older tandem duplication. This can be assessed using Figure S31 but it's still hard to disentangle which copies are homologous because of the WGD vs. because of a dispersed duplication. I REALLY think the authors need to do a better job of discriminating between these hypotheses. As an example, consider PSMT3. The authors present a cartoon showing one possible way it evolved in Figure S28. However, examination of Figure S31 reveals other explanations could also be consistent with the data. The BIG problem to consider here is that PSMT3 could have evolved as an old tandem duplication of PSMT1 (or vice versa), long before the WGD. Figure 31 clearly shows there is homology between PSMT3 and PSMT1, but less than within the copies of PSMT3. Within the cluster labelled PSMT3, there are 3 Psom genes:

- Pso04G00300
- Pso01G11570
- Pso05G43680

Within the cluster labelled PSMT1, there is only one Psom gene:

- Pso04G00330

Thus, a hypothesis consistent with the data is that either PSMT1 or PSMT3 arose from a tandem-duplication (of the other) prior to the WGD, then after the WGD, the copy of PSMT1 was deleted from the other WGD segment, while another dispersed duplication of PSMT3 gave rise to the other copy (either Pso01G11670 or Pso05G43680, whichever is not part of the WGD) but not in any cluster. This seems more parsimonious than assuming that PSMT3 arose from a dispersed duplication and just happened to land right next to PSMT1, in the proper cluster. This simply cannot be evaluated using the gene tree and dot plot shown in Figure S28. We need to see analysis of WGD regions vs. putative dispersed duplicates vs. putative tandem duplicates (assessing all nearby genes, not just those that are immediate neighbours -- tandem duplication can duplicate several genes at once, that can subsequently be deleted).

Similar arguments apply for the other genes where the authors are claiming that a dispersed duplicate occurred and landed within the cluster (PSAT1, PSCXE1, THS, and SALAT). In each case, the authors need to conclusively disprove the hypothesis that an old tandem duplication gave rise to the putative "dispersed duplicate" by evaluating any cases where other genes nearby on the same chromosome have detectable/substantial %ID, as this may indicate an old tandem duplication. I think in many cases this simply won't be possible to confidently reconstruct what happened, but that's ok -- better to be clear about the uncertainty than be wrong. This is vitally important as it helps us understand how these clusters formed and the current analysis is simply not up to a high standard.

3. Given the ambiguities in how these genes have evolved (comment #2), I have doubts about the validity of "donor" and "recipient" locus definitions. This affects all usages of these terms (e.g. lines 363-393). For example, if PSMT1/PSMT3 arose from a tandem duplication, then the labelling of other

genes as "donor" would be wrong.

4. It is unclear why it is surprising that morphinan production is lower in setigerum than somniferum, given that somniferum has been actively under artificial selection for this trait. Also, the previous section just (nicely) showed that the promoter regions of one of the two clusters in setigerum have degraded, so it's even less surprising that setigerum has lower production. Unless I'm mistaken, the Li (2020) paper didn't show extensive correlation between copy number and production, they showed that deletions of particular genes corresponded to loss of alkaloid production, but not that increases in copy number were explanatory (?). Finding that setigerum has "twice as many copies of STORR..." (line 512) doesn't seem relevant after the previous section showing lower expression, but I see the point that is being made from line 520-532.

5. It is claimed that the role of transposable elements in gene duplication was investigated, but nothing conclusive is shown here, all that is shown is that there are TEs near these genes, this does little to nothing to clarify how these genes evolved. Claims must stand on evidence, and again, it feels like the authors are telling a story that goes along with our current understanding of how genomes evolve, rather than testing whether this is in fact what happened.

Minor comments:

- line 260-267: the concepts of donor and recipient loci are not clear as described.
- line 354: add reference to other places this has been reported
- line 568: this is not an example of punctuated equilibrium. Please reconsider the wording here.
- line 350: repeating a previous comment from my last review (!): please change all usage of "concerted" as this term already has another meaning (see https://en.wikipedia.org/wiki/Concerted_evolution). There are many other words that are more suitable, such as correlated or coordinated.
- line 528: seeing a difference in copy number of genes is not evidence of artificial selection. Again, this is another case of telling a story using data that is consistent with what we already know (somniferum experienced artificial selection), rather than using data to test among alternative hypotheses.

Reviewer #2 (Remarks to the Author):

The authors have explained and addressed the issues and questions raised by the reviewer.

Reviewer #3 (Remarks to the Author):

In the revision of the manuscript, titled, "Three chromosome-scale Papaver 1 genomes reveal punctuated patchwork pathway evolution leading to morphinan and noscapine biosynthesis", authors have considerably improved the content, especially by including content that appeals to a broader audience. The authors managed to address almost all my comments. The authors have now provided all scripts, including parameters that were used for genome assembly and analysis. After going through the responses to my queries, there are few points that remain unresolved, and I am listing them here.

1. I agree with your assessment. I do appreciate your explanation on why you have decided to re-do scaffolding. I need to mention here that I was one of the reviewers for Li et al. paper and I did flag the discrepancy on contig N50. But in their response, they mentioned that they added new datasets, and I think I missed few details that should have been asked for the previous group to describe and correct the stats. But my understanding is that the authors established genome assembly for PS7 lines, and the contig N50 for that assembly is over 7Mb. So, the N50 is not your assembly contig N50, but it's PS7. Could you please check and confirm this aspect? I know and understand that this is out of the scope of your study, but it is still essential to have a table with genome stats for PS7 and NH11 side by side. Since there are many genome assemblies, it would be better if readers get a sense of which assembly is of use for them. This table could be part of your supplementary table.
2. Another issue is related to PS7 line. While I am convinced about the rationale of re-scaffolding, my primary question remains as what the key differences between PS7 are and NH11 lines (this study). In terms of specialized metabolites biosynthesis, both lines are comparable, so, in terms of genome-wide differences, it may be helpful to perform such analysis and report. This part could be included in the supplementary information. As mentioned in my first comment, it is extremely important to bring the difference between these two lines clear to enables readers and the scientific community to choose the most appropriate assembly for their downstream analysis. Also, if these two lines have some structural differences at the genome levels that could be associated with the biosynthesis of key metabolites, that would help to identify or at least predict enzymes involved in the biosynthesis pathways. Again, as this part is slightly different from the original storyline, authors should provide this information in the supplementary info section.
3. I sincerely appreciate the authors effort to include section 8.4. It does improve this manuscript and its applications.
4. From the response of authors, "To solve this problem, we developed our method for three *Papaver* species with two closed WGD events based on the work of Sankoff (Zheng et al, *Evol Bioinform.* 2007) and Pedro Feijao (Feijão et al, *TCBB*, 2011). ", authors described why this method was needed. I am not an expert on ancestral genome reconstruction, but the rationale discussed here does make sense. I agree that the species that were picked for AEK construction does not have WGD, and therefore, probably that model is not applicable here. As plants do undergo whole-genome duplication more often as these serve as the basis for innovations in terms of new metabolites evolution, authors should describe this part in more detail in the manuscript. I think authors should consider moving their response to my query to the discussion part. Indeed, this is one of the aspects which will be helpful for people who are working on other complexed genomes and understand the rationale to choose this algorithm. Its strength and weakness should be discussed in more detail. Unless I have missed it, I am not able to find this part in the discussion section or anywhere in the paper.
5. From the author's response on comment 11, "If the editor and the Reviewer #3 insist that the genome assembly errors may compromise the conclusions about the number of chromosomal fissions and fusions, we are willing to remove the content related to chromosomal fissions and fusions". While I do find the interpretation and analysis intuitive, I also believe that since this genome is not experimentally validated, the chances of error can not be ruled out no matter how confident authors could be with their datasets. As I previously mentioned, using a hybrid assembly approach, *Ophiorrhiza pumila* genome an accurate assembly and quite improve assembly stat, yet, when they tested the assembly gaps experimentally, they could still find one mis-assembly. Compared to just 21 assembly gaps, It is hard for me to believe that for this study, assembly is perfect when this assembly has hundreds of gaps. I propose authors to include a cautionary statement in the results section and in the discussion section. Authors may include that the entire basis of this hypothesis is based on the accuracy of genome assembly, which despite using orthogonal evidence, could lead to misassembly. Even with the cutting-edge sequencing data and widely used assembly methods, assembly errors are inevitable, especially for genomic regions with large and repetitive segments. Authors could cite relevant articles here to make this point and to bring the importance of experimentally validated genome assemblies. I think in this way, authors will allow readers to choose and see this hypothesis with caution. I am not in favor of removing this part of the result, but I certainly want it to make clear to the readers what caution require to believe this result and why this caution is required.
6. Authors have included a small section on the gene-cluster part in the discussion (Page 29, line 387

onwards). I wonder if this part should move to page 32, line 443. Probably it will be more appropriate and go well with the flow. Just a minor comment, and its up to the authors to follow this suggestion. Further, in this section (page 32, line 443), the authors tried to describe the phenomena of gene clusters in plants by using, "Why gene clustering and non-clustering are found in the same biosynthetic pathway is intriguing". I feel authors should elaborate on this aspect. One of the key findings that is emerging is the fact that many of these gene clusters are syntenic, and probably, are key to retain or loose a metabolite phenotype. Studies on Arabidopsis, Ophiorrhiza pumila, Tomato, Opium poppy, and several other studies have different interpretations. Few studies have proposed co-expression, and few have discussed gene-clustering as a means to have the advantage of gene segregation.

Other than these, I am satisfied with the answers provided by the authors.

Reviewer #4 (Remarks to the Author):

Dear authors, dear editors,

The revised manuscript was improved in many aspects. A general remark: the authors answered all my questions very detailed in their rebuttal letter, but unfortunately these questions did not motivate the authors to clarify the raised issues in the manuscript. Still, I recommend to accept the manuscript, as it includes interesting results and insights into how pathways evolve.

ELisabeth Kaltenegger

Specific comments, that should be considered by the authors:

Page 4

"Genome assembly and annotation":

I still believe that the occurrence of the different BIAS in the three Papaver species, that were analysed by the authors, do not fit to this paragraph where genome assembly and annotation is described. As the BIA bouquet is the phenotype, that is studied, and the basis of their hypothesis, it should be described in an individual chapter. The data is available \diamond suppl. Fig. 2. According to this Fig., Papaver rhoes produces low amount of Noscapine, Thebaine, Codeine, Morphine, which supports the theory, that P. rhoes has a "primitive" BIA pathway. Also, the author nicely described this primitive pathway in their rebuttal: "P. rhoes has a primitive metabolic pathway, leading to the formation of Dopamine and 4-Hydroxyphenylacetaldehyde, the precursor of morphinans (the genes on the pathway are: PrhUNG23530.0 (TYDC), Prh05G30560.0 (TryAT)). It also has the NCS (PrhUNG12800.0), encoding the enzyme to convert Dopamine and 4-Hydroxyphenylacetaldehyde into (S)-Norcoclaurine, the first step of morphinan biosynthesis." I think this information is very interesting not only to me, but also to the broad audience. I would recommend to include it in the main manuscript.

Page 6

Line 93: „In addition, a collinearity confirming a lack of the ... in Papaver genus"... please check the grammar of the sentence.

Line 97:

The authors nicely explained in the rebuttal letter how they estimated the divergence time and refer to the suppl. Section 8.1. I would recommend to include this very clear explanations in the main manuscript. The paragraph in its actual format is in my opinion confusing.

Page 10 – associated to Fig.2:

I think this explanation should also be included in the main manuscript: "After reassignment of chromosome IDs, we found the BIA gene cluster were located at chr4 rather than chr11." This is an important result of you re-assembly and explains the discrepancy between your previous and recent publications.

Page 11, Line 141:

Sorry, you did not get my point. Although *P. somniferum* and *P. setigerum* share the WGD-1, only in *P. somniferum* the gene cluster including the 10 noscapine and the 5 morphinan genes evolved while in *P. setigerum*, only the "5 morphinan gene cluster" evolved. Thus, the post WGD diploidization differ strongly between the two species. I think this is the most intriguing result of your research. Also, having your analyses of BIAs in the three *Papaver* species in mind – "innovation of the alkaloid biosynthesis pathways" is not a correct term, as *P. rhoeas* – most likely resembling the ancestral state, as it did not experience any WGD event - already produces trace amounts of morphinan and noscapine. So, evolution & selection optimized the biosynthesis of both BIA types in *S. somniferum*.

Page 14, "dispersed duplications"

Thanks for your explanation. This is very interesting. I would suggest to include a shortened version of this explanation to your main manuscript, maybe this would be helpful and interesting for the broader audience. The term "dispersed duplication" alone is not very informative.

Page 18: Line 215

P. rhoeas, that possesses the "basic" genetic components of BIA pathway, produces morphinan and noscapine in traces, thus you should adjust your statement. Its rather not "in principle,....should allow...." . It seems to be a fact, that the simple and ancestral pathway truly is sufficient to produce traces of morphinan and noscapine. Especially concerning your explanation in the rebuttal (see above).

Page 19, Line 223: no, you did not revise the sentence. The "compared to" is still missing.....

Thank you for all your valuable comments. We have provided a response letter addressing all the issues raised by the reviewers. For clarity, all reviewer comments or quoted contents are in italicized fonts. A point-to-point response to each comment is provided in normal fonts. References to revised manuscript contents are also provided where needed.

REVIEWER COMMENTS

Reviewer #1 (Remarks to the Author):

The authors have made a revision that addresses some of the issues raised in the last review. The new figures in the supplementary materials now provide useful information that goes part of the way to assessing whether tandem vs. dispersed duplication is responsible for cluster evolution, but the study really has not clearly demonstrated this VERY important point. It is not suitable for publication until this aspect is clarified and actually tested rigorously (comment #2). In several places, I still feel that the manuscript is putting forward one possible explanation for how things evolved, but not really doing a good job to test among different alternative explanations, as described in the comments below. I think this paper has a lot of REALLY interesting data -- I really like the comparison with the other two poppy species -- but it is still being hampered by these aspects to the point where it is not yet suitable for publication.

LINE NUMBERS refer to the line numbers in the word document included as track changes.

RESPONSE: Thanks for your comments. We have provided a clarification of the point you suggested and accordingly revised our manuscript by tuning down the conclusions and discussing the alternative explanations.

Major comments:

1. The writing needs a lot of work. For example, the first few sentences of the abstract are unclear and halting in their style and not precise in their meaning:

- line 22: selection operates on things, not selection of, and this implies that evolution only operates through selection on this variation, which is not true/general (selection also acts on SNPs, indels, rearrangements, etc).

- line 23: what is "single genomic variation"? This is not standard terminology

- line 25: what does "forwardly or backwardly" mean? What is the patchwork model? This is not self-evident and everything in the abstract should be crystal clear.

- line 34: what does "evolved coordinately" mean? Again, this is non-standard and unclear.

- line 34: what are "accumulated expressions". Again, non-standard and unclear.

These problems extend throughout the paper and go far beyond what I can reasonably edit as a reviewer -- the science is interest but it is not presented at the standard of a journal like Nature Communications, and this is not just about whether the language is polished but also whether it is

clear in its presentation of ideas. This should not be a barrier to publication when the science is sound, but it is important that it is rendered in the proper style. As an example of how this hampers communication -- the introduction is too short and fails to review the main ideas being discussed. As an example, the patchwork model is only introduced on line 605 and then it simply uses the name of the model as the definition, which is not clear and this is not something well-known to a general audience. One line 608 it is said that the BIA cluster is the first example in plants of the patchwork model, but what about the extensive work done by Nutzman and Osborn (<http://dx.doi.org/10.1016/j.copbio.2013.10.009>) on clustering plant secondary metabolic pathways? Why does previous work not conform? I don't know what the patchwork model is, and I work in this general area of the field. This is just one example but in general I find the scholarship aspect of this paper (clarity of terminology, clarity of concepts, linking with existing literature) is still lacking and this is an important part of academic publication.

RESPONSE: Thanks for your comments. We apologize for the unclarity in our writing and have edited the paper thoroughly. Based on your questions and suggestions on the writing, we have carefully revised the abstract and introduction, and also made several necessary changes in other sections. We added a brief introduction of pathway evolution models (line 78-84) to make it clear to a general audience. We trust you will find the writing of manuscript is improved in the current format.

2. While the authors have responded to my comments about STORR, they haven't really changed the presentation in a substantial way. I don't think it will be evident to any reader why this shows evidence for STORR evolution as the authors explain. Figure 3 is a cartoon showing a hypothesis, it doesn't present any evidence. Figure S20 shows evidence, but it is so convoluted it is hard to see anything besides "it's complicated". I can accept the author's story about how they think STORR evolved as one possible explanation, but Figure S20 doesn't really make that particularly clear, and other less-parsimonious explanations are possible (parsimony isn't always == truth when it comes to phylogenetics).

More importantly, the authors have largely ignored suggestions I made to improve Figure 4 and to more clearly identify whether genes arose from tandem duplication or dispersed duplication. I think it is REALLY important to nail this down because tandem duplication is very common and immediately leads to the evolution of a cluster, whereas the chance of a dispersed duplicate fortuitously landing right next to other genes to form a cluster is small --- so seeing evidence of this is very interesting but requires a high burden of proof. So the burden of proof here lies on showing that a gene is a dispersed duplicate (tandem duplicate is the null expectation). While I like this figure 4 as an illustration, I don't think this qualifies as comprehensive evidence about whether these genes arose from old tandem duplications vs. "dispersed duplication" -- the blastp results added to Figure 4 don't really allow us to assess whether a gene arose from a tandem vs. dispersed duplication -- they just show one line of "best blast hit", but this best blast hit could just be due to the WGD if that happened after an older tandem duplication.

This can be assessed using Figure S31 but it's still hard to disentangle which copies are homologous because of the WGD vs. because of a dispersed duplication. I REALLY think the authors need to do a better job of discriminating between these hypotheses. As an example,

consider PSMT3. The authors present a cartoon showing one possible way it evolved in Figure S28. However, examination of Figure S31 reveals other explanations could also be consistent with the data. The BIG problem to consider here is that PSMT3 could have evolved as an old tandem duplication of PSMT1 (or vice versa), long before the WGD. Figure 31 clearly shows there is homology between PSMT3 and PSMT1, but less than within the copies of PSMT3. Within the cluster labelled PSMT3, there are 3 Psom genes:

- Pso04G00300

- PSo01G11570

- PSo05G43680

Within the cluster labelled PSMT1, there is only one Psom gene:

- Pso04G00330

Thus, a hypothesis consistent with the data is that either PSMT1 or PSMT3 arose from a tandem-duplication (of the other) prior to the WGD, then after the WGD, the copy of PSMT1 was deleted from the other WGD segment, while another dispersed duplication of PSMT3 gave rise to the other copy (either Pso01G11670 or Pso05G43680, whichever is not part of the WGD) but not in any cluster. This seems more parsimonious than assuming that PSMT3 arose from a dispersed duplication and just happened to land right next to PSMT1, in the proper cluster. This simply cannot be evaluated using the gene tree and dot plot shown in Figure S28. We need to see analysis of WGD regions vs. putative dispersed duplicates vs. putative tandem duplicates (assessing all nearby genes, not just those that are immediate neighbours -- tandem duplication can duplicate several genes at once, that can subsequently be deleted).

Similar arguments apply for the other genes where the authors are claiming that a dispersed duplicate occurred and landed within the cluster (PSAT1, PSCXE1, THS, and SALAT). In each case, the authors need to conclusively disprove the hypothesis that an old tandem duplication gave rise to the putative "dispersed duplicate" by evaluating any cases where other genes nearby on the same chromosome have detectable/substantial %ID, as this may indicate an old tandem duplication. I think in many cases this simply won't be possible to confidently reconstruct what happened, but that's ok -- better to be clear about the uncertainty than be wrong. This is vitally important as it helps us understand how these clusters formed and the current analysis is simply not up to a high standard.

RESPONSE: We inferred the putative ancestral origin of BIA biosynthetic genes and constructed possible evolutionary models using the three *Papaver* genome data, based on our comparative analysis, current computational strategies and parsimonious assumption. We believe that sequencing more species, with novel computational methods becoming available and using less-parsimonious assumption may lead to an updated evolution model of BIA gene cluster. Given the current data, we admit that the alternative explanations of BIA gene evolution as reviewer #1 pointed out are equally possible. Thus, we tuned down the conclusions of the BIA genes evolution in section "Recruitment of new genes to BIA gene cluster locus" and "Discussion" as follows. We revised our paper accordingly to acknowledge the alternative explanations.

- Line 258-261 "We found all four types of loci appeared exactly once in *P. somniferum*,

but twice in *P. setigerum*, while only the prior states of donor and recipient loci were observed once in *P. rhoeas* (Fig. 3a, Supplementary Figs. 20, 21), supporting our hypothesis.”

- Line 263-265, in Figure 3 legend “The collinearity of genes in the donor loci and the recipient loci showing both prior and post states of FT event in three *Papaver* species supporting the "fusion, translocation" (FT) event leading to STORR formation at BIA gene cluster. ”
- Line 272-274 “Considering the phylogeny and WGD history among the three species, we proposed an evolutionary model based on the parsimonious assumption to illustrate a burst of genomic rearrangements giving the birth of STORR in current BIA cluster (Fig. 3b).”
- Line 281-285 “Our *STORR* evolutionary model, inferred using the three *Papaver* genome data, based on our comparative analysis, current computational strategies and parsimonious assumption, represents one hypothesis. More species sequencing data with novel computational methods and less-parsimonious assumption may lead to an updated evolutionary model.”
- Line 296-304 “Specifically, we inferred five genes including *PSCXE1*, *PSAT1*, *PSMT3*, *SALAT*, and *THS* may be assembled into BIA gene cluster by putative dispersed duplications (the gene and their inferred original copy are non-syntenic and not adjacent) while *CYP82X2* was a putative tandem duplication of *CYP82X1* (Fig. 4b, Supplementary Figs. 22-30). Considering the difficulty of distinguishing dispersed duplication from old tandem duplication with gene deletions, we do not rule out the possibility that *PSCXE1*, *PSAT1*, *PSMT3*, *SALAT*, and *THS* were assembled into BIA gene cluster by ancient tandem duplications with follow-up gene deletions (Supplementary Fig. 32).”
- Line 424-430 “We showed that duplication was a major factor contributing to the formation of BIA gene cluster. The origin and putative type of the duplications were inferred based on synteny, WGD and *BlastP* analysis of three *Papaver* genomes. Alternative explanations of the BIA gene cluster formation such as "old tandem duplications" or other less-parsimonious explanations is possible but shall be tested with more sequencing data and genome analyses.”
- We tuned down the duplication types as "putative dispersed duplication" and "putative tandem duplication" in manuscript and revised the Figure 4.
- We also added a new Supplementary Figure 32 to discuss alternative explanations of *PSMT3*, *THS*, *SALAT*, *PSAT1*, *PSCXE1* based on old tandem duplications.

For your comments on *STORR*, we are delighted about your acceptance of our interpretation. In addition, reviewer #3 also appreciates our evolutionary model of *STORR* by saying "*Authors have a very elegantly description of the origin of STORR*" (see the comments #12 of reviewer #3 in previous review comments). Moreover, we think that our parsimonious explanation of *STORR* formation based on current three *Papaver* genomes is one of the many possible explanations. We

agree with you that "parsimony isn't always == truth when it comes to phylogenetics", so we tuned down our conclusions about *STORR* evolutionary model and revised the results as well as discussion sections.

For comments on Figure 4, in our last response to your previous comments, we thought that we have addressed your previous comments as follows. Would you please let us know what additional modifications are needed?

- We added the Supplementary Figure 31 to show the identity of all genes in noscapine and morphinan clusters as well as their close paralogs.
- We added the detail explanation of the evidence to infer the ancestral origin and duplication type of each novel gene in BIA gene cluster.
- For "*This would be SO much clearer if the paralogs to the pre-fusion modules within somniferum were shown on Figure 4. Where are they?*" from your last comment about Figure 4, we have shown the paralogs of pre-fusion modules in three *Papaver* species in Figure 3a and Supplementary Figure 20, and these two figures showed the evidence of our *STORR* story. In our manuscript, Figure 3 is the story of *STORR* and Figure 4 is the story of whole BIA cluster evolution. The flow is from one gene to whole cluster, and we think this flow is smooth.

For comments on other genes (*PSMT3*, *THS*, *SALAT*, *PSATI*, *PSCXE1*), thanks for the alternative explanation you pointed out. We revised the manuscript to include the alternative explanation and added Supplementary Figure 32 (**Figure R1**).

Figure R1. The alternative possible explanations based on old tandem duplications of the formation of *SALAT* (a), *THS* (b), *PSAT1* (c), *PSCXE1* (d), and *PSMT3* (e).

3. Given the ambiguities in how these genes have evolved (comment #2), I have doubts about the validity of "donor" and "recipient" locus definitions. This affects all usages of these terms (e.g. lines 363-393). For example, if *PSMT1/PSMT3* arose from a tandem duplication, then the labelling of other genes as "donor" would be wrong.

RESPONSE: We apologize for the unclear statement leading to your misunderstanding. At lines 363-393, we meant to say the ancestral origin and the BIA gene cluster rather than "donor" and "recipient" locus. In addition, we defined "donor" and "recipient" locus based on the formation of *STORR*. For other genes, we did not define the donor and recipient locus. We have revised the

misused terms in our manuscript at lines 333-335. We hope our revision is clear.

4. *It is unclear why it is surprising that morphinan production is lower in setigerum than somniferum, given that somniferum has been actively under artificial selection for this trait. Also, the previous section just (nicely) showed that the promoter regions of one of the two clusters in setigerum have degraded, so it's even less surprising that setigerum has lower production. Unless I'm mistaken, the Li (2020) paper didn't show extensive correlation between copy number and production, they showed that deletions of particular genes corresponded to loss of alkaloid production, but not that increases in copy number were explanatory (?). Finding that setigerum has "twice as many copies of STORR..." (line 512) doesn't seem relevant after the previous section showing lower expression, but I see the point that is being made from line 520-532.*

RESPONSE: We agree that it is not surprising that *P. setigerum* has lower production since the artificial selection of *P. somniferum* and the highly expression of BIA related genes. We have revised the manuscript.

For the correlation between copy number and BIA productions, Li *et al.* described the extensive correlation between copy number and production, and showed the correlations of production with copy number deletion and copy number increase in their paper. The evidence in the paper includes the following:

- In Li *et al.* (2020) paper "Abstract", they said "*Copy number variation in critical BIA genes correlates with stark differences in alkaloid production linking noscapine production with an 11-gene deletion, and increased thebaine/decreased morphine production with deletion of a T6ODM cluster*".
- In their "CNV among cultivars" section they said "*Cultivars with similar CNV profiles in their BIA genes (Fig. 6f) also tended to have similar profiles in their alkaloid production (Fig. 6e; Mantel test on distance matrix: $r = 0.75$, p value = 0.002)*"
- Still in "CNV among cultivars" section, they said "*At the individual gene level, strong covariation was observed between CNVs and the alkaloids produced by the genes involved. Most prominently, the entire noscapine pathway containing 11 biosynthetic genes is deleted in five of the 10 cultivars (L, P, T, 11 and 40; Fig. 5), which is linked to lack of noscapine production in these cultivars (Fig. 6a). MLP-15 is massively duplicated across all cultivars, with fifteen copies observed in cultivars M, PS1, PS4, PS7, and BC (Fig. 5), which is reflected in its extreme expression profile in the latex (Supplementary Figs. 7, 8). As these varieties also produce high levels of noscapine*".

For the transition between our sections "Clustered BIA genes are co-regulated and evolved in a coordinated manner" and "Summation of gene expression contributes to morphinan production", we have revised the manuscript as you suggested for consistency.

5. *It is claimed that the role of transposable elements in gene duplication was investigated, but nothing conclusive is shown here, all that is shown is that there are TEs near these genes, this*

does little to nothing to clarify how these genes evolved. Claims must stand on evidence, and again, it feels like the authors are telling a story that goes along with our current understanding of how genomes evolve, rather than testing whether this is in fact what happened.

RESPONSE: We agree with you. About three fourth of *Papaver* genomes are repetitive elements and about half are TEs, making it difficult to associate specific TEs with the recruitment of individual BIA gene. As you said "nothing conclusive is shown here", we removed this part from our manuscript.

Minor comments:

- line 260-267: the concepts of donor and recipient loci are not clear as described.

RESPONSE: We have revised the definition of donor and recipient loci in manuscript and added genomic coordinate information in Supplementary Figure 20 to make the concept clear.

- line 354: add reference to other places this has been reported

RESPONSE: We have added the reference.

- line 568: this is not an example of punctuated equilibrium. Please reconsider the wording here.

RESPONSE: We have removed punctuated equilibrium in our manuscript.

- line 350: repeating a previous comment from my last review (!): please change all usage of "concerted" as this term already has another meaning (see https://en.wikipedia.org/wiki/Concerted_evolution). There are many other words that are more suitable, such as correlated or coordinated.

RESPONSE: Thanks for your suggestion. We have used coordinated evolution to describe the phenomenon we found.

- line 528: seeing a difference in copy number of genes is not evidence of artificial selection. Again, this is another case of telling a story using data that is consistent with what we already know (somniferum experienced artificial selection), rather than using data to test among alternative hypotheses.

RESPONSE: Thanks, we agree with you and have revised this part in our manuscript.

Reviewer #2 (Remarks to the Author):

The authors have explained and addressed the issues and questions raised by the reviewer.

RESPONSE: Thanks for all of your valuable comments to help us to improve our manuscript.

Reviewer #3 (Remarks to the Author):

In the revision of the manuscript, titled, "Three chromosome-scale Papaver I genomes reveal punctuated patchwork pathway evolution leading to morphinan and noscapine biosynthesis", authors have considerably improved the content, especially by including content that appeals to a broader audience. The authors managed to address almost all my comments. The authors have now provided all scripts, including parameters that were used for genome assembly and analysis. After going through the responses to my queries, there are few points that remain unresolved, and I am listing them here.

1. I agree with your assessment. I do appreciate your explanation on why you have decided to re-do scaffolding. I need to mention here that I was one of the reviewers for Li et al. paper and I did flag the discrepancy on contig N50. But in their response, they mentioned that they added new datasets, and I think I missed few details that should have been asked for the previous group to describe and correct the stats. But my understanding is that the authors established genome assembly for PS7 lines, and the contig N50 for that assembly is over 7Mb. So, the N50 is not your assembly contig N50, but it's PS7. Could you please check and confirm this aspect? I know and understand that this is out of the scope of your study, but it is still essential to have a table with genome stats for PS7 and NH11 side by side. Since there are many genome assemblies, it would be better if readers get a sense of which assembly is of use for them. This table could be part of your supplementary table.

RESPONSE: In results section "Building an improved assembly by Hi-C resc scaffolding" of Li's paper, they said "Our effort to improve the genome assembly of *Papaver somniferum* was based on re-scaffolding the recently published high-quality draft assembly using a *de novo* Hi-C dataset (see section "Methods"). After Hi-C scaffolding, contigs clustered into 11 chromosome-scale scaffolds (hereafter, cScaffs, Supplementary Fig. 2)" and "We further evaluated the accuracy of these assemblies by mapping $\sim 10\times$ PacBio reads from two libraries: the HN1 cultivar used by Guo et al. and the PS7 cultivar used in our Hi-C assembly (Supplementary Method 1). We then counted the number of scaffolded-gaps with >1 read mapping across the gap (mapping quality >10). Reads from both cultivars showed similar patterns, with $\sim 25\%$ of scaffold joins in our Hi-C assembly having mapped reads, compared to $\sim 4.7\%$ of scaffold joins in the Guo et al. assembly (Supplementary Table 3). Taken together, these analyses show that the Hi-C scaffolding has improved both assembly quality and contiguity, but that it remains a draft quality genome.".

These indicated three points:

- Li et al. just did a re-scaffolding work based on our HN1 contigs and *de novo* Hi-C data from PS7 cultivar
- Except the Hi-C data, Li et al. also generated $\sim 10\times$ PacBio data of PS7 cultivar. However, the PacBio data was just used to validate the resc scaffolding results. They have not used the PacBio data for *de facto* genome assembly.
- Li et al. also about 28x Illumina Paired-End data of PS7 cultivar. However, the data is

just used to estimate the copy number of key BIA genes. They have not used the Illumina data to assembly any new contigs.

Therefore, the contigs of Li assembly are from our HN1 cultivar, while the scaffold structure is from PS7 cultivar.

For the contig N50 issue, we checked the global statistics of Li's NCBI submission (PRJNA508405) which Li *et al.* proposed in Data availability section. We found that the contig N50 is 1,838,815 (https://www.ncbi.nlm.nih.gov/assembly/GCA_010119995.1) rather than over 7 Mb (**Figure R2**), consistent with what we recalculated (Figure R8 in our previous response letter).

ASM1011999v1

Organism name: Papaver somniferum (opium poppy)
Infraspecific name: Cultivar: Roxanne
BioSample: SAMN10524692
BioProject: PRJNA508405
Submitter: University of Calgary
Date: 2020/02/04
Assembly level: Chromosome
Genome representation: full
GenBank assembly accession: GCA_010119995.1 (latest)
RefSeq assembly accession: n/a
RefSeq assembly and GenBank assembly identical: n/a
Assembly method: Phase Genomics' Proximo Hi-C genome scaffolding platform v. 2019; Juicebox v. 1.9.1
Expected final version: no
Reference guided assembly: ASM357369v1
Genome coverage: 13.0x
Sequencing technology: Illumina

IDs: 5703781 [UID] 16532568 [GenBank]

History (Show revision history)

Comment

Global statistics

Total sequence length	2,637,752,765
Total ungapped length	2,636,327,527
Gaps between scaffolds	0
Number of scaffolds	11
Scaffold N50	252,943,167
Scaffold L50	5
Number of contigs	9,646
Contig N50	1,838,815
Contig L50	415
Total number of chromosomes and plasmids	11
Number of component sequences (WGS or clone)	11

Figure R2. The NCBI page about Li's submission. The global statistics show contig N50 is 1.8 Mb.

To find the reason of the incorrect contig N50 in Li's paper, we double checked the methods section "*Hi-C resc scaffolding and positioning of the BIA genes*", and found the re-scaffolding method description as "*The sequences of the Guo assembly were broken into segments at gaps of 100 Ns, representing the inter-scaffold gaps of unknown size, and at large gaps of ≥ 1000 Ns. The resulting split genome was comprised of 35,732 resulting segments, which were considered as contigs in the subsequent processing. The lengths of these contigs varied from 132 bp to 38.3 Mb, with the N50 length and the N50 number of the contigs of 7.6 Mb and 104, which provides a draft assembly with sufficient contiguity for making high confidence Hi-C based proximity-guided*

rescaffolding". The gap size of our assembly at 2018 (Guo assembly) is estimated according to the distance of paired-end and mate-pair links (Supplementary Materials "Genome assembly" in Guo *et al.* 2018): "Scaffolds assembly. Later, contigs were linked into scaffolds with paired-end and mate-pair information, estimating gaps between the contigs according to the distance of paired-end and mate-pair links. In addition, 10X data was used to validate and support correct phasing during scaffolding", and there are many gaps smaller than 100 Ns (**Figure R3**). Therefore, Li *et al.* ignored these smaller gaps and considered "the contigs" as sequence with gaps smaller than 100 Ns. However, this is incorrect because contig means "continuous sequence without any gaps" (definition is from <https://www.ncbi.nlm.nih.gov/grc/help/definitions/>). These problems led to an inflated contig N50 being reported in Li's paper.

Here, we would like to point that the sequence in Li's assembly is still from HN1 (ours) rather than PS7 (Li's) and our improved assembly of HN1 should be considered as the reference genome of opium poppy. We have revised Supplementary Table 2 by adding the comparison.

Figure R3. Gap length distribution in Guo assembly. (a) All gap length. (b) Gap length small than 500 bp.

2. Another issue is related to PS7 line. While I am convinced about the rationale of re-scaffolding, my primary question remains as what the key differences between PS7 are and NH11 lines (this study). In terms of specialized metabolites biosynthesis, both lines are comparable, so, in terms of genome-wide differences, it may be helpful to perform such analysis and report. This part could be included in the supplementary information. As mentioned in my first comment, it is extremely important to bring the difference between these two lines clear to enables readers and the scientific community to choose the most appropriate assembly for their downstream analysis. Also, if these two lines have some structural differences at the genome levels that could be associated with the biosynthesis of key metabolites, that would help to identify or at least predict enzymes involved in the biosynthesis pathways. Again, as this part is slightly different from the original storyline, authors should provide this information in the supplementary info section.

RESPONSE: We have obtained HN1 cultivar from our collaborator Ian A. Graham group, but we don't have PS7 cultivar. Chance is slim that we will obtain the PS7 cultivar from Facchini group due to the pandemic situation and opium poppy being an internationally regulated crop. Therefore,

we do not know the differences between these two cultivars in terms of biosynthesis of key metabolites, which is also beyond the scope of this paper.

As for genome level difference, we are not able to compare our HN1 genome assembly and Li's PS7 re-scaffold results because Li *et al.* used our HN1 contigs. To compute the structural variation between these two cultivars, we downloaded the Illumina Paired-End data of PS7 ([https://www.ncbi.nlm.nih.gov/sra/SRX5120089\[accn\]](https://www.ncbi.nlm.nih.gov/sra/SRX5120089[accn])), and detected 20,340 deletions, 3,487 insertions, 2,289 duplications and 20,836 BNDs by *Manta* software.

As our response of your comment #1, Li's assembly is just a re-scaffolding of our HN1 contig by Hi-C data from a different cultivar PS7 without generating any new contigs, while our improved assembly is generated by our HN1 contig and HN1 Hi-C data. Therefore, our improved assembly should be still considered as the reference genome for downstream analysis.

We provided the vcf file at <https://drive.google.com/file/d/1dAvlG1ferHt8RhLzn7zcY6QaVrCepvUr/view?usp=sharing>, however, we hesitate to include the mutation results into our work due to following two reasons:

- We do not have the PS7 cultivar to estimate the error rate;
- Our story focus on the inter-species comparison, and the inter-cultivar analysis is beyond our scope.

3. I sincerely appreciate the authors effort to include section 8.4. It does improve this manuscript and its applications.

RESPONSE: Thank you.

4. From the response of authors, "To solve this problem, we developed our method for three Papaver species with two closed WGD events based on the work of Sankoff (Zheng et al, Evol Bioinform. 2007) and Pedro Feijao (Feijão et al, TCBB, 2011). ", authors described why this method was needed. I am not an expert on ancestral genome reconstruction, but the rationale discussed here does make sense. I agree that the species that were picked for AEK construction does not have WGD, and therefore, probably that model is not applicable here. As plants do undergo whole-genome duplication more often as these serve as the basis for innovations in terms of new metabolites evolution, authors should describe this part in more detail in the manuscript. I think authors should consider moving their response to my query to the discussion part. Indeed, this is one of the aspects which will be helpful for people who are working on other complex genomes and understand the rationale to choose this algorithm. Its strength and weakness should be discussed in more detail. Unless I have missed it, I am not able to find this part in the discussion section or anywhere in the paper.

RESPONSE: Thanks for your comments. We have revised our manuscript to clarify the functions of the post-WGD diploidization in three *Papaver* species. The detail information about this part was shown in Supplementary Material section 9.4. We agree that the discussion of strength and weakness of our method will be helpful for researchers who are working on other complex

genomes. The main story of this manuscript is about gene cluster evolution and this discussion will interrupt the flow. Therefore, we revised the methods section "Ancestral genome reconstruction" by adding the response as your suggestion.

*5. From the author's response on comment 11, "If the editor and the Reviewer #3 insist that the genome assembly errors may compromise the conclusions about the number of chromosomal fissions and fusions, we are willing to remove the content related to chromosomal fissions and fusions". While I do find the interpretation and analysis intuitive, I also believe that since this genome is not experimentally validated, the chances of error can not be ruled out no matter how confident authors could be with their datasets. As I previously mentioned, using a hybrid assembly approach, *Ophiorrhiza pumila* genome an accurate assembly and quite improve assembly stat, yet, when they tested the assembly gaps experimentally, they could still find one mis-assembly. Compared to just 21 assembly gaps, It is hard for me to believe that for this study, assembly is perfect when this assembly has hundreds of gaps. I propose authors to include a cautionary statement in the results section and in the discussion section. Authors may include that the entire basis of this hypothesis is based on the accuracy of genome assembly, which despite using orthogonal evidence, could lead to misassembly. Even with the cutting-edge sequencing data and widely used assembly methods, assembly errors are inevitable, especially for genomic regions with large and repetitive segments. Authors could cite relevant articles here to make this point and to bring the importance of experimentally validated genome assemblies. I think in this way, authors will allow readers to choose and see this hypothesis with caution. I am not in favor of removing this part of the result, but I certainly want it to make clear to the readers what caution require to believe this result and why this caution is required.*

RESPONSE: Thanks for your kind suggestion. We added the cautionary statement in both results section "Ancestral genomes and accelerated nonrandom post-WGD rearrangements" and methods section "Ancestral genome reconstruction". As you commented, we agree that the research of reconstruction ancestral genomes (not only our method) will be affected by the quality of assembly. Therefore, for investigated individual important fission and fusion patterns, experimentally validated genome assemblies are important to obtain more reliable conclusions.

*6. Authors have included a small section on the gene-cluster part in the discussion (Page 29, line 387 onwards). I wonder if this part should move to page 32, line 443. Probably it will be more appropriate and go well with the flow. Just a minor comment, and its up to the authors to follow this suggestion. Further, in this section (page 32, line 443), the authors tried to describe the phenomena of gene clusters in plants by using, "Why gene clustering and non-clustering are found in the same biosynthetic pathway is intriguing". I feel authors should elaborate on this aspect. One of the key findings that is emerging is the fact that many of these gene clusters are syntenic, and probably, are key to retain or loose a metabolite phenotype. Studies on *Arabidopsis*, *Ophiorrhiza pumila*, *Tomato*, *Opium poppy*, and several other studies have different interpretations. Few studies have proposed co-expression, and few have discussed gene-clustering as a means to have the advantage of gene segregation.*

RESPONSE: Thanks for your comment. We have provided more discussion regarding the gene clustering by putting our work in the context of similar studies in this field, and adding several key references. We trust you will find the point is clarified in our revision as the following (Line 488-500):

"Among plant metabolic gene clusters, the genetic architecture of a core gene cluster plus several peripheral or satellite genes is rather common^{9,54}. The gene clusters can be continuous without any intervening genes such as avenacin (oat) gene cluster³⁴, while some have genes of unknown function separating the cluster genes such as thalianol²⁵, DIMBOA⁵⁶ and morphinan gene cluster^{2,24}. There are also biosynthetic pathways such as tomatine, morphine and cucurbitacin C encoded by a core cluster with satellite genes or cluster⁵⁴. The morphine biosynthetic pathway is composed of a core cluster coupled with several peripheral genes (in *trans*) or satellite gene groups²⁴. Recent studies also show that collinearity exists within a single gene cluster such as noscapine gene cluster where the cluster is organized in modules that correspond to early, middle and late steps of the pathway². The diverse architecture of metabolic gene clusters shows the evolutionary history of gene clustering for secondary metabolic pathways is complex and species-specific, involving complicated genomic variations and selection processes."

Other than these, I am satisfied with the answers provided by the authors.

RESPONSE: Thanks for your valuable comments.

Reviewer #4 (Remarks to the Author):

Dear authors, dear editors,

The revised manuscript was improved in many aspects. A general remark: the authors answered all my questions very detailed in their rebuttal letter, but unfortunately these questions did not motivate the authors to clarify the raised issues in the manuscript. Still, I recommend to accept the manuscript, as it includes interesting results and insights into how pathways evolve.

ELisabeth Kaltenegger

RESPONSE: Thanks for your valuable comments to help us to improve our manuscript.

Specific comments, that should be considered by the authors:

Page 4

"Genome assembly and annotation":

I still believe that the occurrence of the different BIAS in the three Papaver species, that were analysed by the authors, do not fit to this paragraph where genome assembly and annotation is described. As the BIA bouquet is the phenotype, that is studied, and the basis of their hypothesis, it should be described in an individual chapter. The data is available \diamond suppl. Fig. 2. According to this Fig., Papaver rhoeas produces low amount of Noscapine, Thebaine, Codeine, Morphine,

which supports the theory, that *P. rhoes* has a “primitive” BIA pathway. Also, the author nicely described this primitive pathway in their rebuttal: “*P. rhoeas* has a primitive metabolic pathway, leading to the formation of Dopamine and 4-Hydroxyphenylacetaldehyde, the precursor of morphinans (the genes on the pathway are: PrhUNG23530.0 (TYDC), Prh05G30560.0 (TryAT)). It also has the NCS (PrhUNG12800.0), encoding the enzyme to convert Dopamine and 4-Hydroxyphenylacetaldehyde into (S)-Norcoclaurine, the first step of morphinan biosynthesis.” I think this information is very interesting not only to me, but also to the broad audience. I would recommend to include it in the main manuscript.

RESPONSE: We have revised results section “Genome assembly and annotation” and separate this part into “Quantification of noscapine and morphinans in three *Papaver* species”.

In addition, the information about “*P. rhoeas* has a primitive metabolic pathway leading to the formation of morphinan precursors such as dopamine, 4-Hydroxyphenylacetaldehyde and (S)-Norcoclaurine encoded by TYDC (PrhUNG23530.0) and TryAT (Prh05G30560.0), NCS (PrhUNG12800.0), respectively” was added to discussion part in line 445-448.

Page 6

Line 93: „In addition, a collinearity confirming a lack of the ... in *Papaver* genus” ... please check the grammar of the sentence.

RESPONSE: We have revised this sentence as “In addition, a collinearity analysis of three *Papaver* species with grape (*Vitis vinifera*) and ancestral eudicot karyotype genome (Supplementary Figs. 15, 16) confirmed a lack of the whole genome triplication (γ event) occurred in core eudicots in *Papaver* genus.”

Line 97:

The authors nicely explained in the rebuttal letter how they estimated the divergence time and refer to the suppl. Section 8.1. I would recommend to include this very clear explanations in the main manuscript. The paragraph in its actual format is in my opinion confusing.

RESPONSE: Thank you for your comments. We have revised the methods section "Phylogenomic analysis and divergence time estimation" by moving more detail from rebuttal letter.

Page 10 – associated to Fig.2:

I think this explanation should also be included in the main manuscript: “After reassignment of chromosome IDs, we found the BIA gene cluster were located at chr4 rather than chr11.” This is an important result of you re-assembly and explains the discrepancy between your previous and recent publications.

RESPONSE: We have revised our manuscript by adding "After reassignment of chromosome IDs,

we found the BIA gene cluster was located at chr4 rather than chr11." into results section "Genome assembly and annotation".

Page 11, Line 141:

*Sorry, you did not get my point. Although *P. somniferum* and *P. setigerum* share the WGD-1, only in *P. somniferum* the gene cluster including the 10 noscapine and the 5 morphinan genes evolved while in *P. setigerum*, only the "5 morphinan gene cluster" evolved. Thus, the post WGD diploidization differ strongly between the two species. I think this is the most intriguing result of your research. Also, having your analyses of BIAs in the three *Papaver* species in mind – "innovation of the alkaloid biosynthesis pathways" is not a correct term, as *P. rhoeas* – most likely resembling the ancestral state, as it did not experience any WGD event - already produces trace amounts of morphinan and noscapine. So, evolution & selection optimized the biosynthesis of both BIA types in *S. somniferum*.*

RESPONSE: We have revised our description in line 226-228: "These results suggest a post-WGD diploidization might have played a part in the optimization of the alkaloid - morphinan - biosynthesis pathways in ancestor of *P. somniferum* and *P. setigerum*." as you suggested.

Page 14, "dispersed duplications"

Thanks for your explanation. This is very interesting. I would suggest to include a shortened version of this explanation to your main manuscript, maybe this would be helpful and interesting for the broader audience. The term "dispersed duplication" alone is not very informative.

RESPONSE: We have included a shortened version of this explanation. "Specifically, we found five genes including *PSCXE1*, *PSAT1*, *PSMT3*, *SALAT*, and *THS* were assembled into BIA gene cluster by putative dispersed duplications (the gene and their inferred original copy are not adjacent) while *CYP82X2* was a putative tandem duplication of *CYP82X1* (Fig. 4b, Supplementary Figs. 22-30)."

Page 18: Line 215

**P. rhoeas*, that possesses the "basic" genetic components of BIA pathway, produces morphinan and noscapine in traces, thus you should adjust your statement. Its rather not "in principle,....should allow....". It seems to be a fact, that the simple and ancestral pathway truly is sufficient to produce traces of morphinan and noscapine. Especially concerning your explanation in the rebuttal (see above).*

RESPONSE: Thank you for you comments. We have revised the description as "The genetic components of BIA pathway should allow the biosynthesis of morphinan and noscapine albeit at their original loci before clustering, raising questions on the necessity of gene clustering in evolution."

Page 19, Line 223: no, you did not revise the sentence. The “compared to” is still missing.....

RESPONSE: Thanks for your comments. We double checked the sentence and the context, and make sure that our sentence is correct. Here, we would like to describe the genes duplicated from A to B, where A is " putative ancestral copies at remote origin loci of largely low and non-tissue-specific expression", and B is "the BIA gene cluster that displays a high and coordinated expression in stem".

REVIEWERS' COMMENTS

Reviewer #1 (Remarks to the Author):

The authors have addressed my comments sufficiently, and only a few minor issues remain. This paper presents some interesting data, and while I disagree on some interpretations, there has been sufficient clarification on the points I raised previously.

Minor comments:

- Line 154: The difference between forward, backward, and patchwork models could be explained more clearly. Not obvious what is the testable difference here without reading the referenced papers.

- I still feel the authors are mis-representing the likelihood of [tandem duplication + deletion] vs. [dispersed duplication], and that the former is a much more parsimonious model. But they have acknowledged more clearly the tandem duplication hypothesis and the new Figure S32 is a good addition. I still feel that Figure 4 is misleading in suggesting that these are putative dispersed duplicates when they are more likely to be tandem duplicates. The authors still insist in the Discussion that old tandem duplications is less parsimonious. What is the evidence for this? Can the authors point to frequent occurrences of dispersed duplicates in other taxa? Because tandem duplication happens all the time. But I'm fine with the authors leaving this as a clearly stated opinion about what they think is parsimonious.

- There remain grammar/language issues throughout.

[Editor: Reviewer #3 states in Remark to Editor section that (s)he is satisfied with the revision.]

Reviewer #4 (Remarks to the Author):

Dear Editors, dear authors,

All issues and questions raised were addressed by the authors. The results are complex, but the manuscript in its actual version is interesting to read and provides a lot of input. Thanks to the authors for their great work.

Thank you for all your valuable comments. We have provided a response letter addressing all the issues raised by the reviewers. For clarity, all reviewer comments or quoted contents are in italicized fonts. A point-to-point response to each comment is provided in normal fonts. References to revised manuscript contents are also provided where needed.

REVIEWER COMMENTS

Reviewer #1 (Remarks to the Author):

The authors have addressed my comments sufficiently, and only a few minor issues remain. This paper presents some interesting data, and while I disagree on some interpretations, there has been sufficient clarification on the points I raised previously.

Minor comments:

- Line 154: The difference between forward, backward, and patchwork models could be explained more clearly. Not obvious what is the testable difference here without reading the referenced papers.

RESPONSE: We have revised the introduction of the three pathway evolution models to make it clear.

- I still feel the authors are mis-representing the likelihood of [tandem duplication + deletion] vs. [dispersed duplication], and that the former is a much more parsimonious model. But they have acknowledged more clearly the tandem duplication hypothesis and the new Figure S32 is a good addition. I still feel that Figure 4 is misleading in suggesting that these are putative dispersed duplicates when they are more likely to be tandem duplicates. The authors still insist in the Discussion that old tandem duplications is less parsimonious. What is the evidence for this? Can the authors point to frequent occurrences of dispersed duplicates in other taxa? Because tandem duplication happens all the time. But I'm fine with the authors leaving this as a clearly stated opinion about what they think is parsimonious.

RESPONSE: Thanks for acceptance of the alternative hypothesis of the gene evolution (Fig. S37 (Supplementary Fig. 32 in previous version)). Based on our current data and analysis strategies, we obtained the evolutionary model of BIA related genes as a working hypothesis. Inclusion of sequencing data from additional species may lead to an updated evolutionary model or equally possible alternative hypotheses. We revised the statements in Discussion section as "Alternative explanations of the BIA gene cluster formation such as "old tandem duplications" are equally possible but shall be tested with more sequencing data and genome analyses".

- There remain grammar/language issues throughout.

RESPONSE: We have revised the remaining grammar/language issues in our manuscript.

[Editor: Reviewer #3 states in *Remark to Editor* section that (s)he is satisfied with the revision.]

RESPONSE: Thanks for all of your valuable comments to help us improve our manuscript.

Reviewer #4 (Remarks to the Author):

Dear Editors, dear authors,

All issues and questions raised were addressed by the authors. The results are complex, but the manuscript in its actual version is interesting to read and provides a lot of input. Thanks to the authors for their great work.

RESPONSE: Thanks for all of your valuable comments to help us improve our manuscript.